# A weakened interface in the P182L variant of HSP27 associated with severe Charcot-Marie-Tooth neuropathy causes aberrant binding to interacting proteins

T Reid Alderson[1,2,†,‡] (iD), Elias Adriaenssens[3,†] (iD), Bob Asselbergh[4,5] (iD), Iva Pritišanac[6] (iD), Jonas Van Lent[3], Heidi Y Gastall[1], Marielle A Wälti[2], John M Louis[2], Vincent Timmerman[3,*] (iD), Andrew J Baldwin[1,**] (iD) & Justin LP Benesch[1,***] (iD)

## Abstract

HSP27 is a human molecular chaperone that forms large, dynamic oligomers and functions in many aspects of cellular homeostasis. Mutations in HSP27 cause Charcot-Marie-Tooth (CMT) disease, the most common inherited disorder of the peripheral nervous system. A particularly severe form of CMT disease is triggered by the P182L mutation in the highly conserved IxI/V motif of the disordered C-terminal region, which interacts weakly with the structured core domain of HSP27. Here, we observed that the P182L mutation disrupts the chaperone activity and significantly increases the size of HSP27 oligomers formed *in vivo*, including in motor neurons differentiated from CMT patient-derived stem cells. Using NMR spectroscopy, we determined that the P182L mutation decreases the affinity of the HSP27 IxI/V motif for its own core domain, leaving this binding site more accessible for other IxI/V-containing proteins. We identified multiple IxI/V-bearing proteins that bind with higher affinity to the P182L variant due to the increased availability of the IxI/V-binding site. Our results provide a mechanistic basis for the impact of the P182L mutation on HSP27 and suggest that the IxI/V motif plays an important, regulatory role in modulating protein–protein interactions.

**Keywords** charcot-marie-tooth disease; intrinsically disordered regions; molecular chaperones; NMR spectroscopy; short linear motif
**Subject Categories** Neuroscience; Translation & Protein Quality
**The EMBO Journal (2021) 40: e103811**

## Introduction

HSP27 (also known as HSPB1) is a systemically expressed small heat-shock protein (sHSP) that performs diverse functions under basal and stressful cellular conditions (Kampinga *et al*, 2015; Janowska *et al*, 2019; Mogk *et al*, 2019). With significant roles in the maintenance of protein homeostasis, regulation of the redox environment, prevention of apoptosis, and stabilization of the cytoskeletal network, the biological activity of HSP27 is critical to overall cellular health (Bruey *et al*, 2000; Arrigo, 2001; Gusev *et al*, 2002; Basha *et al*, 2012; Haslbeck & Vierling, 2015). Dysregulation of the activity or expression of HSP27 can result in debilitating diseases, including cancers (Ciocca & Calderwood, 2005; Straume *et al*, 2012), neurodegenerative diseases (Outeiro *et al*, 2006), and neuropathies (Evgrafov *et al*, 2004). More than 30 heritable HSP27 mutations are implicated in Charcot-Marie-Tooth (CMT) disease (Nefedova *et al*, 2015; Adriaenssens *et al*, 2017; Echaniz-Laguna *et al*, 2017; Muranova *et al*, 2020; Vendredy *et al*, 2020), a group of neuropathies that affects *ca*. 1 in 2,500 individuals and is the most common inherited disorder involving the peripheral nervous system (Barisic *et al*, 2008; Timmerman *et al*, 2014). CMT disease is characterized by progressive demyelination (type 1 CMT), axonal loss (type 2 CMT), or their combination (Rossor *et al*, 2013; Pipis *et al*, 2019). When the affected axons exclusively include motor neurons, the disease is referred to as distal hereditary motor neuropathy (dHMN) (Rossor *et al*, 2013; Beijer & Baets, 2020). Transgenic HSP27 mouse models develop CMT disease-like symptoms

1  Chemistry Research Laboratory, University of Oxford, Oxford, UK
2  Laboratory of Chemical Physics, National Institutes of Health, Bethesda, MD, USA
3  Peripheral Neuropathy Research Group, Department of Biomedical Sciences, Institute Born Bunge, University of Antwerp, Antwerpen, Belgium
4  Neuromics Support Facility, VIB Center for Molecular Neurology, VIB, Antwerpen, Belgium
5  Neuromics Support Facility, Department of Biomedical Sciences, University of Antwerp, Antwerp, Belgium
6  Molecular Medicine Program, The Hospital for Sick Children, Toronto, ON, Canada
   *Corresponding author. Tel: +32 3 265 10 24; E-mail: vincent.timmerman@uantwerpen.be
   **Corresponding author. Tel: +44 0 1865 275420; E-mail: andrew.baldwin@chem.ox.ac.uk
   ***Corresponding author. Tel: +44 0 1865 285420; E-mail: justin.benesch@chem.ox.ac.uk
   †These authors contributed equally to this work
   ‡Present address: Department of Biochemistry, Toronto, ON, Canada

(D'Ydewalle *et al*, 2011), thereby implicating HSP27 as a direct driver of CMT disease onset.

HSP27 contains three domains: a conserved α-crystallin domain (ACD) that is flanked by an N-terminal domain (NTD) and a C-terminal region (CTR), which is disordered and contains a highly conserved IxI/V motif. Similar to the roles of other intrinsically disordered protein regions (Wright & Dyson, 2014), the CTRs of sHSPs play key roles in promoting solubility and contributing to the regulation of interactions with other proteins (Carver *et al*, 2017; Holt *et al*, 2019), including liquid–liquid phase separation (Morelli *et al*, 2017). Biophysical studies on HSP27 and related sHSPs have shown that intermolecular contacts between all three domains (Hochberg & Benesch, 2014; Haslbeck & Vierling, 2015; Clouser *et al*, 2019; Kaiser *et al*, 2019) facilitate the assembly of large, dynamic oligomers with an average mass near 500 kDa (Jehle *et al*, 2010; Baldwin *et al*, 2011, 2012; McDonald *et al*, 2012; Hochberg *et al*, 2014). In mammalian sHSPs, the central residue in the IxI/V motif is generally proline, and this short linear motif (SLiM) is involved in binding to the structured ACD. High-resolution structures have been obtained for the ACD (Rajagopal *et al*, 2015) bound to a peptide containing the IxI/V motif (Hochberg *et al*, 2014). In addition, the IxI/V motif can facilitate interactions between HSP27 and other proteins that contain this motif, including the HSP70 co-chaperone Bcl-2-associated anthanogene-3 (BAG3) (Rosati *et al*, 2011), which contains two IxI/V motifs and forms an HSP27-BAG3-HSP70 ternary complex that enables substrate transfer between the two chaperones (Rauch *et al*, 2017).

While the majority of CMT-associated mutations in *HSPB1* reside in the structured ACD (Adriaenssens *et al*, 2017; Weeks *et al*, 2018), mutations within the disordered NTD and CTR also cause CMT disease (Chalova *et al*, 2014; Muranova *et al*, 2015). One of the most clinically severe variants is caused by the P182L mutation, which falls in the intrinsically disordered CTR and changes the highly conserved Pro residue within the IxI/V motif ($^{181}$Ile-Pro-Val$^{183}$ in HSP27) to a Leu. Symptoms caused by the P182L mutation typically manifest within the first 5 years of life, whereas mutations in the ACD generally result in adult-onset symptoms (Dierick *et al*, 2008). Pro182 is the only residue within the IxI/V motif that is implicated in CMT disease; however, the molecular basis of the impacts of the P182L mutation remains unknown.

Here, we sought to understand the biophysical significance of the P182L mutation in HSP27. We observed that the P182L mutation significantly increases the average molecular mass of soluble HSP27 oligomers and disrupts its ability to prevent substrate aggregation. Expansion microscopy, which enables imaging at the nanoscale, confirmed that the P182L variant forms larger oligomers *in vivo*. These large P182L oligomers were also detected in induced pluripotent stem cells (iPSCs) from a CMT patient containing the P182L mutation. We investigated the binding of the IxI/V motif to the ACD with NMR spectroscopy, revealing that the affinity of the P182L variant is decreased by an order of magnitude. This suggests that the weakened affinity for the ACD causes the IxI/V motif binding site to become more exposed for other IxI/V-containing HSP27-interacting proteins. We verified this hypothesis by demonstrating that a number of IxI/V-containing proteins, including BAG3, an HSP70 co-chaperone with two IPV motifs, bind with enhanced affinity for the P182L variant of HSP27. We validated these results with targeted mutations that either disrupt

the IxI/V motif of the interactor or the ability of HSP27 to bind to IxI/V motifs. Given the highly conserved nature of the IxI/V across sHSPs, our findings suggest that this motif plays a regulatory role in mediating interactions among this important family of molecular chaperones.

# Results

To shed light on the structural impact of the P182L mutation (Fig 1A and B), we investigated the biophysical and functional properties of the wild-type (WT) and P182L variants of full-length HSP27. Our experiments also used the isolated ACD (Fig 1C) to characterize the binding of the IxI/V motif in its WT and P182L forms. The ACD folds into a β-strand-rich dimer (Fig 1D) and the IxI/V motif binds to the ACD in the groove located between the β4 and β8 strands (Fig 1D) through hydrophobic interactions and hydrogen bonds to the backbone atoms of V111, T113, and L157 (Fig 1E). In WT HSP27, the IxI/V motif is manifested as $^{181}$IPV$^{183}$, and the Ile and Val residues penetrate the hydrophobic β4/β8 groove in the ACD (Fig 1E).

### The P182L variant has impaired chaperone activity

To determine the effect of the P182L mutation on the chaperone activity of HSP27, we purified recombinant human HSP27 and the CMT-implicated P182L variant and tested their ability to prevent the aggregation of two model substrate proteins, malate dehydrogenase (MDH) and insulin. The WT form of HSP27 proved highly effective at inhibiting the aggregation of both substrates (Fig 1F and G, red traces), as observed previously and indicative of its potent anti-aggregation activity (Alderson *et al*, 2019). However, the P182L variant was unable to prevent the aggregation of either substrate (Fig 1F and G, blue traces), revealing a drastic loss of function *in vitro* (Appendix Fig S1A). Indeed, based on the elevated light scattering in the presence of the chaperone, the P182L form of HSP27 appeared to co-aggregate with MDH and insulin. Control experiments indicated that the P182L variant does not aggregate appreciably when incubated alone at high concentrations and 40°C (Appendix Fig S1B) and is correctly folded (Appendix Fig S1C), suggesting that P182L is stable in solution and the loss of P182L chaperone activity is not due to misfolding or self-aggregation. This is consistent with the CMT-causing P182S mutant exhibiting similar thermal stability to the WT protein, with respective melting temperatures of 64 and 70°C (Chalova *et al*, 2014). The absence of self-aggregation indicates that P182L promotes substrate aggregation under these conditions, as has been observed for other sHSP systems (Ungelenk *et al*, 2016), including the cardiomyopathy-causing R120G variant of αB-crystallin (Bova *et al*, 1999). The P182L mutation therefore prevents HSP27 from attenuating the aggregation of these client proteins.

### P182L forms large oligomers both in vitro and in vivo

To understand the reduced chaperone activity of P182L, we used size exclusion chromatography coupled to multi-angle light scattering (SEC-MALS) to characterize its oligomeric properties in solution. For WT HSP27 and its isolated ACD, the SEC-MALS data

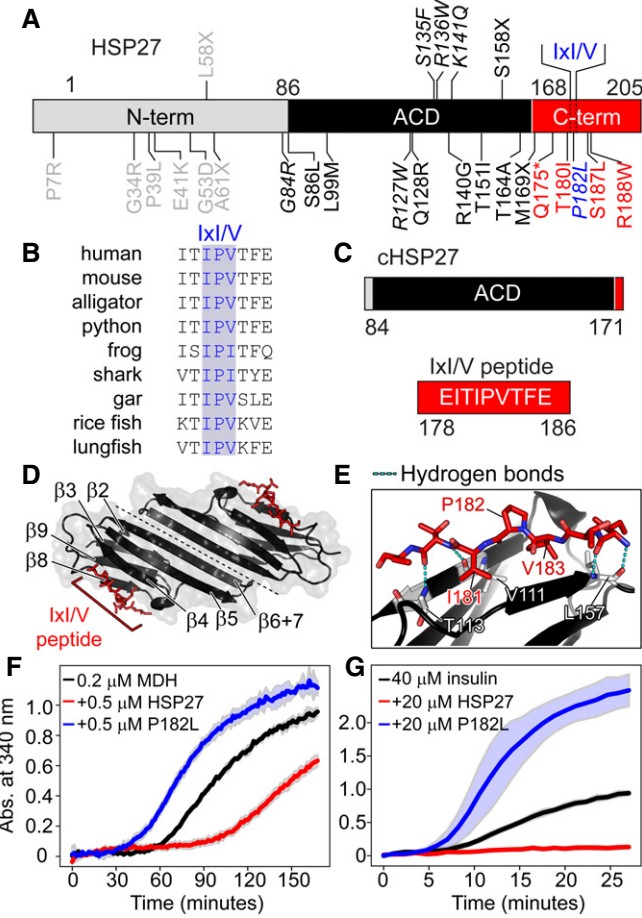

**Figure 1. The P182L mutation alters the IxI/V motif of HSP27 and disrupts its chaperone activity.**

A Domain architecture of human HSP27: N-terminal domain (N-term), α-crystallin domain (ACD), and C-terminal region (C-term). The conserved IxI/V motif in the C-terminal region is indicated. Missense, nonsense, and frameshift mutations associated with CMT disease and dHMN are shown. The P182L mutation in the IxI/V motif is blue. Residues with multiple mutations are italicized (e.g., P182L, P182S). The asterisk denotes a nonsense mutation and the X indicates a frameshift mutation.

B Amino acid sequence alignment of IxI/V motifs and adjacent residues from vertebrate HSP27 orthologues: human (*Homo sapiens*), mouse (*Mus musculus*), alligator (*Alligator mississippiensis*), python (*Python bivittatus*), frog (*Xenopus laevis*), shark (*Callorhinchus milii*), gar (*Lepisosteus oculatus*), rice fish (*Oryzias latipes*), and lungfish (*Protopterus annectens*).

C Domain boundaries of the ACD used in this study (cHSP27) and residues in the C-terminal IxI/V peptide.

D Three-dimensional structure of the ACD dimer (black, PDB: 4mjh) bound to the IxI/V peptide from the C-terminal region (red). The β-strands of the ACD are numbered.

E Zoomed-in region from (D) showing the contacts made between the IxI/V peptide and the ACD.

F Malate dehydrogenase (MDH; 0.2 μM) was incubated at 40°C in the absence (black) or presence of 0.5 μM WT HSP27 (red) or 0.5 μM P182L HSP27 (blue).

G Insulin (40 μM) was incubated at 40°C in the absence (black) or presence of 20 μM WT HSP27 (red) or 20 μM P182L HSP27 (blue). The y-axes depict the absorbance at 340 nm due to the formation of large aggregates. The solid lines represent the average of three replicates with the filled area reflecting ± one standard deviation.

revealed average molecular masses of approximately 500 and 20 kDa (Fig 2A), respectively. By contrast, the P182L variant formed large oligomers with an average molecular mass near 14 MDa (Fig 2A), almost a 30-fold increase over the WT protein. With a monomeric mass of 22.7 kDa, the P182L variant forms an oligomeric assembly comprising an average of *ca.* 620 subunits, whereas the WT protein forms oligomers centered on *ca.* 24 subunits. As deduced by SEC-MALS, the ensemble-averaged hydration radius of P182L at 38 nm is four-fold larger than the 9-nm oligomers formed by WT HSP27, consistent with a *ca.* 30-fold increase in mass.

To confirm these results by an independent method, we extracted whole cell protein lysates from HeLa cells stably overexpressing HSP27 WT or the P182L mutant and separated these protein lysates over a sucrose gradient. The respective fractions were loaded on Western blot and demonstrated a clear shift toward larger protein complexes for the P182L mutant (Fig 2B). Negative-stain electron microscopy (EM) also revealed the formation of larger P182L oligomers (Appendix Fig S2). Together, the SEC-MALS, sucrose gradient, and negative-stain EM data establish that the disease-causing P182L mutation shifts the oligomeric distribution of HSP27 toward significantly larger species.

To visualize these soluble oligomeric states of HSP27 in cells, we performed expansion microscopy, which allows imaging of HSP27 at the nanoscale and analysis of the distribution of soluble HSP27 in the cytoplasm. Expansion microscopy increases the effective resolution by more than four-fold compared to conventional microscopy (Chen *et al*, 2015). As a result, soluble HSP27 assemblies that were indistinguishable by standard confocal microscopy could be visualized individually in HeLa cells (Fig 2 C). Comparing the number of detectable spots per cytoplasmic volume shows that expression of the P182L mutant leads to a two-fold reduction in the density of HSP27 assemblies (Fig 2D). Furthermore, the fluorescence intensity of each spot is determined by the number of HSP27 subunits and therefore reflects indirectly the size of the oligomeric assemblies. Our measurements indicated that the P182L mutation increases the proportion of spots with higher intensities (Fig 2E).

Overexpression of the P182L mutant has been reported to cause increased formation of large insoluble HSP27-containing cytoplasmic aggregates (Ackerley *et al*, 2006; Almeida-Souza *et al*, 2010; Haidar *et al*, 2019). Indeed, upon transient transfection of V5-epitope-tagged P182L HSP27 in HeLa cells, we found that more than 25% of cells contained large, high-intensity aggregates visible by conventional microscopy, whereas comparable transient expression of WT HSP27 caused aggregation in fewer than 2% of cells (Appendix Fig S3A and B). This was also confirmed by Western blotting, which showed a greater insoluble fraction for P182L compared to WT HSP27, while the sum of both the soluble and insoluble fractions indicated equal expression levels (Appendix Fig S3C and D). To avoid that our expansion microscopy data would be confounded by the presence of large cytoplasmic aggregates, we only included cells without large aggregates and confirmed that average fluorescence intensities in expansion microscopy images were similar for both genotypes (Appendix Fig S3E–H). Furthermore, P182L and WT HSP27 have the same integrated spot intensities, defined as the product of the individual spot intensity and the

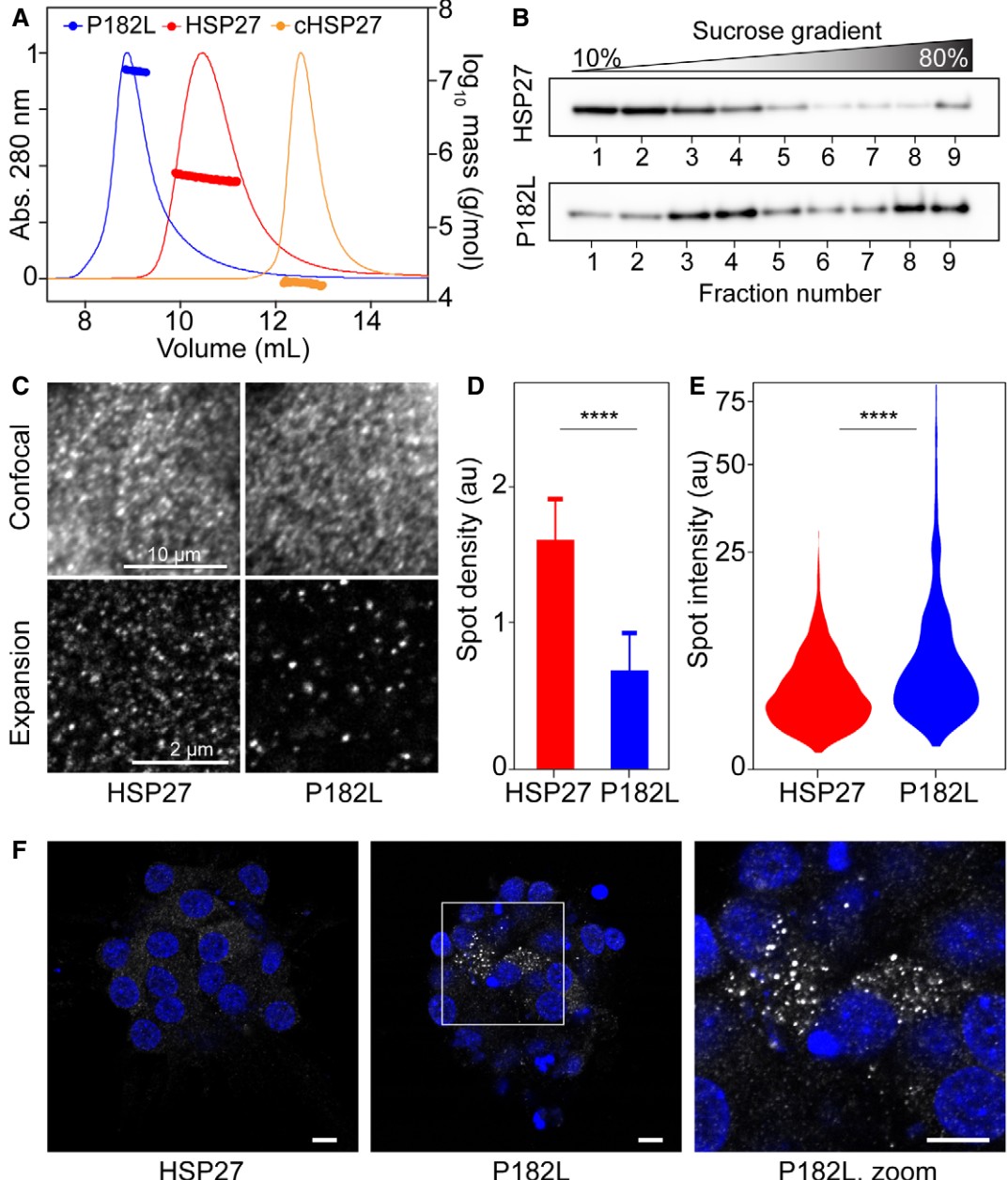

**Figure 2. The P182L variant of HSP27 forms larger oligomers both *in vitro* and *in vivo*.**

A   SEC-MALS data from WT HSP27 and the P182L variant, with cHSP27 included for comparison. All samples were injected at 40 μM concentration (monomer). The respective average molecular masses are 14 MDa, 470 kDa, and 18 kDa for P182L, WT HSP27, and cHSP27. The normalized UV signals are plotted on the left y-axis as solid lines, and the molecular masses are plotted on the right y-axis as circles.

B   Fractionation of HeLa cells overexpressing either HSP27 or the P182L mutant by a 10–80% sucrose gradient. Western blot against anti-V5 is shown.

C   Expression of V5-epitope-tagged HSP27 (WT or P182L mutant) in HSP27 knockout HeLa cells and immunostaining of HSP27. The scale bars correspond to the non-expanded dimensions, derived by dividing by the sample expansion factor of 4.3.

D   The P182L mutation decreases the density of HSP27 spots upon detection of local fluorescence intensity maxima and quantification of the number of detected spots in expanded samples ($n = 30$). The error bar is the standard deviation for the different cytoplasmic regions. The asterisks correspond to a $P < 0.0001$ as obtained from a *t*-test.

E   Distribution of the intensity of individual spots ($n = 3,342$) after Gaussian fitting of fluorescence intensity in the neighborhood of detected spots. The asterisks correspond to a $P < 0.0001$ as obtained from a Mann–Whitney *U* test. Data were plotted with a square-root transformation of the *y*-axis to improve the visualization of the complete distribution of intensities.

F   Immunostaining of HSP27 (white) in motor neurons differentiated from CMT patient-derived iPSCs that harbor the heterozygous P182L mutation reveals large, cytoplasmic aggregates (center and right panels). The control motor neurons that contain WT HSP27 do not show evidence of any aggregates. The nucleus is stained with Hoechst (blue). Scale bar, 10 μm.

Source data are available online for this figure.

spot density (Appendix Fig S3G), which additionally confirm that the P182L and WT forms are expressed at similar levels. Combined, these results show that assemblies of the P182L mutant are larger and more sparsely distributed in the cytoplasm, consistent with a higher oligomeric state *in vivo* than the WT.

P182L is a heterozygous mutation, so CMT patients will carry one copy of the mutant allele and one of the WT. We therefore wondered if the larger P182L oligomers might be composed of mixtures of WT and P182L subunits due to hetero-oligomerization. To this end, we first tested whether the P182L variant can hetero-oligomerize with WT HSP27 *in vitro*. Gel filtration chromatography showed that WT HSP27 assimilates into the larger P182L oligomers (Appendix Fig S4A). To verify whether hetero-oligomerization also occurs in cells, we extracted protein lysates from HeLa cells that overexpress the V5-epitope-tagged P182L variant on top of the endogenous WT protein. A Western blot analysis showed that the P182L variant and WT protein indeed hetero-oligomerize and that the P182L variant can hetero-dimerize with WT HSP27 by forming an intermonomer disulfide bond at C137 (Appendix Fig S4B), the lone cysteine residue. Hetero-oligomerization was additionally confirmed by the observations that P182L increased the insoluble fraction of endogenous WT HSP27 (Appendix Fig S3D), and that WT HSP27 co-immunoprecipitated with P182L HSP27 (Appendix Fig S4C). Together, these data demonstrate that the P182L variant hetero-oligomerizes with WT HSP27 *in vitro* and in HeLa cells.

Finally, we sought to establish in a CMT patient-derived cell line whether P182L also exists in a larger oligomeric form than the WT protein. To this end, we used iPSCs from a CMT patient harboring the heterozygous P182L mutation that we differentiated to spinal motor neurons. When immunostained for HSP27 and imaged with confocal microscopy, the P182L motor neurons contained many bright aggregate-like clusters (Fig 2F), which were absent in the control iPSC line (Fig 2F) that expresses the WT form of HSP27. Taken together, these biophysical data and functional assays demonstrate that the P182L mutation increases the size of oligomers both *in vitro* and *in vivo* toward very large states that have aberrant activity in preventing protein aggregation.

## The P182L mutation significantly lowers binding affinity for the IxI/V motif

To understand the molecular basis of the oligomeric dysregulation caused by the P182L mutation, we sought to characterize the binding interaction between the ACD and the IxI/V motifs of the WT form ($^{181}$IPV$^{183}$) and the P182L variant ($^{181}$ILV$^{183}$). Previous solution-state NMR studies of WT HSP27 observed resonances only from the disordered CTR (Carver *et al*, 1995; Alderson *et al*, 2017). We therefore turned to the isolated ACD, which forms a stable dimer (2 × 10 kDa) that has been characterized by solution-state NMR (Rajagopal *et al*, 2015; Clouser & Klevit, 2017; Alderson *et al*, 2019, 2020) and is known to bind to a peptide encompassing the IxI/V motif of the CTR (Delbecq *et al*, 2012; Hilton *et al*, 2013a; Hochberg *et al*, 2014; Hochberg & Benesch, 2014; Rauch *et al*, 2017; Freilich *et al*, 2018; Liu *et al*, 2018).

Using NMR spectroscopy, we determined the $K_d$ of the ACD binding to a peptide comprising the WT IPV motif (Fig 3A–C). Since the ACD was uniformly $^{15}$N labeled, each amide bond in the

protein contributed a signal in the 2D $^1$H-$^{15}$N heteronuclear single quantum coherence (HSQC) spectrum (Fig 3A and B). During the peptide titration, increasing the amount of added peptide led to the disappearance of resonances from ACD residues in the β4/β8 groove ("slow" exchange) (Fig 3A) and progressive changes in the chemical shifts of other residues ("fast" exchange) (Fig 3B), *i.e.*, chemical shift perturbations (CSPs). To a first approximation, analogous to a procedure described previously (Delbecq *et al*, 2012), we assumed that the group of continuously varying residues were in fast exchange and fit these resonances to equation 2 (see Materials and Methods) yielding a global $K_d$ of *ca*. 125 μM at 25°C (Fig 3C). Next, using the same approach, we measured the binding affinity of the peptide bearing the P182L mutation implicated in the onset of CMT (Fig 3A–C), revealing that the mutation lowers the binding affinity for the ACD by an order of magnitude from *ca*. 125 μM to 1 mM (Fig 3C), corresponding to a $\Delta\Delta G$ of *ca*. 5 kJ/mol at 25°C. Importantly, the resonances in the ACD reporting on the binding were the same for both the WT and P182L peptides, indicating the binding site is the same in both cases (Fig 3D and E).

In addition to monitoring structural changes to the protein upon binding, we also collected 1D NMR spectra of the isolated WT and P182L peptides. For both peptides, the resonances cluster in the disordered region of the $^1$H spectrum (8–8.5 ppm) (Appendix Fig S4E), indicating the absence of appreciable secondary structure in either peptide. This rules out that the difference in binding affinity between the P182L and WT peptides comes from a variation in their secondary structure. Our results thus demonstrate that the P182L peptide binds in the same site as the WT peptide but with significantly lower affinity, and elicits common structural changes to the ACD upon binding.

## The IxI/V motif interconverts between multiple bound conformations

Based on the changes in chemical shift between free and bound states, the residues in the ACD that are the most significantly impacted by IxI/V binding are clustered in and near the β4/β8 groove, and exhibit slow chemical exchange (Fig 3D and E). Over the course of a titration, intensity from these resonances is lost and then re-appears in locations corresponding to the bound state. Using the software TITAN, which is capable of simultaneously analyzing the residues in both exchange regimes (Waudby *et al*, 2016), we determined $K_d$s of 132 μM for WT and 1,285 μM for P182L (Appendix Fig S5A–D, Appendix Table S2). These values are very close to those obtained from the CSP analysis of only the fast-exchanging residues and reinforce how the affinities of WT and P182L peptides for the ACD differ by an order of magnitude.

At high peptide concentrations, where, according to our $K_d$ measurements, we would expect the bound state to be almost exclusively populated, we noticed that the observed signal intensities for the slow-exchanging resonances were unexpectedly low and the fitted $R_2$ values of the bound state by TITAN extremely large (Appendix Fig S6A). These observations suggest that the "bound" state itself undergoes chemical exchange with a third, previously unidentified, conformational state in which the peptide is also bound. To test this hypothesis, we obtained 2D NMR spectra of the ACD in the presence of a 10-fold molar excess of peptide

as a function of increasing temperature from 25 to 50°C. By increasing the temperature, the rate of interconversion between the two bound forms will increase, and we should see the reappearance of signals for the broadened resonances. Indeed, at elevated temperatures, signal intensity for the bound state began to recover to the expected levels (Appendix Figs S6B and S7B and C). These results reveal an unexpected and interesting detail in the binding of peptides with the ACD domain: in addition to a free and an initial bound state, a further bound state becomes populated at high peptide concentrations.

### The P182L IxI/V motif binds with an attenuated association rate

To determine the origin of the weakened binding interaction in the P182L variant, we turned to Carr–Purcell–Meiboom–Gill (CPMG) relaxation dispersion (RD) (Palmer *et al*, 2001; Sugase *et al*, 2007b; Hansen *et al*, 2008b; Baldwin & Kay, 2009; Demers & Mittermaier,

2009). CPMG RD experiments can simultaneously probe the kinetics (association and dissociation rate constants; $k_{off}$, $k_{on}$), thermodynamics (populations of the interconverting states), and structural changes (differences in chemical shifts; $|\Delta\omega|$) involved in the interconversion between the principally populated ground state and a sparsely populated, conformationally excited state, whose fractional population is on the order of a few percent (Palmer *et al*, 2001; Sugase *et al*, 2007b; Hansen *et al*, 2008a, 2008b).

We interrogated peptide binding kinetics and thermodynamics with $^{15}$N CPMG RD experiments on the isotopically labeled ACD in the presence of unlabeled WT or P182L peptide. This experiment probes the $^{15}$N transverse relaxation rate ($R_{2,eff}$) as a function of the number of 180° refocusing pulses ($\nu_{CPMG}$), and variations in $R_{2,eff}$ are indicative of millisecond conformational exchange between two or more states. In the absence of peptide, resonances in the β4/β8 groove resonances showed no variation in $R_{2,eff}$ with the refocusing pulse frequency (Appendix Fig S8A

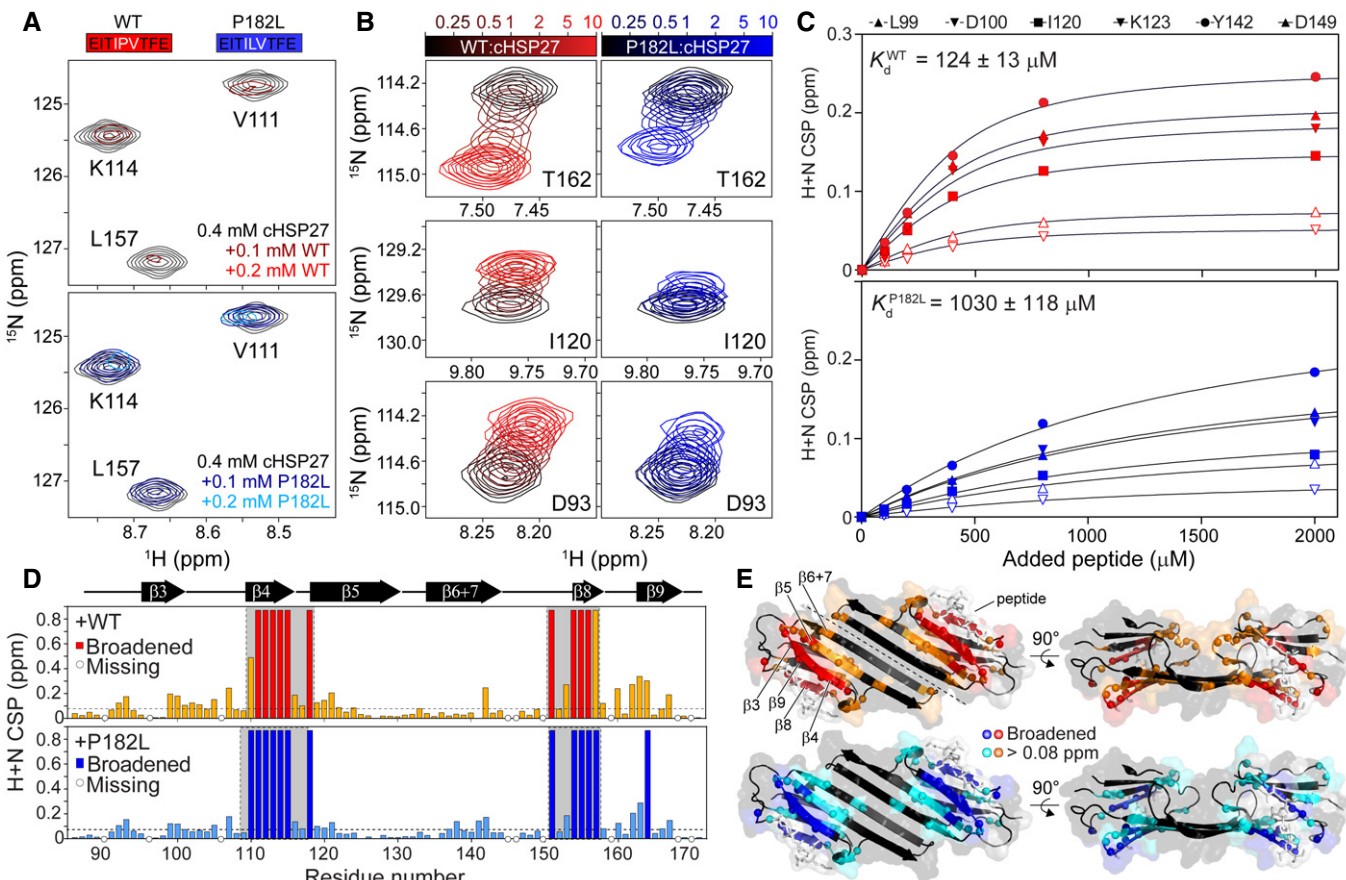

**Figure 3. The P182L mutation diminishes the affinity of the IxI/V motif for cHSP27.**

A   Amino acid sequences of the WT peptide (red) and P182L peptide (blue). The IxI/V motif is indicated with white text. Below: regions of 2D $^1$H-$^{15}$N HSQC spectra of $^{15}$N-labeled cHSP27 (black) in the presence of increasing amounts of WT (red) or P182L peptide (blue). Resonances in the β4 and β8 strands broaden and disappear upon peptide binding, indicative of slow-to-intermediate exchange.

B   Zoomed-in regions of resonances during the peptide titrations. The color bars indicate the amount of added WT or P182L peptide.

C   The combined $^1$H$^N$ and $^{15}$N chemical shift changes, denoted as chemical shift perturbation (CSP), are plotted as a function of added peptide concentration. Lines indicate the best fit to equation 2. The global $K_d$ ± one standard deviation is shown in the upper left. Equation 2 only applies to resonances that are in fast exchange; the resonances in panel A therefore cannot be fit in this manner.

D   CSPs shown as a function of residue number for the WT (red) and P182L peptide (blue).

E   X-ray structure of cHSP27 bound to the WT peptide (PDB: 4mjh), shown in white sticks, with the results from (D) plotted onto the structure.

and B). Adding a small amount of WT peptide (*ca.* 2% ACD-peptide complex), however, led to large variations in $R_{2,\text{eff}}$ with the refocusing pulse frequency, which are indicative of conformational exchange on the millisecond timescale (Fig 4A and B, Appendix Figs S8A and B, and S9A). The magnitude of the effect depended approximately linearly on the amount of added peptide (Appendix Fig S9B and C), which revealed that the conformational exchange event is intermolecular and the experiment effectively monitors peptide binding. Similar data were obtained for the P182L peptide-ACD complex (Fig 4A and B, Appendix Figs S8B and S9D and E), albeit with more peptide required to achieve a similar effect, further supporting the notion that the P182L peptide binds with lower affinity than the WT peptide (Fig 3C). The residues that yield dispersions are largely within 4 Å of the peptide in the bound state, based on the crystal structure of the peptide-bound ACD (Appendix Fig S8C and D).

To extract kinetic and structural information about peptide binding, we fit the CPMG RD data that were recorded at two static magnetic field strengths initially using a naïve two-state model. This model assumes interconversion between a free (unbound) and peptide-bound form of the ACD. The $^{15}$N |Δω| values obtained from CPMG RD show that similar ACD regions experienced conformational exchange in both peptide-bound states (Fig 4C, *ii. iii.*, Appendix Figs S8A and B, and S9A). Moreover, the fitted $^{15}$N |Δω| values were highly similar for both the WT and P182L peptides, indicating that the peptides bind in a similar mode to the same site on the ACD (Appendix Fig S10, Appendix Table S1), as expected from the titration data (Fig 3D and E). The fitted |Δω| values from the CPMG experiments on samples with a bound state populated to ca. 2% were consistent with the chemical shift changes observed more directly in the titration data (Appendix Fig S6B–E) suggesting that the CPMG RD and HSQC experiments were detecting the same end state. The fitted values for $k_{\text{off}}$ were very similar for P182L and WT peptides, at 610 ± 39/s and 515 ± 53/s, respectively (Fig 4C *i*). This points toward the origin of the reduced binding affinity displayed by P182L deriving from an attenuated association rate.

While the observed chemical shift changes between free and bound states observed in the CPMG and titration experiments were quantitatively consistent, we noted a discrepancy between the measured $K_{\text{d}}$s. In the CPMG and titration analyses, respectively, we obtained values of 1,762 ± 120 and 124 ± 13 μM for the WT $K_{\text{d}}$, and 8,373 ± 534 μM and 1,030 ± 118 μM for the P182L $K_{\text{d}}$ (Appendix Table S2). The ΔΔ$G$ values, between WT and mutant, are highly consistent when measured using either the CPMG or titration approach, but the absolute values vary by roughly an order of magnitude. To account for this, we noted that our titration analysis had already revealed the existence of (at least) 3 conformational states, and that upon addition of peptide, we can infer the presence of at least two bound states from the significantly increased observed $R_2$ values of residues in the vicinity of the β4/β8 groove. When analyzing CPMG RD data, it is common to assume that the $R_2$ values of the two states are identical (Palmer *et al*, 2001). However, in certain situations (Hansen *et al*, 2009; Baldwin *et al*, 2012), failing to account for differences in the $R_2$ rates can lead to incorrectly fitted values of $p_{\text{B}}$ (Ishima & Torchia, 2006). To avoid this possibility, we constrained our two-state CPMG analysis fit by enforcing the $K_{\text{d}}$ values to match those observed in the titration, and allowed the

$R_2$ of the bound state to be a free parameter. The quality of the resulting fits we obtained was excellent, resulting in slightly lowered reduced $\chi^2$ values over the naïve analysis where $R_{2,\text{unbound}} = R_{2,\text{bound}}$. Notably, the fitted |Δω| values were essentially unchanged in the two different analysis methods (Appendix Fig S11), indicating that the CPMG and titration data are in quantitative agreement (Appendix Fig S11). This result demonstrates that, for the residues in the vicinity of the β4/β8 groove, the bound state is undergoing conformational exchange on an "intermediate" timescale under these conditions. Our observations lend insight into the mechanism by which P182L has a lowered affinity versus the WT peptide for the ACD: when comparing the on and off rates determined by CPMG and the titration, in all forms of analysis, the larger difference is in the on rate (Appendix Tables S2 and S3).

## The P182L IxI/V motif samples a larger range of conformational space

To understand the mechanistic basis of the slower on rate of the P182L peptide, we hypothesized that the rigid Pro residue in the unbound WT IxI/V motif would restrict overall flexibility of the adjacent Ile and Val residues, thereby pre-arranging the IxI/V motif for facile binding to the ACD. By contrast, the P182L mutation would remove the torsional constraints imposed by the restrictive Pro side chain and enable easier exploration of the accessible conformational space.

To test these hypotheses, we performed 200 ns of all-atom molecular dynamics (MD) simulations on peptides of the same compositions used for the NMR binding experiments (Fig 4D and E). The φ and ψ dihedral angles of P182 (WT) or L182 (P182L) directly impact the conformation of the adjacent I181 and V183 residues within the IxI/V motif. We computed the free energy landscape for both the WT and P182L peptides as a function of φ and ψ dihedral angles for P182 or L182 (Fig 4D). As expected, the analysis revealed that the central Pro residue in the WT IxI/V motif significantly limits the sampling of backbone φ and ψ dihedral angles to a narrow region in Ramachandran space (Fig 4D). Notably, the conformation of the bound state of the IxI/V motif, as determined by X-ray crystallography (Hochberg *et al*, 2014), exhibits similar P182 φ/ψ angles (Fig 4D). By contrast, the P182L variant explores a much larger area of Ramachandran space (Fig 4D), and some of the other P182L conformers no longer adopt the necessary conformation for binding to the β4/β8 groove of the ACD. For example, near φ/ψ + 50°/−50°, large rotations about the I181 CO–P182 N and P182 CO–V183 CA bonds have moved the side chains of I181 and V183 out of plane (Fig 4D), which would sterically clash with the atoms in the β4/β8 groove.

In addition to a comparison of the φ/ψ dihedral angles about the P182 or L182 residue, we performed a principal component analysis (PCA) over the MD trajectories to capture variations between the predominant structural and dynamical features of the two peptides. The conformations of the WT peptide predominantly occupy two clusters in the space of the first and second principal components (Fig 4E), whereas the P182L variant instead shows a markedly broader distribution (Fig 4E). The larger spread in the PC space of the P182L peptide reflects a broader sampling of conformational space (Fig 4E). Our results indicate that one effect of the P182L mutation is an increase in conformational flexibility, which causes

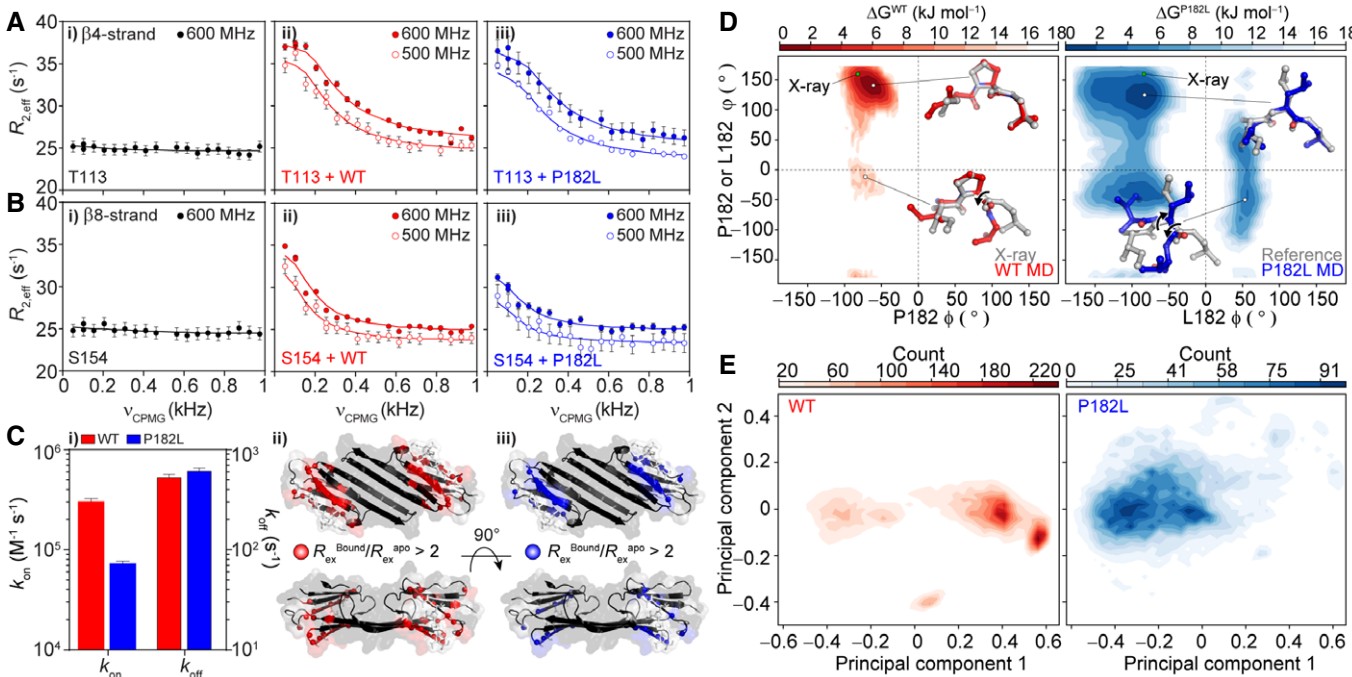

**Figure 4. The P182L mutation attenuates the IxI/V association rate constant.**

A, B    *i* $^{15}$N CPMG RD data for residues in the β4 strand (T113) and the β8 strand (S154) in the absence of added peptide. CPMG data are shown for the same residues in the presence of *ii. ca.* 2% cHSP27-WT peptide complex or *iii. ca.* 2% of the cHSP27-P182L peptide complex. Uncertainties in $R_{2,eff}$ values are derived from the standard deviations of fitted peak intensities obtained from duplicated $\nu_{CPMG}$ values.

C    The association ($k_{on}$) and dissociation ($k_{off}$) rates for WT (red) and P182L (blue) IxI/V peptide binding to the ACD, calculated directly from the CPMG RD data. Residues that are impacted by the presence of a small amount of peptide are indicated by red (*ii.*, WT) and blue (*iii.*, P182L) spheres. In both cases, residues in similar ACD regions show enhanced $R_{ex}$, indicative of chemical exchange at similar sites. Error bars were obtained using standard error propagation and represent one standard deviation from the calculated value.

D    *i* Free energy landscape of the WT (left) and P182L (right) peptides calculated from the MD trajectories as a function of the P182 or L182 φ/ϕ angles. A lower value of ΔG corresponds to a higher relative population. Select IxI/V conformations extracted from the MD simulations are shown for the indicated φ/ϕ angle pairs in red (WT MD) and blue (P182L MD). The conformation of the IxI/V motif in the crystal structure is shown in gray (X-ray) and the P182L reference structure in gray (Reference), as obtained with *in silico* mutagenesis using PyMol. The black arrows indicate rotations about the φ or ϕ angles. The green circle denotes the φ/ϕ angles of P182 when bound to cHSP27, measured from the crystal structure (PDB: 4mjh).

E    Contour plot showing the first and second principal components of a PCA for the MD trajectories for the WT (left) and P182L peptide (right). The larger spread in the P182L peptide suggests a larger sampling of the conformational landscape.

the IxI/V motif to less frequently adopt the conformation required to bind to the hydrophobic β4/β8 groove.

## HSP27 interactors with IxI/V motifs bind with higher affinity to the P182L variant

Our results demonstrate that the P182L mutation causes HSP27 to assemble into significantly larger oligomers by disrupting the binding of its own IxI/V motif. In addition, it is known that the P182L mutation also changes the affinities of its interactions with other proteins, such as the poly C-binding protein 1 (PCBP1) which bound with higher affinity to the P182L variant of HSP27 (Geuens *et al*, 2017). We note that PCBP1 contains multiple IxI/V motifs and therefore could compete with the IxI/V in HSP27 for binding to its β4/β8 groove (Hilton *et al*, 2013b). Since our data demonstrate that the IxI/V-binding site in P182L would be more accessible to IxI/V-containing proteins, we hypothesized that interactions between HSP27 and other IxI/V-containing proteins will be effectively strengthened.

To test this hypothesis, we first searched the human proteome for all instances of the IxI/V SLiM according to its traditional definition of [V/I]x[V/I] and then determined how many of these proteins overlapped with previously identified HSP27-binding proteins. The human proteome search yielded a total of 128,604 instances or 1.1% of all tripeptides (Appendix Table S4, Appendix Fig S12A and B). Next, we limited our search to the disordered regions of the proteome, because an IxI/V motif would need to be accessible in order to bind to the ACD, and SLiMs typically occur in unstructured regions of proteins (Davey *et al*, 2012). This reduced the number of [V/I]x[V/I] motifs by *ca.* 12-fold to 15,643 (Appendix Table S4, Appendix Fig S12A and B). Differences between the frequencies of [V/I]x[V/I] motifs in disordered and structured regions provide further insight into the amino acids that are enriched in these [V/I]x[V/I] motifs (Fig 5A, Appendix Fig S12C). The largest enrichment in disordered [V/I]x[V/I] motifs is found for x = proline (+6.7%), whereas the largest depletion is observed for x = leucine (–4.1%) (Fig 5A, Appendix Fig S12C). While proline is a disorder-promoting amino acid that is enriched in disordered regions (Theillet *et al*,

2014), we note that our bioinformatics analyses here focus specifically on the nature of the amino acid between the Val or Ile residues within a [V/I]x[V/I] motif. Statistical tests reveal that a specific enrichment of Pro within [V/I]x[V/I] motifs is not simply due to the underlying amino acid frequencies (Appendix Fig S12D). These results demonstrate that, in disordered regions, proline-containing [V/I]x[V/I] SLiMs are the most abundant type. This suggests that, in the crowded intracellular environment, HSP27 will be exposed to many [V/I]P[V/I] motifs from different proteins that can compete for binding to the ACD.

Next, we computed the intersection of known HSP27-binding proteins (Stark *et al*, 2006) and proteins that contain a [V/I]P[V/I] SLiM (Appendix Fig S12E). Of the known 449 HSP27-binding proteins, a total of 22 proteins contain [V/I]P[V/I] SLiMs in disordered regions (Appendix Fig S12E, Appendix Table S5). Some of the identified proteins contain multiple such motifs in their primary sequences (Appendix Fig S12E, Appendix Table S5), which could further enhance their binding affinity to the P182L variant of HSP27 through multi-dentate interactions. To directly test whether P182L HSP27 binds with higher affinity to IxI/V-containing proteins, we reanalyzed a comparative HSP27 interactomics dataset (Haidar *et al*, 2019), in which V5-epitope-tagged P182L was immunoprecipitated from HeLa cells and statistically enriched interactors were identified by LC-MS/MS (Fig 5B). Of the 11 proteins that were specifically enriched in the P182L dataset, eight of these proteins contained [V/I]x[V/I] motifs (Fig 5B).

To validate these interactions, we used co-immunoprecipitation assays that enable determination of the enrichment with the P182L variant versus the WT protein. We pulled down V5 epitope-tagged WT or P182L HSP27 and blotted for five different proteins (UBXN1, PCBP1, HSPB5, FAM83D, BAG3) (Fig 5C and D). For each target, we found that P182L pulled down an appreciably larger fraction than WT HSP27 (Fig 5C and D), indicating that these proteins bind to P182L with significantly higher affinity. WT HSP27 was unable to pull down UBXN1, PCBP1, or FAM83D, which suggests that these interactions are either specific to the P182L mutant or too labile with WT to be captured with this method. In an earlier study on PCBP1 (Geuens *et al*, 2017), we showed that WT HSP27 does interact with PCBP1, but found that this interaction was not straightforward to identify with co-immunoprecipitation assays. This suggests that WT HSP27 might also interact with UBXN1 and FAM83D in a transient manner, while the significantly tighter interactions with P182L could sequester these proteins away from their other binding partners.

We then sought to confirm that these tighter interactions between P182L and its IxI/V-motif-bearing interactors actually depend on the IxI/V motif. The co-chaperone BAG3 contains two IxI/V motifs in its N-terminus (Fig 5E), and previous work has indicated that BAG3 binds to sHSPs, including HSP27, via these IxI/V motifs (Carra *et al*, 2008; Fuchs *et al*, 2010; Rauch *et al*, 2017). Given the increased availability of the IxI/V-binding site in the ACD of P182L HSP27 and the enrichment in IxI/V motifs in P182L interactors noted above, we reasoned that mutating both IxI/V motifs in BAG3 would abrogate the P182L-BAG3 interaction. Indeed, we created a BAG3 mutant in which both IPV motifs were mutated to GPG (BAG3-IPV mutant) and observed that it lost its interaction with the P182L variant (Fig 5E, Appendix Fig S13A and B). These findings suggest that IxI/V-containing proteins, at least in part, use these motifs to bind more tightly to the P182L mutant.

Finally, we asked whether the more accessible IxI/V-binding site in P182L is responsible for the higher affinity interactions with these targets. We hypothesized that mutating the β4/β8 groove of P182L to prevent IxI/V binding would abolish interactions with IxI/V-containing proteins. To this end, we introduced the S155Q mutation that prevents IxI/V binding (Delbecq *et al*, 2012; Baughman *et al*, 2020) into the P182L variant (S155Q/P182L). We pulled down V5 epitope-tagged S155Q/P182L and blotted for the same interactors as above. We found that the double mutant S155Q/P182L was either completely unable to pull down the interactors (UBXN1, PCBP1, or FAM83D) or pulled down a much lower amount relative to P182L (BAG3 and HSPB5) (Fig 5F), suggesting that the β4/β8 groove comprises the primary interaction site for these targets. Collectively, our results demonstrate that IxI/V-containing proteins bind with higher affinity to the P182L variant of HSP27 via the more available β4/β8 groove.

## Discussion

We have investigated the biophysical and functional impact of the CMT disease-causing P182L mutation in the highly conserved IxI/V motif of the intrinsically disordered CTR of HSP27. We found that the clinically severe P182L variant of HSP27 displays reduced chaperone activity and exists in a significantly larger oligomeric form both *in vitro* and *in vivo*, including within CMT patient-derived iPSC motor neurons. To our knowledge, the P182L variant of HSP27 assembles into the largest soluble sHSP oligomers observed thus far, surpassing *E. coli* IbpB (2–3 MDa) (Strózecka *et al*, 2012). In addition, the P182L variant forms significantly larger oligomers than any of the CMT-related mutants of HSP27 studied to date across the entire polypeptide sequence (Chalova *et al*, 2014; Muranova *et al*, 2015; Weeks *et al*, 2018). Our *in vitro* experiments made use of purified proteins, indicating that the large P182L oligomers can be comprised solely of the mutant protein; however, while we observe qualitative agreement with the *in vitro* data, our *in cellulo* data cannot rule out that other components, in addition to WT HSP27 (Appendix Fig S3), might be present within the large P182L oligomers.

Because the structural basis for the dysregulation of HSP27 by the P182L mutation has remained unknown, we sought to provide atomic-level details regarding the impact of this single point mutation. To this end, we used NMR spectroscopy to compare the binding of the WT and P182L IxI/V motifs to the ACD of HSP27. Importantly, recent NMR data have demonstrated that the isolated ACD adopts a similar conformation to that observed in the full-length protein (Clouser *et al*, 2019), suggesting that the ACD provides a convenient system to investigate IxI/V binding. Our NMR data on the ACD (Figs 3 and 4) revealed a lowered association rate constant and hence affinity of P182L IxI/V binding to the ACD. A reasonable explanation for this decreased association rate constant is provided by our MD simulations (Fig 4), which show that Pro182 restricts the conformational landscape of the IxI/V motif, thereby placing it in a binding-competent conformation more often than the P182L variant. The larger sampling of conformational space by the P182L variant and larger loss of conformational entropy upon binding could thus contribute to its slower association rate constant (Pritišanac *et al*, 2019). However, we note that other

factors also regulate biomolecular interactions, including solvation, electrostatics, and hydrogen bonding propensity (Schreiber *et al,* 2009), and the rigidity imparted by Pro182 may be just one contributing factor. For example, it was shown that introducing polar interactions downstream of the IxI/V motif (F185H) can increase the binding affinity (Freilich *et al,* 2018), perhaps by establishing a new hydrogen bond between the H185 side chain and the P150 carbonyl (Appendix Fig S14). Thus, amino acids outside of the IxI/V motif can also contribute to this interaction. In addition, van der Waals contacts between the side chains of [181]Ile and [183]Val in

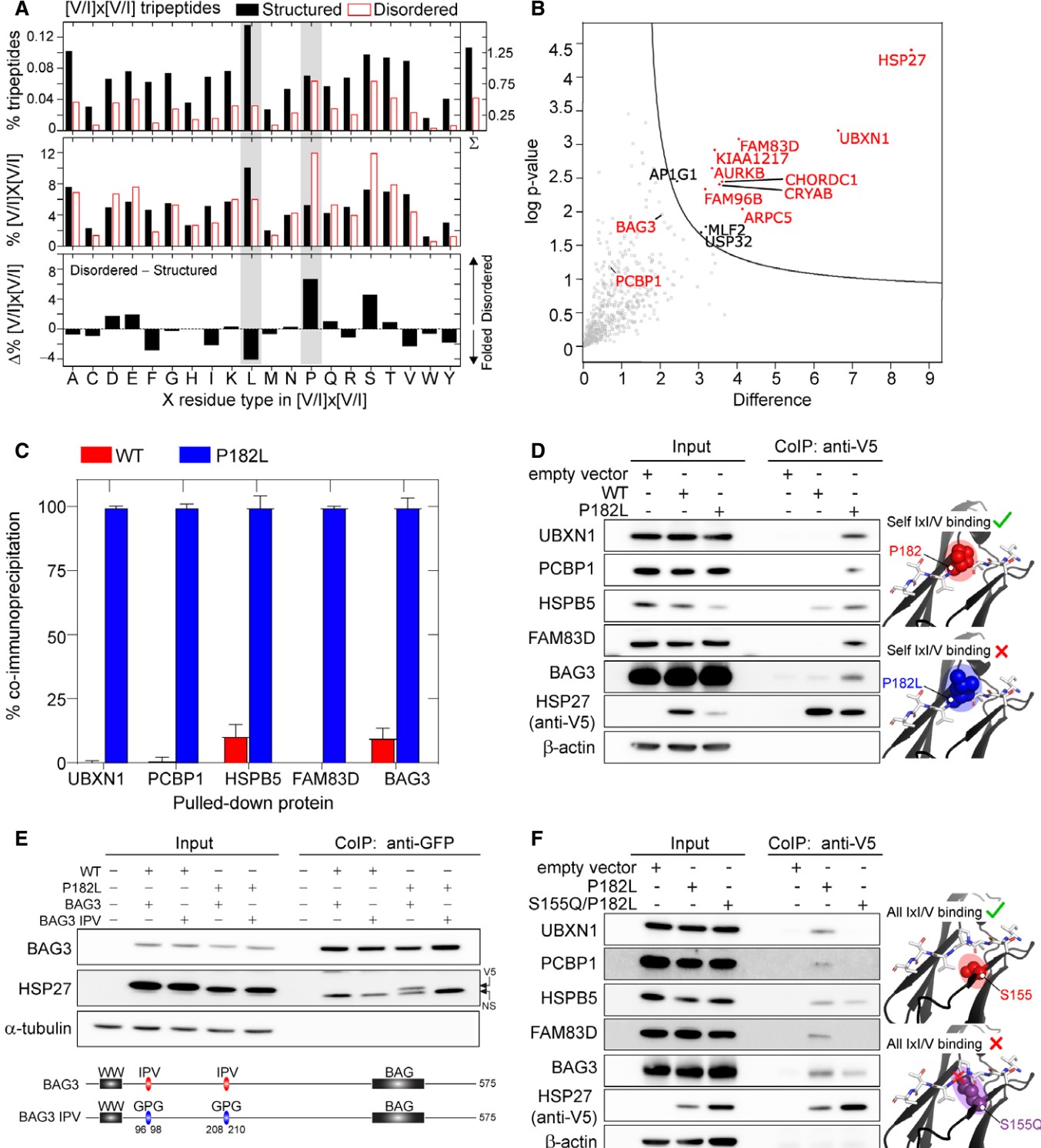

**Figure 5.**

**Figure 5.  The P182L variant binds with enhanced affinity to IxI/V-containing proteins.**

A    Percentage of [V/I]x[V/I] tripeptides among (*top*) the total number of all tripeptides (% tripeptides) or (*middle*) the total number of [V/I]x[V/I] tripeptides in the structured (black) or disordered regions (red) ([V/I]x[V/I] %) shown as a function of the X residue type. (*Bottom*) The difference between [V/I]x[V/I] % for disordered and structured regions shows the enrichment (positive) or depletion (negative) of specific [V/I]x[V/I] motifs in disordered regions. The gray bars indicate the [V/I]L[V/I] and [V/I]P[V/I] motifs studied in this work, *i.e.*, ILV and IPV.

B    Volcano plot representing the interactors that are enriched for the P182L mutant versus GFP as a negative control, obtained with affinity-enrichment mass spectrometry. Proteins that co-immunoprecipitated significantly more ($P < 0.05$) with the P182L variant are shown in the upper right quadrant. Interactors with one or more [I/V]x[I/V] motifs are displayed in red while other significantly enriched interactors are displayed in black.

C, D  Co-immunoprecipitation from HeLa cells stably overexpressing V5-epitope-tagged HSP27 (WT or P182L mutant) using anti-V5 beads. Co-immunoprecipitation was quantified and presented as a percentage relative to the P182L variant ($n = 3$). The mean $\pm$ one standard deviation ($n = 3$) is shown. *Right*: the location of Pro182 (red) in the crystal structure of peptide-bound cHSP27 (PDB: 4mjh) and *in silico* creation of the P182L mutation (blue). The peptide is shown in white sticks. P182L specifically weakens binding of the HSP27 IxI/V motif.

E    Co-immunoprecipitation from HeLa cells stably overexpressing V5-epitope-tagged HSP27 (WT or P182L mutant) and transiently transfected with BAG3-eGFP (WT or IPV-mutant) using anti-GFP beads. Western blots for anti-GFP (BAG3), V5 (HSPB1), and α-tubulin. NS indicates a non-specific band. Note that detection of BAG3 (anti-GFP) and HSP27 (anti-V5) occurs with different antibodies. The locations of the IPV-to-GPG mutations in BAG3 are depicted below.

F    Co-immunoprecipitation from HeLa cells transiently transfected with V5-epitope-tagged HSP27 (P182L or S155Q/P182L mutant) using anti-V5 beads ($n = 2$). *Right*: the location of Ser155 (red) in the crystal structure of peptide-bound cHSP27 (PDB: 4mjh) and *in silico* creation of the S155Q mutation (purple) showing a clash with the bound peptide. The peptide is shown in white sticks. S155Q disrupts binding of all IxI/V motifs to HSP27.

Source data are available online for this figure.

the IxI/V motif and the hydrophobic β4/β8 groove are critical for binding as substitution of the Ile and Val side chains with Gly completely disrupted the binding of an IxI/V-bearing peptide (Rauch *et al*, 2017).

The complex binding mechanism of the IxI/V motif is underscored by our observation that there are at least two interconverting bound states (Appendix Figs S6A and B, S7A and S11). The interconversion between multiple bound forms has previously been identified in other protein–protein interaction mechanisms, including those that involve the association of an intrinsically disordered region and a structured domain (Sugase *et al*, 2007a; Schneider *et al*, 2015; Delaforge *et al*, 2018), including sHSPs (Delbecq *et al*, 2012; Clouser *et al*, 2019). In our case, the intrinsically disordered CTR of HSP27 docks into the β4/β8 groove of the structured ACD. Our identification of more than one conformation of the ACD-bound IxI/V suggests that this region binds in a dynamic manner.

We note that our observation of more than one bound state is consistent with the binding mechanism of the IxI/V motif in the related sHSP αB-crystallin, for which it was shown that the peptide-bound form exists in up to three different conformations (Delbecq *et al*, 2012). We have established previously that the affinity for binding the second peptide to the dimer is lower than the affinity for the first (Hilton *et al*, 2013a) which acts to control assembly and disassembly in the context of the full-length αB-crystallin oligomers (Baldwin *et al*, 2011, 2012; Hochberg & Benesch, 2014). Thus, the IxI/V motif exhibits considerable motions in its free state, where it is intrinsically disordered, as well as in its bound state via the accession of multiple interconverting conformations that quite possibly regulate assembly in HSP27, and sHSPs more generally.

The P182L mutation lowers the binding affinity of the IxI/V motif for the β4/β8 groove and dysregulates oligomeric assembly. If the P182L variant decreases the overall binding affinity for the IxI/V motif, how does the full-length protein form larger oligomers than the WT protein? Studies on the related α-crystallin sHSPs provide a rationale. In αB-crystallin, the IxI/V motif, despite its proximity to the β4/β8 groove (Jehle *et al*, 2010; Baldwin *et al*, 2011), is nevertheless predominantly unbound in WT oligomers with ca. 1% of the CTDs being bound under physiological solution conditions and temperature (Baldwin *et al*, 2011, 2012). In αA-crystallin, a recent

cryoEM study found few inter-ACD contacts; 70–80% of the subunits bound the IxI/V intramolecularly (Kaiser *et al*, 2019), consistent with an earlier crystal structure (Laganowsky & Eisenberg, 2010). In solution, the concomitant binding of two IxI/V motifs precipitates ejection of a monomeric subunit from the oligomer from αB-crystallin (Baldwin *et al*, 2011, 2012). If a similar process governs the oligomerization of HSP27, we would expect a slower on rate of the IxI/V to lead to slower ejection of monomers from the oligomers, and so, the oligomeric size would increase (Baldwin *et al*, 2011, 2012), in line with our observations of P182L.

In addition to the CTD, the NTD is also known to play an important role in facilitating oligomerization in a range of sHSPs (Kaiser *et al*, 2019). For HSP27 specifically, WT-like oligomerization has been observed for CTR-truncated forms of HSP27 but not NTD-truncated forms (Lelj-Garolla & Mauk, 2012). Recent studies of HSP27 oligomeric assembly identified interactions between specific regions in the largely disordered NTD (e.g., [6]VPF[8,] and Val85) and the IxI/V-binding site in the β4/β8 groove (Clouser *et al*, 2019; Collier *et al*, 2019), indicating that multiple regions of HSP27 compete for binding to the β4/β8 groove. Attenuating the binding affinity of the IxI/V motif in the CTR could enable other sites in the NTD to bind more frequently and hence significantly alter the oligomeric assembly landscape of HSP27 (Clouser *et al*, 2019; Collier *et al*, 2019).

Taken together, these results and our present work indicate a role for the CTR in regulating rather than underpinning oligomerization. Because the IxI/V motif can bind to the β4/β8 in an intra- or intermolecular manner (Kaiser *et al*, of the P182S variant, namely its assembly into2019) or are transiently bound (Baldwin *et al*, 2011), it may serve as the "gatekeeper" that regulates what binds to the β4/β8 groove (Hochberg & Benesch, 2014). Prior studies independently support such a role, as mutants in HSP27 that disrupt binding of its IxI/V motif have been shown to increase rather than decrease the molecular mass, including the CMT-causing mutation P182S (Chalova *et al*, 2014), the [181]IPV[183] to [181]GPG[183] variant (Freilich *et al*, 2018), and other IxI/V mutations (Pasta *et al*, 2004; Hilton *et al*, 2013a; Freilich *et al*, 2018). The behavior of the P182S variant, specifically its assembly into a significantly larger oligomeric form than the WT protein, suggests that the P182S mutation acts similarly

as P182L by lowering the affinity for the β4/β8 groove (Chalova *et al*, 2014). In addition, the physiological O-GlcNAcylation of T184, which is immediately upstream of the [181]IPV[183] motif, disrupts binding of the IxI/V motif and increases the average HSP27 oligomer size to > 1 MDa (Balana *et al*, 2019), indicating regulation of IxI/V binding through post-translational modification. Given the universal nature of the IxI/V motif in sHSPs, we speculate that evolution has finely tuned its affinity for the ACD to optimally balance the binding of self versus others, thereby influencing oligomerization and chaperone activity. It will be of interest to assess how mutations that increase the affinity of the IxI/V motif will affect the properties of HSP27; for example, the F185H mutation significantly lowered the $K_d$ of peptide binding (Freilich *et al*, 2018).

Dysregulated binding of the NTD to the β4/β8 groove could have implications for the chaperone mechanism of HSP27, as recent NMR data have indicated that aggregation-prone substrates can either bind to the NTD, the ACD, or the β4/β8 groove (Mainz *et al*, 2015; Freilich *et al*, 2018; Baughman *et al*, 2020; Liu *et al*, 2018, 2020), and HSP27-substrate interactions involving the β4/β8 groove may not always prevent substrate aggregation (Baughman *et al*, 2020). For such substrates that interact with the β4/β8 groove, the binding affinities appear to be very weak (Baughman *et al*, 2018; Freilich *et al*, 2018; Liu *et al*, 2018, 2020), and these interactions would exist in competition with binding of the IxI/V motif. Therefore, increased availability of the β4/β8 groove in P182L IxI/V would leave its β4/β8 groove more accessible to interactions with other inter- and intramolecular contacts and could reduce the number of NTD-substrate interactions and decrease the overall chaperone activity of the HSP27 chaperone. At least for the substrate tau, it was shown that the NTD is required to prevent substrate aggregation, whereas binding of tau to the β4/β8 groove does not prevent substrate aggregation (Baughman *et al*, 2020).

As another consequence, an interacting partner may become dysregulated due to aberrant interactions with P182L that disrupt other interaction networks. A previous study found that competing interactions between HSP27 self-binding and substrate binding regulated HSP27 function and oligomerization (Freilich *et al*, 2018). For five different IxI/V-containing proteins (UBXN1, PCBP1, HSPB5, FAM83D, and BAG3), our co-immunoprecipitation data demonstrate that they all bind more tightly to the P182L variant than WT HSP27. We also showed that disruption of the β4/β8 groove in P182L via the S155Q mutation prevents these interactions (Fig 5) and that, for BAG3, the interaction with P182L depends on its own IPV motifs (Fig 5). Notably, WT HSP27 was unable to pull down three of these interactors (UBXN1, PCBP1, and FAM83D), which suggests that they may only transiently associate with the WT protein via the β4/β8 groove. By increasing the affinity of these interactions via the P182L mutation, however, we were able to clearly identify these targets as interactors. Bioinformatic identification of proteins that harbor [V/I]P[V/I] SLiMs may help instead to identify targets that are not known HSP27 interactors but could potentially bind to HSP27 via weak, transient interactions with the β4/β8 groove.

The disrupted interaction network caused by the P182L mutation most strongly impacts WT HSP27 itself. Mutations in HSP27 that cause CMT disease and dHMN, including P182L and P182S, are generally autosomal dominant, meaning that one allele will contain the mutated form of the gene and the other allele will contain the

WT gene. Assuming relatively equal levels of expression *in vivo*, a mixed population of WT and P182L HSP27 will be present inside the cell and hetero-oligomers of varying ratios of WT:P182L will presumably exist, as has been established for other disease-causing variants of HSP27 *in vitro* (Nefedova *et al*, 2013). Indeed, we demonstrated here that P182L and WT HSP27 hetero-oligomerize *in vitro* and inside HeLa cells (Appendix Fig S3). Therefore, the available cellular pool of WT HSP27 will not only be depleted by having just one allele from which it is expressed but also by the local sequestration of the WT protein by P182L oligomers.

Other interaction partners are also affected by the dysregulated interactions of P182L. For instance, we showed previously that sequestration by the P182L variant causes a loss of translational repression by the RNA-binding protein PCBP1 (Geuens *et al*, 2017). The BAG3/HSP27 co-chaperone/chaperone interaction, on the other hand, promotes new protein–protein interactions, in part by bringing together HSP27 with other BAG3-bound chaperones, including HSP70 (Rauch *et al*, 2017; Meister-Broekema *et al*, 2018). Moreover, BAG3 is recruited to stress granules via its interactions with HSPB8 where the BAG3-HSPB8-HSP70 complex plays a key role in ensuring stress granule functionality (Ganassi *et al*, 2016). Likewise, it has been shown that HSP27 is also recruited to stress granules (Ganassi *et al*, 2016; Jain *et al*, 2016) and that it plays a direct role in chaperoning proteins inside stress granules (Liu *et al*, 2020). Therefore, disturbing the dynamic balance of the HSP27-BAG3 interaction might therefore affect a wider chaperone network and contribute to destabilization of proteostasis (Meriin *et al*, 2018). Notably, mutations in the BAG3 IPV motifs also contribute to human disease: a Pro-to-Leu mutation is implicated in myofibrillar myopathy (Selcen *et al*, 2008; Freilich *et al*, 2018; Meister-Broekema *et al*, 2018), and another Pro-to-Ser mutation is implicated in CMT disease (Shy *et al*, 2018; Adriaenssens *et al*, 2020). The P209L mutation in BAG3, which replaces one of the two IPV motifs with [208]ILV[210], lowered its binding affinity for HSP27 to the same degree as deleting that IPV motif (Meister-Broekema *et al*, 2018). The presence of disease-causing mutations in the IxI/V SLiMs of HSP27 and BAG3 suggests important roles for these motifs.

In summary, our results provide an unexpected mechanistic explanation for the consequences of a mutation in the conserved IxI/V motif, which causes a severe form of CMT. Namely, the P182L mutation makes the IxI/V-binding site of HSP27 more accessible for interactions with other IxI/V-bearing proteins, leading to higher affinity interactions with such proteins. Given the high level of conservation in the IxI/V motif across various mammalian sHSPs, we anticipate that our results will relate to other sHSP systems and implicate this motif as a key mediator of sHSP-target interactions.

## Materials and Methods

Peptides were synthesized by Biomatik with N-terminal acetylation and C-terminal amidation. The sequences of the WT and P182L peptides are [178]EITIPVFE[185] and [178]EITILVFE[185]. The peptides were dissolved in NMR buffer (30 mM sodium phosphate, 2 mM EDTA, pH 7), but the pH values of the solutions were found to be in the 4–5 range. Because cHSP27 is highly pH-sensitive (Clouser & Klevit, 2017; Alderson *et al*, 2019, 2020), the pH of all peptide solutions was corrected to 7 using small volumes of concentrated NaOH.

Protein samples for NMR, chaperone activity assays, CD, negative-stain EM, and SEC-MALS were expressed in *E. coli* and purified as described previously (Alderson *et al*, 2019). WT HSP27 comprises residues 1–205, encompassing the entire amino acid sequence. The P182L mutation was introduced into the WT HSP27 expression plasmid via site-directed mutagenesis. The full-length P182L variant of HSP27 was expressed as the WT form but went into inclusion bodies, which were solubilized in 8 M urea, spun at 20,000 ×g for 20 min, and then dialyzed into 30 mM sodium phosphate, 100 mM NaCl, 2 mM EDTA, and pH 7 buffer. The supernatant was filtered with a 0.22-μm filter and then loaded onto a Superdex 200 gel filtration column equilibrated in 30 mM sodium phosphate buffer, 100 mM NaCl, 2 mM EDTA at pH 7.

cHSP27 encompasses residues G84–K171 of HSP27, and the expression plasmid contains an N-terminal hexahistidine tag followed by a tobacco etch virus (TEV) protease cleavage site. The Gly overhang that remains after TEV protease cleavage corresponds to G84 in the HSP27 amino acid sequence. cHSP27 was expressed and purified as described previously (Alderson *et al*, 2019). Following exchange into a buffer without reducing agent, cHSP27 forms an intermolecular disulfide bond involving C137 from adjacent subunits. The NMR buffer used was 30 mM sodium phosphate, 2 mM EDTA at pH 7 with 6% $D_2O$ added for the lock. For NMR titration and relaxation dispersion studies, this disulfide bond was left intact to minimize contributions from exchange between the monomer and dimer. The formation of the disulfide bond is readily identifiable in the 2D $^1H$-$^{15}N$ HSQC spectra of $^{15}N$-cHSP27 via the resonances from R136 and C137; upon reduction or C137S mutation, these resonances become very weak or broaden into the noise (Rajagopal *et al*, 2015; Alderson *et al*, 2019). In addition, the oxidized dimer simplifies the analysis of titration and CPMG RD data, as only a single state of the protein is present, rather than a mixture of two forms (non-covalent dimer and monomer).

## Chaperone activity assays

Chaperone activity assays were completed in triplicate, with the mean and one standard deviation reported. Assays were performed using a 96-well plate and a FLUOstar Omega Microplate Reader or Tecan Infinite M200 PRO plate reader. The substrates were porcine heart MDH (Sigma-Aldrich) and human insulin (Sigma-Aldrich). MDH was prepared at 0.2 μM and was incubated at 40°C in 30 mM sodium phosphate, 100 mM NaCl, 2 mM EDTA at pH 7. The increase in absorbance at 340 nm was monitored over time in the absence and presence of 0.5 μM WT or P182L HSP27. The MDH aggregation assay proceeded for 3 h. Likewise, the aggregation of 40 μM insulin at 40°C in the same buffer as above was monitored by the increase in absorbance at 340 nm upon the addition of 1 mM DTT. Insulin aggregation reactions were monitored in the absence or presence of 20 μM WT or P182L HSP27, and the reactions proceeded for 0.5 h. Control experiments were performed using 20 μM WT or P182L HSP27 incubated alone in the same buffer as above at 40°C for 3 h. Chaperone activity (Appendix Fig S1) is defined as 1 – Θ, where Θ is the maximum absorbance for the chaperone plus substrate mixture divided by the maximum absorbance of the substrate alone. Values of 1 and 0, respectively, indicate maximum protection and the absence of protection against aggregation.

## SEC-MALS

Molecular masses were estimated by analytical SEC with in-line MALS (DAWN Heleos-II, Wyatt Technology Inc., Santa Barbara, CA), refractive index (Optilab T-rEX, Wyatt Technology, Inc.), and UV detectors (Waters 2487, Waters Corp., Milford, MA), along with ASTRA version 6 software that was provided with the instrument. To characterize the isolated proteins, 250 μg of each protein was injected to a pre-equilibrated Superdex-200 column (for P182L and HSP27) or Superdex-75 column (for cHSP27 with the C137S mutation) in 30 mM sodium phosphate, 100 mM NaCl, 2 mM EDTA buffer at pH 7 and eluted at a flow rate of 0.5 ml/min. The monomer concentration of WT and P182L HSP27 corresponded to 40 μM. The calculated molar masses of HSP27, P182L, and cHSP27 (C137S) were 470 kDa, 13,600 kDa, and 17.5 kDa, respectively, within 1–1.2% error. The theoretical mass of the cHSP27(C137S) dimer is 19.8 kDa, so the slightly lower value as determined by SEC-MALS likely reflects contributions from exchange with the free monomer. Note that the C137S mutation does not impact the overall structure of the cHSP27 dimer (Alderson *et al*, 2019). The molecular mass of WT HSP27 is in good agreement with previous studies (Jovcevski *et al*, 2015; Mishra *et al*, 2018), with masses of 670 kDa obtained by analytical SEC and 400 kDa by SEC-MALS. The hydration radii of WT and P182L HSP27, as determined by SEC-MALS, are 9 and 38 nm, respectively.

For the hetero-oligomerization experiment, 250 μg of WT HSP27 was loaded onto a Superose 6 10/300 column in 20 mM sodium phosphate, 100 mM NaCl, 2 mM EDTA at pH 7. A second sample was then prepared in which 230 μg of WT HSP27 was mixed with 230 μg of the P182L variant, and this sample was equilibrated overnight at room temperature before injection onto the column.

## Negative-stain EM

Samples for negative-stain EM were prepared at 4.5 μM monomer concentration in 30 mM sodium phosphate, 100 mM NaCl, 2 mM EDTA at pH 7. Samples were loaded onto carbon-coated copper EM grids (Ultrathin Carbon Film/Holey Carbon; Ted Pella) and incubated there for 60 s. Excess sample was blotted off the grid, and then, the grids were washed with deionized water and stained with 0.5% uranyl formate for 20 s. Excess stain was removed by blotting. Negative-stain EM images were then collected on a FEI Tecnai T12 electron microscope operating at 120 kV, which was equipped with a Gatan US1000 CCD camera.

## Cell lines, cell culture, and transfection

HeLa cells were cultured in MEM medium (Thermo Fisher Scientific) supplemented with 10% fetal bovine serum (FBS; Thermo Fisher Scientific), 1% glutamine (Thermo Fisher Scientific), and 1% penicillin/streptomycin. Cells were maintained at 37°C and 5% $CO_2$ atmosphere. To create HSP27 CRISPR knockout lines, we generated a sgRNA containing pSpCas9(BB)-2A-PURO (PX459) V2.0 vector (Addgene, #62988). The most specific sgRNA was selected with *crispr.mit.edu* by targeting the first exon of the *HSPB1* gene. Through molecular cloning following Ran *et al* (2013), the sgRNA was cloned into the *Bbs*I restriction site and successful insertion was verified by Sanger sequencing. HeLa cells were seeded in a 6-well plate and

transiently transfected using polyethylenimine (PEI). After 24 h, the medium was supplemented by 1 μg/ml puromycin to select for positively transfected cells. The puromycin was removed after 72 h as all the cells were dead in a non-transfected well. The positively transfected cells were then serially diluted in a 96-well plate with an estimated 0.5 cells/well. Wells that contained clonal expansion for more than one colony were discarded. Other wells starting from a single colony were expanded and collected to verify the genome editing both at the protein and DNA level. Knockout clones were identified by Western blotting using a monoclonal HSP27 antibody (Enzo Life Sciences SPA-800). The successful clones were screened further by DNA sequencing of the edited genomic region, after subcloning the amplicons into pUC19 plasmids. This allowed to identify the genome edits for each of the respective alleles. Only clones with an identified stop codon on each allele were selected for further experiments.

For microscopy experiments, HSP27 knockout HeLa cells were grown on standard cover glasses (12 mm #1.5) at a concentration of 50.000 cells/well. The cells were transiently transfected with HSP27 wild-type or P182L mutant ORF in pLenti6/V5 plasmids (Thermo Fisher Scientific).

**Expansion microscopy sample preparation and imaging**

Cells were immunostained and processed for expansion microscopy employing a protocol by Chozinski *et al* (Chozinski *et al*, 2016). Twenty-four hours after transfection, cells were fixed for 20 min in 3.2% paraformaldehyde and 0.1% glutaraldehyde in PBS with 5-min reduction in borohydride. After three PBS washes, samples were blocked and permeabilized for 1 h at room temperature in normal goat serum (1:500, Dako) and 0.5% Triton X, dissolved in PBT buffer (PBS + 0.5% BSA + 0.02% Triton X). Primary antibody incubation over night at 4°C with 1:500 anti-HSP27 (Enzo Life Sciences SPA-800) in PBT was followed by four 5-min and one 30-min PBT washes, 1-h secondary antibody incubation at room temperature (1:500 in PBT, AlexaFluor488 goat anti-mouse, Thermo Fisher Scientific A11001), four 5-min PBT washes and one 30-min PBS wash. Control coverslips for conventional confocal microscopy were nuclear counter stained with Hoechst 33342 (1/20,000 for 10 min at room temperature) and mounted on microscopy slides in antifade mounting medium (Dako). Samples for expansion microscopy were cross-linked for 10 min in 0.25% glutaraldehyde in PBS. Gelation was done in a mixture of 2 M NaCl, 2.5% (w/w) acrylamide, 0.15% (w/w) N, N′-methylenebisacrylamide, 8.625% (w/w) sodium acrylate in PBS with polymerization initiated with TEMED and APS. Polymerized gels were incubated for 30 min at 37°C in a digestion buffer containing 8 U/ml proteinase K. Cover glasses were removed from the digested gels, which were placed in high volumes (> 30 ml) of distilled water that were exchanged at least five times until full expansion of the gels. Finally, the gels that had expanded 4.3 times in all dimensions were trimmed, nuclear counter stained with Hoechst 33342 (1/5,000 in water for 30 min at room temperature), positioned in 50 mm diameter glass bottom dishes (WillCo Wells GWSt-5040), and immobilized using 2% agarose.

Image stacks of the expanded sample were acquired on a Zeiss LSM700 with Plan-Apochromat 63×/1.40 objective, using 85 × 85× 445 nm voxel dimensions (corresponding to approximately 20 × 20 × 100 nm in a non-expanded sample). We measured minimal spot diameters (FWHM in expansion corrected scales) in

the range of 60–80 nm (data not shown), confirming that the physical size of the particles is still below this limit and cannot be directly quantified.

**Image processing and analysis**

Cytoplasmic regions of interest were randomly selected from image stacks (200 × 200 pixels, single slice), and the HSP27 distribution was analyzed using the Fiji distribution of ImageJ (Schindelin *et al*, 2012; Schneider *et al*, 2012). After noise filtering (Gaussian blur, sigma 0.7), high-intensity spots were detected with the Find Maxima tool (fixed noise level 50). The intensity and size of the spots were measured with the GaussFit_OnSpot plugin (Peter Haub and Tobias Meckel) using circle shape and Levenberg Marquard fit mode and 10-pixel (850 nm) rectangle half size. All images were processed in batch using ImageJ macro scripts. Data analysis and plotting was done using R (R: A language and environment for statistical computing. R Foundation for Statistical Computing, 2015). Misfitted spots were removed by excluding spots smaller than 20 nm and larger than 200 nm (FWHM in expansion microscopy corrected dimensions). For each genotype, 15 cytoplasmic regions of the same size as the expansion microscopy images in Fig 2C were analyzed (4 × 4 μm), which resulted in the detection of 3342 spots in total. Mean intensities of the entire images were measured using ImageJ, after background correction by subtraction of a Gaussian blurred image (500-pixel radius).

**Sucrose gradient assay**

HeLa cells were lentivirally transduced with HSPB1-V5 (wild type or P182L mutant) and collected for protein extraction with RIPA buffer [1% Nonidet P-40, 150 mM NaCl, 0.1% SDS, 0.5% deoxycholic acid, 1 mM EDTA, 50 mM Tris–HCl pH 7.5, cOmplete Protease Inhibitor Cocktail (Roche Applied Science, Indianapolis, IN, USA), Phospho-STOP inhibitor mix (05 892 970 001, Roche Applied Science, Indianapolis, IN, USA)] for 30 min on ice. Protein concentrations were quantified using BCA (23225, Pierce BCA Protein Assay Kit). The sucrose gradient was generated by diluting an 80% sucrose stock solution in PBS to 70, 60, 50, 40, 30, 20 and 10%. Equal amounts from each fraction were layered on top of each other and allowed to equilibrate on ice for 30 min. Equal amounts of protein lysate were then loaded on top of the gradient. Note that this is total whole cell protein lysate and thus contains both soluble and non-soluble proteins. Samples were centrifuged for 60 min at 55,000 rpm (164,025 *g*) in a TLA-110 rotor (Beckman). Nine samples were collected and prepared for SDS–PAGE analysis. After separation on 4–12% NuPAGE gels (Life Technologies, Carlsbad, CA, USA). Proteins were transferred to nitrocellulose membranes (Hybond-P; GE Healthcare, Wauwatosa, WI, USA) and decorated with antibodies against V5 (R96025, Invitrogen, Carlsbad, CA, USA). Samples were detected using enhanced chemiluminescent ECL Plus (Pierce, Life Technologies, Carlsbad, CA, USA) and LAS4000 (GE Healthcare, Wauwatosa, WI, USA).

We note that, for both WT and P182L HSP27, two populations of oligomers are observed in the sucrose gradient assay, whereas only a single elution peak is obtained in the SEC-MALS experiments. Observation of a single elution peak by SEC-MALS, however, does not mean that only a single population of oligomers is present:

HSP27 and other sHSPs form a polydisperse ensemble of oligomers (Hochberg & Benesch, 2014), and gel filtration elution profiles reflect the ensemble-averaged nature of the sample. Moreover, our observation of two populations of oligomers by the sucrose gradient assay is consistent with analytical SEC performed as a function of HSP27 concentration (Jovcevski *et al*, 2015). At high protein concentrations, only a single elution peak was observed, corresponding to large oligomers; decreasing the total protein concentration caused the dissociation of large oligomers into smaller species, accompanied by two elution peaks (Jovcevski *et al*, 2015). Our SEC-MALS experiments were performed at an HSP27 concentration of 40 μM (monomer), which promotes formation of large oligomers and a single elution peak via gel filtration. By contrast, our sucrose gradient assays contained a significantly lower HSP27 concentration, thereby promoting the formation of small and large oligomers. Finally, the sucrose gradient assay detected HSP27 from mammalian cells that may phosphorylate HSP27 and cause elevated dissociation of oligomers. These orthogonal assays highlight the complex nature of HSP27's oligomeric assembly, and no one technique can capture all aspects of this dynamic process on its own.

### Co-immunoprecipitation

Stable HeLa cell lines for HSPB1-V5 (wild type or P182L mutant) were either untransfected and collected or transiently transfected with wild-type or IPV-mutant BAG3-GFP constructs using linear polyethylenimine (PEI) (PolySciences Europe, Hirschberg an der Bergstrasse, Germany) twenty-four hours before collection. The IPV-mutant of BAG3-GFP was generated by site-directed mutagenesis by replacing both the first and second IPV motifs of BAG3 with GPG (the respective amino acids are 96–98 and 208–210). After collection, cells were lysed in lysis buffer [20 mM Tris–HCl pH 7.4, 2.5 mM MgCl$_2$, 100 mM KCl, 0.5% Nonidet P-40, Complete Protease inhibitor (Roche Applied Science, Indianapolis, IN, USA)] and incubated on ice for 30 min. Protein lysates were cleared by centrifugation for 10 min at 20,000 *g*, and equal amounts of supernatant (NP40-soluble fraction only) were loaded on GFP-Trap beads (gta-100, Chromotek, Martinsried, Germany) or anti-V5 agarose beads (A7345, Sigma-Aldrich, Saint Louis, MO, USA). Beads were incubated with the protein lysate for 1 h at 4°C and washed three times with wash buffer [20 mM Tris–HCl pH 7.4, 2.5 mM MgCl$_2$, 100 mM KCl, Complete Protease inhibitor (Roche Applied Science, Indianapolis, IN, USA)]. Beads were supplemented with NuPAGE LDS sample buffer (Life Technologies, Carlsbad, CA, USA) and loaded on 4–12% NuPAGE gels (Life Technologies, Carlsbad, CA, USA). Proteins were transferred to nitrocellulose membranes (Hybond-P; GE Healthcare, Wauwatosa, WI, USA) and decorated with antibodies against GFP (ab290, Abcam, Cambridge, UK), V5 (R96025, Invitrogen, Carlsbad, CA, USA), UBXN1 (16135-1-AP, Proteintech, Manchester, UK), PCBP1 (1G2, Novus Biologicals, Centennial, CO, USA), CRYAB/HSPB5 (MAB4849, R&D Systems, Minneapolis MN, USA), FAM83D (PA5-99011, Invitrogen, Carlsbad, CA, USA), BAG3 (10599-1-AP, Proteintech, Manchester, UK), GAPDH (GTX100118, GeneTex, Irvine, CA, USA), β-actin (A5441, Sigma-Aldrich, Saint Louis, MO, USA), or Tubulin (ab7291, Abcam, Cambridge, UK). Samples were detected using enhanced chemiluminescent ECL Plus (Pierce, Life Technologies, Carlsbad, CA, USA) and LAS4000 (GE Healthcare, Wauwatosa, WI, USA).

### Detection of HSP27 monomer and dimers by Western blot

HeLa cells were seeded in a 6-well plate and transiently transfected using polyethylenimine (PEI). After 24 h, the cells were collected for protein extraction by centrifugation and lysed in RIPA buffer [1% Nonidet P-40, 150 mM NaCl, 0.1% SDS, 0.5% deoxycholic acid, 1 mM EDTA, 50 mM Tris–HCl pH 7.5, cOmplete Protease Inhibitor Cocktail (Roche Applied Science, Indianapolis, IN, USA), Phospho-STOP inhibitor mix (05 892 970 001, Roche Applied Science, Indianapolis, IN, USA)] for 30 min on ice. Protein concentrations were quantified using BCA (23225, Pierce BCA Protein Assay Kit). For detection of dimers, protein lysates were boiled for 5 min at 95°C in non-reducing NuPAGE LDS sample buffer (Life Technologies, Carlsbad, CA, USA). To dissociate the dimers into monomers, the sample buffer was complemented by 100 mM DTT prior to boiling. Both samples were loaded on the same SDS–PAGE gel and analyzed by Western blot as described above. HSP27 contains only a single cysteine residue (C137) and can form a disulfide bond across the ACD dimer interface.

### Interactomics for the P182L variant

We reanalyzed a dataset that was previously published (Haidar *et al*, 2019). Label-free MS/MS was performed on co-immunoprecipitation samples obtained after mixing anti-V5 beads and protein lysates from HeLa cells overexpressing HSP27-P182L-V5 or GFP-V5. The data were analyzed as previously described, and the volcano plot in Fig 5 displays the P182L versus the GFP negative control datasets after a two-way ANOVA test to calculate the –log *P*-values. From this volcano plot, we verified which of the significantly enriched interactors contained an [I/V]x[I/V] motif.

### Generation of iPSCs from fibroblasts

Primary human fibroblasts were obtained from skin biopsies of a CMT patient and a healthy control individual after informed consent and with the approval of the medical ethical committee of the University of Antwerp. The samples were obtained from a 29-year-old male patient who was diagnosed with the disease at the age of 5 years; the patient was first described in Evgrafov *et al* (2004) and further clinical details can be obtained from Dierick *et al* (2008). Human iPSCs were generated by the VIB Stem Cell Institute (Leuven, Belgium) according to a previously published protocol (Takahashi *et al*, 2007). In brief, the human iPSCs were generated from the patient-derived fibroblasts using the CytoTune-iPS 2.0 Sendai Reprogramming kit (A16517, Thermo Fisher Scientific) which contains the Yamanaka reprogramming factors (Klf4, Oct3/4, Sox2, and cMyc). All factors were transfected using non-integrating Sendai virus vectors. The clearance of Sendai virus was confirmed with the TaqMan iPSC Sendai Detection kit (A13640, Life Technologies). The iPSCs were validated for their pluripotency and for the presence of the disease-causing P182L mutation in *HSPB1* using Sanger sequencing.

### Differentiation of motor neurons from iPSCs

Differentiation of the iPSCs into motor neurons was performed using a previously published protocol (Guo *et al*, 2017; Haidar *et al*, 2019). Briefly, iPSCs were cultured on Matrigel matrix (354234,

VWR Technologies) and in Essential 8 flex medium with supplement (A285501, Life Technologies) and penicillin–streptomycin. To differentiate the iPSCs, the Essential 8 flex medium was replaced by Neuronal medium, consisting of Neurobasal medium (21103049, Life Technologies), DMEM/F12 medium (11320033, Life Technologies) (both media in a 1:1 ratio), and further supplemented by N2 supplement (17502001, Life Technologies), B27 supplement without vitamin A (17504044, Life Technologies), GlutaMAX (35050061, Life Technologies), ascorbic acid (50-81-7, Sigma-Aldrich), and β-mercaptoethanol (31350010, Life Technologies). The dual SMAD inhibitors LDN (S2618, Selleck Chemicals) and SB (1614, Bio-Techne) and the WNT antagonist CHIR (4423/10, Bio-Techne) were added during the first 2 days. In the following days, we added the ensuing list of compounds to allow further differentiation and maturation of the motor neurons: retinoic acid (302-79-4, Sigma-Aldrich), smoothened agonist (SAG, 4366/1, Bio-Techne), γ-secretase inhibitor (DAPT, 2634/10, Bio-Techne), and the neurotrophic factors brain-derived neurotrophic factor BDNF (CSB-EL016134MO, Immunosource), glial cell-derived neurotrophic factor GDNF (21-8506-U002, Immunosource), and ciliary neurotrophic factor CNTF (21-7028-U005, Immunosource). Motor neurons were re-plated at $1 \times 10^5$ cells/ml in laminin-coated wells (Sigma-Aldrich). Additionally, to enhance the purity of the neuronal cultures, a single treatment of 1 μM ARA-C (cytosine β-D-arabinofuranoside, Sigma-Aldrich) was applied after re-plating on day 14. From day 18 onwards, the cells were cultured in this motor neuron maturation medium (containing BDNF, GDNF and CNTF), and the media was changed every other day by replacing half of the medium.

The differentiated motor neurons were fixed on day 37 in 4% paraformaldehyde (Sigma-Aldrich) for 20 min at room temperature. After fixation, cells were permeabilized with 0.5% Triton X-100 (Sigma-Aldrich) in PBS and blocked with 5% bovine serum albumin (BSA, Sigma-Aldrich) for 1 h. Cells were subsequently incubated with primary mouse monoclonal anti-HSPB1 (SPA-800, Enzo Life Sciences) and secondary AlexaFluor donkey-anti-mouse 488 antibody (Life Technologies). The nucleus was stained with Hoechst 33342 (Life Technologies). Cells were imaged with a Zeiss LSM700 laser scanning confocal microscope using a 40×/1.3 Plan-NeoFluar objective. Image analysis was done in the Fiji distribution of ImageJ (Schindelin et al, 2012; Schneider et al, 2012).

## NMR spectroscopy

All NMR spectra were recorded on a 14.1-T (600 MHz $^1$H Larmor frequency) Varian Inova spectrometer equipped with a 5 mm z-axis gradient, room temperature probe. All NMR data were processed with NMRPipe (Delaglio et al, 1995) and visualized with NMRFAM-Sparky (Lee et al, 2015). For quantitative purposes, peak shapes were fit with FuDA (Vallurupalli et al, 2008). For the WT and P182L peptides, 1D $^1$H NMR spectra were recorded at 25°C. The samples were prepared by dissolving natural-abundance peptides at a concentration of 2 mM in 30 mM sodium phosphate, 2 mM EDTA at pH 7 with 6% D$_2$O added to maintain lock. Spectra were recorded with 64 scans, a maximum acquisition time of 1 s, and an interscan delay of 1.5 s for a total duration of 2.7 min.

For the titration of $^{15}$N-cHSP27 with unlabeled peptides, sensitivity-enhanced 2D $^1$H-$^{15}$N HSQC spectra (Kay et al, 1992) were collected with 128* × 512* complex points in $t_1 \times t_2$ with maximum acquisition

times of 71 and 64 ms, respectively. A new sample was prepared for each titration point using the same stock solutions of $^{15}$N-cHSP27 and peptide, which avoids the need to account for sample dilution during the titration. Eight scans were collected per FID with a 1-s interscan delay, leading to a duration of 38.7 min per spectrum. Backbone resonance assignments of cHSP27 were performed previously (Alderson et al, 2019) and are available under BMRB accession code 27046. The titrations in Fig 3 were performed at 25°C, and the 10:1 peptide: cHSP27 sample was then used to record spectra from 25 to 50°C in 5°C increments (Appendix Fig S6B). To calculate the combined, weighted backbone amide CSPs for $^{15}$N-cHSP27 in the presence of increasing amounts of WT and P182L peptides, we used the following equation:

$$\Delta\delta = \sqrt{(\Delta H)^2 + (0.2\Delta N)^2} \tag{1}$$

where $\Delta H$ and $\Delta N$ are the differences in $^1$H$^N$ and $^{15}$N chemical shifts extracted from 2D $^1$H-$^{15}$N HSQC spectra of $^{15}$N-cHSP27 in the presence and absence of peptides. The factor of 0.2 that precedes $\Delta N$ accounts for the larger range of chemical shifts populated by the $^{15}$N backbone amide. To determine the $K_d$ value of WT and P182L peptide binding to the ACD, we fit CSPs from resonances that were in fast exchange to the following equation:

$$\Delta\delta = \Delta\delta_{\max} \frac{\left(([P]_t + [L]_t + K_d) - \sqrt{([P]_t + [L]_t + K_d)^2 - 4[P]_t[L]_t}\right)}{2[P]_t} \tag{2}$$

where $\Delta\delta$ is the measured CSP at a given peptide concentration, $\Delta\delta_{\max}$ is the maximum CSP observed, $[P]_t$ is the total protein concentration, $[L]_t$ is the total added ligand (peptide), and $K_d$ is the fitted dissociation constant. This equation assumes a two-state binding interaction, and the high-quality of the fits of the NMR data to a two-state model suggests that two IxI/V-containing peptides bind concurrently to the ACD dimer in the following manner:

$$M_2 + 2P \rightleftharpoons M_2P_2 \tag{3}$$

where $M$ stands for an ACD monomer, $M_2$ for an ACD dimer, and $P$ for a peptide. The ACD is dimeric under these conditions due to an interdimer disulfide bond involving C137 from both subunits. We note, however, that we cannot rule out a sequential binding model, as the observed NMR signals in fast exchange could also contain contributions from singly bound peptide forms of the ACD (Harkness et al, 2020). For resonances in fast exchange, the chemical shifts would reflect an ensemble-averaged position of the free dimer, the dimer with one peptide bound, and the dimer with two peptides bound (Harkness et al, 2020):

$$M_2 + P \rightleftharpoons M_2P + P \rightleftharpoons M_2P_2 \tag{4}$$

Our CSP data cannot differentiate between concurrent and sequential binding models; however, we note that previous native mass spectrometry experiments, which can resolve individual components in a heterogenous mixture, detected singly bound forms of the ADC from ABC (Hilton et al, 2013a). For fitting CSPs from resonances that were in fast exchange to equation 2, the following

residues were included: T91, D93, W95, W95 side chain, S98, L99, D100, A105, I120, K123, R140, Y142, L144, V148, D149, V153.

To verify the result obtained from fitting CSPs that are in fast exchange, we used the software package TITAN (Waudby *et al*, 2016), which numerically simulates, in Liouville space, the evolution of magnetization during a pulse sequence in the presence of chemical exchange. Therefore, the simulated 2D spectra can be compared to experimental spectra to fit protein–ligand binding kinetics and thermodynamics, even when resonances are not in fast exchange (equation 2). Residues included in the global fit were as follows: T91, D93, W95, W95 side chain, S98, L99, D100, A105, D107, L109, T110, K112, G116, E119, I120, K123, G132, R140, Y142, L144, G147, V148, D149, T151, V153, S154, S155, S156, L157, T162, L163, T164, and A167. For resonances that did not have final, peptide-bound chemical shifts available (K112, T151, S155, S156, L157), the peak positions were estimated based on the temperature titration (Appendix Fig S6) and the CPMG $|\Delta\omega|$ values (Appendix Table S1).

$^{15}$N CPMG RD data were recorded on $^2$H,$^{15}$N-cHSP27 at both 11.7 and 14.1 T (500 and 600 MHz $^1$H Larmor frequency) at 25°C. This enables the extraction of accurate parameters from analysis of the CPMG RD data by globally fitting data from different static magnetic field strengths. Datasets were collected with 70* × 512* complex points in $t_1 \times t_2$ with maximum acquisition times of 39 and 64 ms, respectively. A pseudo-third dimension was encoded by a variable delay between 180° refocusing pulses in a CPMG pulse train ($\tau_{CPMG} = 4\nu_{CPMG}^{-1}$), which were applied during a fixed relaxation delay, $T_{CPMG}$, of 39 ms. Datasets included 19 values of $\nu_{CPMG}$ with one duplicated value included for error analysis. Eight scans were collected per FID with an interscan relaxation delay of 3 s, leading to a duration of 18.3 h per spectrum. The effective transverse relaxation rates ($R_{2,eff}$) were calculated with the following equation:

$$R_{2,eff}(\nu_{CPMG}) = \frac{-1}{T_{CPMG}}\ln\left(\frac{I_{\nu_{CPMG}}}{I_0}\right) \quad (5)$$

In equation 5, $I_0$ stands for the intensity of a peak in a reference spectrum recorded without the $T_{CPMG}$ delay and $I_{\nu_{CPMG}}$ is the intensity obtained with the $T_{CPMG}$ delay and the corresponding value of $\nu_{CPMG}$.

In the CPMG RD experiments, the WT peptide was added to a final concentration of 80 μM, whereas P182L peptide was added at 200 μM. The protein concentration was 1.5 mM for both samples. To assess if the measured dispersions arose from inter- or intramolecular events, we recorded a second experiment at 600 MHz only with a different amount of peptide (23 μM for WT and 300 μM for P182L). The observed change in dispersions (Appendix Fig S4) indicated that the exchange was intermolecular in nature and thus reports on peptide binding. Moreover, residues with dispersions were within 4 Å of the peptide binding site. Two exceptions are T151 and S158, which are greater than 4 Å from the peptide but likely sense the N- and C-terminal residues of the peptide, E178 and E186, which were either absent (E178) or did not yield electron density (E186) in the X-ray structure (PDB: 4mjh). Two residues that are close to the peptide are V111 and S155, and these residues yield small dispersions ($R_{ex} < 2$ s$^{-1}$) that indicate a small $^{15}$N $|\Delta\omega|$, even though the V111 backbone carbonyl is hydrogen-bonded to T184 in the peptide (Fig 1 E; V111$^{CO}$-T184$^{HN}$) and S155 is *ca.* 3 Å from V183 in the $^{181}$IPV$^{183}$ motif. Despite the small $^{15}$N dispersions, the V111 and S155 resonances both become broadened during the peptide titration (Fig 3A,

Appendix Figs S5 and S8A), indicative of slow-to-intermediate exchange. In a 2D $^1$H-$^{15}$N HSQC spectrum, chemical exchange-induced broadening can arise from chemical shift changes to either the $^{15}$N or $^1$H nucleus or both (Waudby *et al*, 2020). Therefore, the broadening of V111 and S155 presumably arises from large $^1$H chemical shift changes upon peptide binding, with $\Delta\omega_H^{V111} = 0.3$ ppm and $\Delta\omega_H^{S155} = 0.5$ ppm, respectively, corresponding to 1,131 rad/s and 1,885 rad/s at 14.1 T. By contrast, peptide binding does not significantly affect the $^{15}$N chemical shifts of V111 and S155 ($\Delta\omega_N^{V111} = 0.2$ ppm, $\Delta\omega_N^{S155} = 0.1$ ppm; 90 rad/s and 45 rad/s at 14.1 T). Thus, the $^{15}$N dispersions arise from residues that are in the immediate vicinity of the bound peptide and experience significant $^{15}$N chemical shift changes upon binding.

Dispersions were fit to a model of two-state chemical exchange using the CATIA software (Vallurupalli *et al*, 2008) in which all residues shared the same $p_B$ and $k_{ex}$ value, with locally fit values of $|\Delta\omega|$ and the intrinsic transverse relaxation rate $R_{2,0}$. A total of 10 residues each from the WT and P182L peptide datasets were included in the global fit: T110, K112, T113, K114, T151, V153, S154, S156, L157, and S158 (Appendix Table S1, Appendix Fig S8). These residues were selected based on a large increase in $R_{ex}$ in the presence of peptide as compared to in the absence of peptide (Appendix Fig S9), where $R_{ex}$ is defined as the difference between $R_{2,eff}$ values at the lowest and highest $\nu_{CPMG}$ fields. In oxidized cHSP27, residues D129, E130, R136, C137, and F138 also show dispersions, but these residues were not included in the fit, as they arise from processes that are not related to peptide binding (Alderson *et al*, 2019). Between the two CPMG RD datasets derived from WT and P182L peptides, there is a large degree of similarity in the fitted $^{15}$N $|\Delta\omega|$ values (Appendix Fig S10), with an RMSD of 0.27 ppm. Because the $|\Delta\omega|$ values reflect the chemical shift changes upon binding to peptide, this agreement between the two independently fit datasets signifies the similarity of the conformations of the peptide-bound forms of cHSP27.

To convert the fitted $p_B$ and $k_{ex}$ values into $k_{on}$ and $k_{off}$, and calculate the $K_d$, the following equations were used:

$$k_{ex} = k_{on}[L]_{free} + k_{off} \quad (6)$$

$$k_{off} = (1 - p_B)k_{ex} \quad (7)$$

$$k_{on} = \frac{p_B k_{ex}}{[L]_{free}} \quad (8)$$

$$[L]_{free} = [L]_t - p_B[P] \quad (9)$$

$$K_d = \frac{k_{off}}{k_{on}} \quad (10)$$

where $[L]_{free}$ is the concentration of free ligand (peptide), $[L]_t$ is the total concentration of ligand (peptide), and $[P]$ is the concentration of protein. A table of $k_{on}$, $k_{off}$, and $K_d$ values derived from CPMG RD is included (Appendix Table S2) with the associated energetics (Appendix Table S3).

**Two- or three-state binding model**

With knowledge of the $K_d$, protein concentration, and ligand concentration, the expected percentage of bound ligand given a two-state interaction can be calculated as follows:

$$p_B^{calc} = \frac{K_d + [L]_t + [P]_t - \sqrt{(K_d + [L]_t + [P]_t)^2 - 4[L]_t[P]_t}}{2[P]_t} \qquad (11)$$

where $[L]_t$ and $[P]_t$ refer to the total concentration of ligand (peptide) and protein (cHSP27), respectively. Thus, at 80 μM of added WT peptide in the presence of 1.5 mM protein for a $K_d$ of 125 μM, approximately 4.9% of cHSP27 should be bound. For the P182L peptide at a concentration of 200 μM in the presence of 1.5 mM protein for a $K_d$ of 1,200 μM, approximately 7.2% of cHSP27 should be bound. These calculations yield significantly larger values of $p_B$ than what was experimentally measured by CPMG RD: 2.4 and 2.0% for WT and P182L, respectively.

While the CSP (Fig 3), lineshape (Appendix Fig S5), and CPMG RD (Fig 4, Appendix Figs S8 and S9) data all fit well to a model of two-state binding, with values of reduced $\chi^2$ near 1.5 for the CPMG RD fitting, the potential presence of a third state is evident from the discrepancy between the $K_d$ from CSP data (equation 11) and the $K_d$ measured by CPMG RD. Moreover, the kinetic parameters extracted from the lineshape analysis with the software TITAN (Appendix Fig S5) and the large fitted values of $\Delta R_2$ ($R_{2,\text{bound}} - R_{2,\text{free}}$) (Appendix Fig S7B and C) also point to the presence of a third state.

Thus, the discrepancy in binding affinity as measured by CSPs, lineshape fitting, and CPMG RD could arise from a third state. This is exemplified by the disappearance of NMR signal intensity from residues in the β4/β8 groove (slow exchange) early in the titration at 25°C (Fig 3A and B), which then do not recover their intensity at the end of the titration with *ca.* 95% of the peptide-bound form (Appendix Fig S6A). The intensity is only recovered at high temperatures (Appendix Figs S6B and S7A), which suggests that the conformational exchange event responsible for the broadening at 25°C has become sufficiently rapid so as to shift the exchange regime from intermediate to fast on the chemical shift timescale. The source of the missing intensity in the titrations, and lower than expected $p_B$ values in the CPMG RD analysis, could be caused by significant heterogeneity of the peptide when bound to cHSP27, leading to multiple interconverting states on the micro-to-millisecond timescale. Multiple conformations of peptide-bound ABC were previously identified with NMR (Delbecq *et al*, 2012), with such heterogeneity perhaps arising from the palindromic nature of the ABC peptide (Ac-P*ERTIPITRE*EK-NH$_2$; palindrome italicized). However, while HSP27 does not have a palindromic sequence, it does have multiple IxI/V motifs: the canonical $^{181}$IPV$^{183}$ motif as well as the $^{171}$ITI$^{181}$ motif. The fact that both CSP and CPMG RD data are well-fit to a model of two-state chemical exchange thus implies that the third state rapidly interconverts with one of the two states probed here. Therefore, a plausible binding model may look like one the scheme below:

$$M_2 + 2P \rightleftharpoons \{M_2P_2\}_A \rightleftharpoons \{M_2P_2\}_B \qquad (12)$$

where the two different bound forms interconvert, which is similar to a previous NMR report (Korzhnev *et al*, 2009). As mentioned, the region near the IxI/V motif in HSP27 contains a second possible binding site, $^{179}$ITI$^{181}$. There may be conversion between IxI/V-bound states on the micro-to-millisecond timescale. Alternative explanations may be segmental motions within the bound peptide or a lock-and-dock type of binding mechanism in which the peptide is initially flexibly tethered before stably binding. While

comparing the two different bound states identified in cHSP27 remains beyond the scope of this paper, the $^{15}$N |Δω| values (i) are highly similar for the WT and P182L peptides (Appendix Fig S9), (ii) seem to agree with the chemical shift changes directly observed in the 2D $^{1}$H-$^{15}$N HSQC spectrum of the peptide-bound state (Appendix Fig S6), and (iii) show excellent correspondence when fit with a naïve two-state model or the global three-state model (vide infra) (Appendix Figs S8 and S11). Collectively, these results suggest that the conformations of the peptide-bound forms are highly similar between the WT and P182L forms and that the CPMG RD and titration data are probing the same end-state conformation. The two-state fits, while sufficient to obtain qualitative insight, do not agree quantitatively. Therefore, we sought to fit the CPMG RD and titration datasets globally.

The specific numerical challenges when comparing two- and three-state analyses have been addressed (Neudecker *et al*, 2006). A robust three-state analysis of CPMG RD data requires a large amount of data in order to accurately traverse the rugged parameter space such that the true minimum is reached (Neudecker *et al*, 2006). In lieu of such data, we performed a global analysis of the data by imposing the $K_d$ measured by CSPs and lineshape analysis on the CPMG RD data. Thus, we fixed the value of $p_B$ to that expected based on the $K_d$ and known concentrations of protein and ligand. In addition, to account for the multiple exchanging conformations in the bound state, we allowed the parameter $\Delta R_2 = R_{2,B} - R_{2,A}$ to adopt non-zero values. In this notation, $R_{2,A}$ is the intrinsic transverse relaxation rate of the ground state (i.e., $R_{2,0}$) and $R_{2,B}$ is the transverse relaxation rate of the excited state. Large, non-zero values of $R_{2,B}$ would be expected to arise from exchange within the bound state on the micro-to-millisecond timescale. Thus, we fit the CPMG RD data at 500 and 600 MHz using a global value of $k_{ex}$, local values of |Δω|, $R_{2,A}$, $R_{2,B}$, and a fixed global value of $p_B$. By fitting the data as such, we obtain highly similar values of $^{15}$N |Δω| that agree with the standard two-state fitting procedure in which the fitting is performed with $k_{ex}$ and $p_B$ globally floated alongside locally fit values of |Δω| with $\Delta R_2 = 0$ (Appendix Fig S11). The reduced $\chi^2$ values are essentially identical between the two fitting methods: 1.55 and 1.39 for the naïve two-state and the global method, respectively. In addition, the fitted $^{15}$N |Δω| values agree very well between the two fitting methods (RMSD = 0.37 ppm).

## Molecular dynamics simulations

All-atom MD simulations of peptides containing the IxI/V motif and the P182L variant were performed using the Gromacs version 4.5.3 simulation package (Berendsen *et al*, 1995) with the AMBER99SB-ILDN force field and the TIP3P water model. To obtain a starting conformation of the IxI/V-containing peptide, chain B comprising residues $^{179}$ITIPVTFE$^{186}$ of HSP27 was extracted from the crystal structure of the ACD bound to its IxI/V motif (PDB: 4mjh). The missing atoms were added with the Modeller web server (Fiser *et al*, 2000). The P182L variant of the peptide was obtained by *in silico* mutagenesis of the P182 residue using PyMOL. After adding the missing atoms, including hydrogens, the WT and P182L peptides, respectively, contained 134 and 139 atoms.

Each MD simulation was performed in a rhombic dodecahedron of size 5.30 by 5.30 by 3.75 nm, which contained 3403 (WT) or 3401 (P182L) water molecules and one sodium ion to maintain

                    

charge neutrality. A total simulation time of 200 ns was achieved using a time step of 2 fs with $10^7$ total steps. The temperature was maintained at 37°C using the velocity-rescaling thermostat (Bussi *et al, 2007*), and the average isotropic pressure was kept at 1 bar with the Parrinello-Rahman barostat (Parrinello & Rahman, 1981). Long-range electrostatics were calculated with the Particle Mesh Ewald summation. Both peptides were subsequently energy minimized and equilibrated for 20 ns prior to analysis. Quantitative analyses of the trajectories including dihedral angle calculations and PCA were performed using in-house Python scripts and the MDTraj Python package (McGibbon *et al, 2015*). The PCA was performed by combining the WT and P182L trajectories into a single trajectory, extracting only the Cα atoms, and aligning all frames to the crystal structure of the bound form of the WT IxI/V motif (PDB: 4mjh). The first two principal components were then calculated from this combined trajectory, with the residues at the N- and C-termini excluded from the analysis.

## Bioinformatic analyses

[V/I]P[V/I] motifs were quantified in the reference human proteome that contained only the longest transcript of each gene, with the UniProt Proteome ID UP000005640 (accessed May 2019). We predicted regions of intrinsic disorder comprising 20 residues or longer by running the DISOPRED3 software (Jones & Cozzetto, 2015) with default settings and the UniRef90 database. An in-house Python script making use of the Biopython package (Cock *et al, 2009*) was written to search for and count specific peptide motifs within the proteome ($M_{\mathrm{proteome}}$) and the database of disordered regions ($M_{\mathrm{disordered}}$; $M_{\mathrm{disordered}} \subseteq M_{\mathrm{proteome}}$). To count specifically within structured regions of the proteome ($M_{\mathrm{structured}}$), we subtracted $M_{\mathrm{disordered}}$ from $M_{\mathrm{proteome}}$ for each queried tripeptide $j$ (1, ..., $N$) in which the central position was iterated over each amino acid $i$ (1, ..., 20), as such:

$$M_{\mathrm{structured}} = \sum_{j=1}^{N} \sum_{i=1}^{20} \left[ M_{\mathrm{proteome}_i} - M_{\mathrm{disordered}_i} \right]_j \tag{13}$$

Removing the summation over $i$ in equation 13 yields the count of specific tripeptides that include a fixed amino acid at the central position, *e.g.*, [V/I]P[V/I]. To calculate the expected number of peptide motifs found in each database based on amino acid frequencies, we used the following equation:

$$N_{r,n} = M_{r,n} \prod_{q=1}^{n} p_{r,\,\mathrm{AA}_q} \tag{14}$$

where $p_{r,AA}$ is the frequency of amino acid $AA$ in the database $r$ ($r$ = [proteome, structured, disordered]) and $M_{r,n}$ is the total number of peptides of length $n$ in the database $r$. The product sum is calculated by computing $p_{r,AA}$ iteratively for each position $q$ in the queried motif ($q$ = 1, 2, ..., $n$). This yields $N_{r,n}$, the expected number of peptides of length $n$ in the database $r$. This assumes that each amino acid in the queried peptide motif has an independent probability of appearing next to any other amino acid, and therefore, we do not account for selective pressures imparted by, e.g., co-evolution of specific motifs. As an illustrative example, the queried tripeptide VSI within the database of disordered regions

($M_{\mathrm{disordered,3}}$ = 2,945,909) with frequencies of Val, Ser, and Ile corresponding to $p_{\mathrm{disordered,\ Val}}$ = 0.04440, $p_{\mathrm{disordered,\ Ser}}$ = 0.1195, and $p_{\mathrm{disordered,\ Ile}}$ = 0.0251 yields an expected total of 392 VSI motifs. This value agrees closely with the observed number of VSI motifs (376).

To assess statistical significance of the observed numbers of motifs, *i.e.*, to test if the [V/I]x[V/I] motifs can be accurately described by the underlying amino acid frequencies, we calculated $\chi^2$ values:

$$\chi^2 = \sum_{i=1}^{n} \frac{(\mathrm{observed}_i - \mathrm{expected}_i)^2}{\mathrm{expected}_i} \tag{15}$$

For the four motifs (IxI, IxV, VxI, VxV), the total $\chi^2$ values ($\chi^2_{\mathrm{tot}}$) were computed for each amino acid x = {A, C, D, ..., V, W, Y}, with expected values determined by equation 14 based on amino acid frequencies from the databases of structured or disordered regions:

$$\chi^2_{\mathrm{tot}} = \sum_{j=1}^{4} \sum_{i=1}^{20} \frac{\left( M_{i,\mathrm{observed}} - N_{i,\mathrm{expected}} \right)_j^2}{\left( N_{i,\mathrm{expected}} \right)_j} \tag{16}$$

where the indices $i$ and $j$, respectively, refer to the amino acid at position x = {A, C, D, ..., V, W, Y} and the queried motif (IxI, IxV, VxI, VxV); the symbol $(M_{i,\mathrm{observed}})_j$ refers to the observed number of $j$ motifs with residue type x = $i$ in the queried database; and $(N_{i,\mathrm{expected}})_j$ refers to the calculated number of $j$ motifs with residue type x = $i$ in the queried database (obtained with equation 14). When collectively analyzing the four motifs for either the disordered or structured region database, there is a total of $\nu$ = 76 degrees of freedom. The $\chi^2_{\mathrm{tot}}$ values are 2,849.9 and 1,081.2 for the structured and disordered regions, respectively. The null hypothesis is rejected, as the obtained *P*-values are highly significant, i.e., $P[\chi^2(\nu) > 119.85]$ = 0.001 for $\nu$ = 76. Therefore, the frequency of [V/I]x[V/I] motifs cannot be described by amino acid frequencies alone.

The Python code that was used for these analyses and the list of disordered regions longer than 20 residues are available from GitHub (https://github.com/reidalderson/bioinformatics).

# Data availability

This study includes no data deposited in external repositories.

**Expanded View** for this article is available online.

## Acknowledgements
We are grateful to Heath Ecroyd (University of Wollongong) and Josée N. Lavoie (University of Laval) for, respectively, providing the HSP27 and BAG3-GFP expression plasmids, Erik Marklund (Uppsala University) and Matteo T. Degiacomi (Durham University) for insightful discussions, Georg K. A. Hochberg (Max Planck Institute for Terrestrial Microbiology) for analysis of HSP27 orthologues, and Katrien Spaas of the VIB CMN Neuromics Support Facility and Vicky De Winter for assistance in expansion microscopy sample preparation and cloning, respectively. We thank Michaela Auer-Grumbach (Medical University of Vienna, Austria) for providing the fibroblasts of the P182L patient, which were used to generate iPSCs. We thank former laboratory members Mansour Haidar and Thomas Geuens for help with the interactomics experiment, which

was reanalyzed in this study. We also thank Ite Verstraeten and Nik Brusse-laers for assistance with experiments. This research was supported in part by the University of Antwerp (TOP-BOF to V. T. and DOC-PRO4 PhD project to J. V. L. and V. T.), the Research Foundation-Flanders (FWO-Vlaanderen; to V. T.), the Medical Foundation Queen Elisabeth (GSKE; to V. T.), the American Muscular Dystrophy Association (MDA; to V. T.), the Association Belge contre les Maladies Neuromusculaires (ABMM; to E. A. and V. T.), the Rotary "Hope in Head" program (to E. A. and V. T.), and the H2020 Solve-RD program "Solving the unsolved rare diseases" under grant agreement 2017-779 257 (to V. T.). E. A. is supported by a FWO Postdoctoral grant (Project number 76110). V. T. is a member of the µNEURO Center of Excellence of the University of Antwerp. T. R. A. acknowledges funding from the NIDDK and the NIH Oxford-Cambridge Scholars Program. A. J. B. holds a David Phillips Fellowship from the Biotechnology and Biosciences Research Council (BB/J014346/1). J. L. P. B. thanks the Engineering and Physical Sciences Research Council (EP/J01835X/1) and Biotechnology and Biosciences Research Council (BB/J018082/1).

## Author contributions

TRA, EA, BA, VT, AJB, and JLPB conceptualized the study. TRA, EA, BA, IP, JVL, HYG, MAW, JML, VT, AJB, and JLPB involved in investigation. TRA, EA, BA, VT, and JLPB wrote original draft. TRA, EA, BA, IP, JVL, HYG, MAW, JML, VT, AJB, and JLPB involved in writing—reviewing and editing.

## Conflict of interest

The authors declare that they have no conflict of interest.

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
