## [Review Process File · The EMBO Journal]

A weakened interface in the P182L variant of HSP27 associated with severe Charcot-Marie-Tooth neuropathy causes aberrant binding to interacting proteins

T. Ried Alderson, Elias Adriaenssens, Bob Asselbergh, Iva Pritisanac, Jonas Van Lent, Heidi Gastall, Marielle Wälti, John Louis, Vincent Timmerman, Andrew Baldwin, and Justin Benesch
DOI: [10.15252/emboj.2019103811](https://doi.org/10.15252/emboj.2019103811)

Corresponding author(s): T. Ried Alderson (reid.alderson@utoronto.ca) , Justin Benesch (justin.benesch@chem.ox.ac.uk), Andrew Baldwin (andrew.baldwin@chem.ox.ac.uk), Vincent Timmerman (vincent.timmerman@uantwerpen.be)

Review Timeline:

Submission Date:	5th Nov 19
Editorial Decision:	10th Dec 19
Revision Received:	30th Mar 20
Editorial Decision:	30th Apr 20
Revision Received:	26th May 20
Editorial Decision:	27th Jul 20
Revision Received:	14th Dec 20
Editorial Decision:	7th Jan 21
Revision Received:	10th Jan 21
Accepted:	14th Jan 21

Editor: Karin Dumstrei

Transaction Report:

Dear Dr. Alderson,

Thank you for submitting your manuscript to the EMBO Journal. Your study has now been seen by three referees and their comments are provided below.

As you can see below the referees find the analysis interesting but they also indicate that significant revisions are needed to consider publication here. If you are willing to significantly extend the analysis and address the raised concerns with inclusion of additional data then I am open to consider a revised version. Please note that we would need some further validation that protein interaction with P182L is affected by looking at some of the other proteins identified besides BAG3 (ref #1)

I should add that it is EMBO Journal policy to allow only a single major round of revision and that it is therefore important to address the raised concerns at this stage.

When preparing your letter of response to the referees' comments, please bear in mind that this will form part of the Review Process File, and will therefore be available online to the community. For more details on our Transparent Editorial Process, please visit our website:

<https://www.embopress.org/page/journal/14602075/authorguide#transparentprocess>

Thank you for the opportunity to consider your work for publication. I look forward to your revision.

Yours sincerely,

Karin Dumstrei, PhD
Senior Editor
The EMBO Journal

Further information is available in our Guide For Authors:

The revision must be submitted online within 90 days; please click on the link below to submit the revision online before 9th Mar 2020.

Link Not Available

Referee #1:

HSP27 is a human small heat shock protein that forms dynamic oligomers. Point mutations in the C-terminal IXI motif of HSP27 are associated with Charcot-Marie-Tooth disease. The authors observe that the P182L mutation in this motif results in Hsp27 oligomers of increased size, lower chaperone activity and enhanced aggregation of substrate proteins. The cause for this seems to be a lower affinity of the mutated motif to the conserved alpha crystallin domain (ACD) of HSP27 as shown by NMR experiments. This might result in the enhanced binding of HSP27 interacting proteins which also contain [I/V]P[I/V] motifs. This is shown directly for the known interactor BAG3. Overall the data and the outlined mechanistically explanations are of interest for the field of small heat shock proteins. The authors put some previously reported findings into context and provide a model explaining the effect of this mutation. Some of the results reported in the manuscript require further support.

Specific comments

The authors refer to HSP27 also as HSPB1. Please adjust.

Figure 1B compares the CTD of small heat shock proteins from different organisms. Lower

organisms do not contain a proline at position 182. What consequences do the results shown in this manuscript have for the understanding of these sHSPs?

The P182L mutant protein was purified under denaturing conditions and refolded. The authors need to control for the structural integrity of the resulting protein. The experiments rely on the correct structure of the HSP27 ACD. In figure S1A, the P182L variant seems to be more aggregation prone, indicative of a potentially unstable, not correctly folded protein. Please include the PDB code for the 3D structure in figure 1D.

The authors do not discuss the oxidation state of the full length proteins in the methods section. They could show a non-reducing gel for the (presumably reduced) protein they used for the full-length protein experiments. The C137S mutant - according to the materials and methods section - was used for gel filtration, but the protein used for NMR was oxidized, hence a different polypeptide. Substitution of proline at position 182 with an isoleucine results in a dramatically reduced affinity of CTD with respect to ACD. How does this result explain the change of the oligomeric size of the protein?

In figure 2B there is a very limited amount of particles present in the negative stain micrograph. Please show a larger field/a more densely populated area.

Figure 2 C: The result shown in the "Expansion" panels is astonishing considering the "Confocal" image right above. At this low magnification (and resolution) both cell lines look quite similar. It would be helpful and interesting to include a normal confocal scan of the pre-expansion sample at a higher magnification. This would allow comparison of the expansion microscopy data (higher resolution of course) to the lower-resolution fluorescence distribution. The drastic difference observed after expansion should also be evident at normal confocal resolution.

In figure 2D, the spot density is lower according to the data shown. However, the expression level of HSP27P182L seems also lower than that of HSP27 according to figure 5C. The authors should discuss this. Also, the fluorescence of each spot seems to roughly double on average (cf. Fig 2E). Combined, this could explain the reduced number of foci.

Figures 3/4: The studies on the affinities of the peptides are limited to the isolated ACD and NMR. If the concluded mechanism is right, the wildtype and mutant peptides should be able to influence the oligomer distribution of wt HSP27 (and vice versa influences should be observed for the P182L peptide). The respective titration experiments with full length Hsp27 are of high interest and needed to test the claimed mechanism.

Can the results obtained for the ACD experiments be directly be transferred to full-length protein? The NMR data were recorded using the oxidized ACD whereas the other experiments were performed with full-length (presumably) reduced protein.

In the NMR experiments to determine the affinity of the C-term domain peptides to HSP27 the authors see differences in affinity of about one order of magnitude between the assays. It seems an orthogonal method is needed in this context.

The authors don't list the residues that were used to calculate the K_d using the CSP data. They however provide this information in the methods section for the CPMG data. The residues should also be given for CSP fitting.

The authors did not assess the effects of the P182S mutant which also causes Charcot-Marie-Tooth disease. Ser seems to be the second most highly populated residue in disordered regions after Pro. It would be interesting to see how this mutation behaves in some of the assays.

The authors argue that for the wild type Hsp27 only resonances of the disordered CTR were observed. Other studies have shown, however, that it is also feasible to observe ordered regions of small heat shock proteins using MAS solid-state NMR (e.g. Jehle et al., 2010). There, an ordered CTD was observed. It would be interesting to compare solid-state NMR spectra of wt and P182L to better understand the structure and dynamics of the CTD-bound state.

V111 is shown to be hydrogen-bonded to the CTR peptide in figure 1E. According to figure 3A, it is an intermediate exchange residue. However, despite its potential relevance for CTR binding, it is

not shown or discussed in the CPMG experiments. Could the authors comment on this? The authors claim that at least 22 proteins should be affected in their interaction with P182L. Besides mentioning previous studies on PCBP1 they only assess one of the claimed interaction partner, BAG3. Data on additional proteins seem required to support a general model. Based on figure 5C, the authors claim stronger binding of BAG3 to HSP27-P182L in co-IP experiments. Based on ECL luminescence, more BAG3 seems to be present in the Co-IPs with HSP27-P182L. But it seems that more BAG3 than HSP27-P182L is present in the complex. How can this be explained with the proposed model assuming a single binding site for IXI on HSP27? Figure 5D: Why is the control switched from actin to tubulin here? Why does BAG3 not pull HSP27 in this experiment? Here and in figure 5C, quantification of the ratio of the bands seems required. The authors postulate that HSP27 will interact with HSP27P182L in vivo sequestering 'functional' Hsp27 into 'dysfunctional' Hsp27P182L complexes, hence explaining the dominance of the mutant allele. The authors do have both proteins at hand. So hetero-oligomer-formation could be assessed. Concerning the bioinformatic analysis the authors need to clarify their view on the IXI binding site in the ACD-beta4-beta8 groove. Is this a "substrate" binding site? Or do the authors envision this site as a specific binding site for interaction partners regulating the oligomer distribution of HSP27? Overall, this genome-wide analysis is interesting. But according to figure S7 C,D, VDV and VPV look much more interesting than IPV (largest deviation. The authors do not discuss these. Adding some statistical analysis would enable the reader to judge the relevance of the highlighted [IV]P[IV] enrichment independently. It would also allow the assessment of other potential enrichments like [V/I]S[V/I] (cf. Figs 5A, S7E), VDV, VPV (cf. Fig S7D) or depletions as central Phe, Trp, Tyr (cf. Fig S7E) or VRV, IGV (cf. Fig S7D). The bioinformatics section needs to be streamlined and reworked.

Referee #2:

Review for:EMBOJ-2019-103811

Dysregulated interactions triggered by a neuropathy-causing mutation in HSP27

The goal of this work is to investigate the alterations in the structure-function relationship of the inherited mutation P182L in human heat shock protein Hsp27 by investigating the oligomeric size, distribution, and chaperone activity of the mutant protein. This is a mutation that is linked to Charcot-Marie-Tooth disease, a prevalent neuropathy. The mutation is located in a well-conserved IXI motif and it disrupts protein-protein interactions, leading to more interactions with other IXI motif proteins (Bag3) and increases sHsp oligomeric size. This study is of moderate-high significance as it suggests a general mechanism for how IXI motif proteins interact with one another, which is important for maintenance of many cellular protein-protein interaction networks. This is a well-written, thorough study that demonstrates disruption of the IXI motif alters oligomerization of Hsp27 and increases interactions with other IXI motif-containing proteins. Interactions with Hsp27 ACD is decreased through a mechanism that likely involves an increase in conformational space sampling by the CTR.

Minor Concerns:

Results:

1. Are there similar levels of protein expression of Hsp27 P182L and wildtype protein? Comparison

of the oligomeric distributions is made in Figure 2, but are overall levels similar?

2. Binding of the mutant peptide to the ACD is similar in location to the wildtype peptide. CD spectroscopy could confirm altered configuration of the mutant peptide.

3. Does P183L bind to multiple other proteins in vivo aside from BAG3 that was expressed?

Discussion:

1. Why would disrupting IxIV motif (and self-association) lead to larger oligomers?

2. Are the larger oligomers formed solely from Hsp27?

3. Is the decrease in chaperone activity due to oligomerization disruption?

4. Any benefit to enhancing IxIV motif interactions?

5. Are the IxIV motif interactions enhanced if more conformational space is sampled?

Referee #3:

The manuscript by Alderson et. Al. deals with the HSP27 human chaperone. The authors address how a point mutation in the HSP27 C-terminal disordered region (P182L) i) increases oligomer formation and ii) reduces chaperone activity.

I have a number of remarks regarding the manuscript, these often refer to the interpretation of the data and should be discussed in detail. Currently, I have the feeling that the authors fail to clearly explain the "unexpected mechanistic explanation for the consequences of a mutation in the conserved IxIV motif, which causes a severe form of CMT.", as they write in the discussion.

Major remarks:

Does the P182L mutation stimulate aggregation? This appears to be the case, especially in figure 1G. Do the blue lines actually represent the aggregation of the substrate, or rather the aggregation of the substrate plus HSP27. In that case the more unstable P182L version of HSP27 obscures the data. Please comment and correct if needed, e.g. by using a significantly lower concentration of HSP27 compared to substrate.

The negative stain EM images (Figure 2B) are unclear and can most likely not be used to draw conclusions regarding aggregation size. First: the shown views are very small and aggregates of various sizes can always be found on EM grids. A statistical analysis of a large number of particles is required to determine cluster sizes. Second, many proteins spontaneously aggregate on EM grids due to the harsh staining conditions. The particle size that is visible on the grids is thus not necessarily representative of particle size in solution. This can also be seen from a comparison of panels A and B in Figure 2. Panel A shows that the oligomer size of the P182L protein is a factor of ~50 larger than the WT protein. In the EM, the difference is much less.

Figure 2C-E. This analysis can only hold in case the WT and the P182L proteins are expressed at the same levels. Please provide proof for that. The spot-density multiplied with the spot-intensity should e.g. be the same for both proteins. Page 8, Line 8: please reword drastic to two-fold.

Figure 3B, C. Why are the residues in panel B not fitted in panel C? Please show the same residues in B and C and some of those also in A.

Page 10: "The P182L mutation thus lowers the binding affinity for the ACD by nearly one order of magnitude from ca. 125 μ M to 1 mM". This sounds like a lot, but the difference is only slightly more than that factor of 2 in DELTA-G. The error in the K_D for the P182L peptide appears very small as the endpoint of the titration is not defined. The authors should take uncertainties in the protein and peptide concentration into account when extracting the errors of the affinities.

Figure 3E: the peptide is hardly visible.

Fitted exchange rates and populations need to be provided for the CPMG experiments and should be discussed in detail also in the main text, now only the material and methods contain significant discussions (speculations).

Do the DELTA ω values that are fitted in the CPMG analysis agree with the delta ω values that are visible in the spectra or that are fitted in the K_D titration experiments?

Do the k_{on} and k_{off} rates agree with the rates that are reported by TITAN?

Why are the k_{off} rates used together with the K_D to get the k_{on} rates. One could also have used the k_{on} rates with the K_D to get the k_{off} rates. If the authors had done that, I assume that they would observe a difference in the k_{off} rates and that they would conclude that the differences in k_{off} determine the differences in K_D . This would contradict the conclusion that "...NMR data point to attenuated affinity and association rate of the P182L IxIV motif."

The MD simulations of the WT and P182L peptides shows that the removal of the proline increases flexibility. This finding is trivial and as expected. The authors then link the "preformed" conformation of the WT peptide with its increased affinity. It is clear that bi-molecular interactions are much more complex than that. Differences in solvation, flexibility of the peptide in the binding pocket (entropy), differences H-bonding propensity, differences in hydrophobicity and much more also determine the final affinity. The correlation that the authors find is most likely only one small part of a much more complex binding event. This should be clearly mentioned as the current presentation of the data is a huge over-simplification.

The bioinformatic analysis is limited and very basic. The authors find that X=PRO is most likely in the [LV]-X-[LV] motif. This is hardly surprising, as PRO is a residue that interferes with secondary structure elements. Maybe the graph (Figure 5) should be scaled by the abundance of the individual aminoacids in disordered regions. In many regards, the lower panel of Figure 5 just shows the propensity that amino acids have to be in disordered regions (independent of the flanking [LV] regions).

The authors identified proteins that contain [LV]-P-[LV] motifs in a number of proteins. I am not sure if the GO analysis reveals any insights. The number of motifs in the most enriched groups and the group-size for the most enriched groups are very small (3 out of 21). The fold enrichment is thus likely highly uncertain. In addition, what do the authors conclude from the analysis. Does it make sense that many different processes are found? Is the chaperone known to be involved in the processes that are identified?

Page 19: ". Indeed, we created a BAG3 mutant in which the IPV motifs were mutated to GPG". Based on the paper, I wonder if the authors also tested an ILV mutant for the BAG3 protein, as the mutant in the HSP29 protein is also a PtoL. This makes the experiments somewhat inconsistent. The GPG motif should have the same preformed conformations as that IPV motif, showing that the preformed conformation is not the only thing that counts (see above).

The authors fail to address why the P182L mutant forms larger oligomers in solution. Based on the reduced affinity between the unfolded part of HSP29 and the folded core, I would expect smaller assemblies. In the discussion they mention a mechanism by which "concomitant binding of two IxIV motifs facilitates subunit exchange via ejection of a monomeric subunit from the oligomer". I am afraid that I really don't understand this. What are the two IxIV motifs binding to? In my understanding, they can bind to one surface patch, either inter- or intramolecular. The P182L mutation weakens both (as it is the same interaction) and the oligomers should be smaller. Please explain in detail.

Page 20: "Therefore, attenuating the binding affinity of the IxIV motif in the CTR would enable other sites in the NTD to bind more frequently and alter the oligomeric landscape of HSP27." Why would the oligomeric state be influenced when another part of the NTD interacts with the core domain? The intermolecular interaction remains present. Please explain.

Finally, I fail to understand why the P182L mutation has reduced activity. The active surface patch of the chaperone is more accessible, and the activity should thus increase. Is the reduced activity solely due to the increase cluster size? Then the chaperone activity could be increased at a lower concentration, where oligomers are more likely to be smaller. Did the authors test this? The

Minor remarks:

References to the structure should be included in the figure legend 1D and E.

We are grateful for the editor's and reviewers' interest in our study, and their careful reading of our manuscript. We appreciate their positive opinions of our work, finding it a "well-written, thorough study" and stating that the "*mechanistic[ally] explanations are of interest for the field of small heat shock proteins*".

However, there were two main queries raised: a comparison of the expression levels of the WT and P182L forms, and whether other interactors beyond BAG3 bind more tightly to the P182L form of HSP27. We have responded to these points at length in what follows, as well as addressing the other, relatively minor, comments. Each query is detailed, with our specific changes described, below.

Responses to the Reviewers' Comments

The reviewers' original comments are shown in *italic* and our replies are coloured blue.

Referee #1:

HSP27 is a human small heat shock protein that forms dynamic oligomers. Point mutations in the C-terminal IXI motif of HSP27 are associated with Charcot-Marie-Tooth disease. The authors observe that the P182L mutation in this motif results in Hsp27 oligomers of increased size, lower chaperone activity and enhanced aggregation of substrate proteins. The cause for this seems to be a lower affinity of the mutated motif to the conserved alpha crystallin domain (ACD) of HSP27 as shown by NMR experiments. This might result in the enhanced binding of HSP27 interacting proteins which also contain [I/V]P[I/V] motifs. This is shown directly for the known interactor BAG3.

Overall the data and the outlined mechanistically explanations are of interest for the field of small heat shock proteins. The authors put some previously reported findings into context and provide a model explaining the effect of this mutation. Some of the results reported in the manuscript require further support.

Specific comments

1.1 *The authors refer to HSP27 also as HSPB1. Please adjust.*

We thank the author for noting this, and we have now corrected the issue. Specifically, we replaced "HSPB1" in the legend of Figure 5 and the Methods section with "HSP27". When discussing the protein and gene, we respectively use HSP27 and *HSPB1*.

1.2 *Figure 1B compares the CTD of small heat shock proteins from different organisms. Lower organisms do not contain a proline at position 182. What consequences do the results shown in this manuscript have for the understanding of these sHSPs?*

We thank the reviewer for posing this intriguing question; it is true for many other sHSPs in non-metazoan organisms that the x in the IxI/V motif is not a proline. However, we are hesitant to extrapolate our results on the consequences of mutating the proline in HSP27 to make conclusions about distantly related sHSPs. This is because the evolutionary trajectories of HSP27 and non-metazoan sHSPs are separated by >400 million years, and consequently differ in sequence at many positions other than in the IxI/V. This background of differences beyond the change at a single position has a profound effect that means it is not possible from alignment alone to ascertain what caused different states to become established in different

lineages. They may have been fixed by selection, or by drift; in either case the ancestral state can become unacceptable later (“epistasis”).

Given that inferences beyond close relatives to HSP27 are fraught when using sequence alignments alone, we recognise that our existing Figure 1B was ill-judged. We have therefore replaced it with an alignment of HSP27s from different vertebrates, showing how this proline is highly conserved.

Our results in this manuscript indicate that the interaction between the IxI/V motif to the ACD of HSP27 limits the maximum size of the oligomers; weakening this interaction causes dysregulated oligomerization that leads to very large, otherwise non-native oligomers. We propose that the IxI/V motif acts as a *regulator* of oligomerization in HSP27 (and possibly other vertebrate sHSPs), and that perturbing its finely tuned binding affinity dysregulates this process.

1.3 *The P182L mutant protein was purified under denaturing conditions and refolded. The authors need to control for the structural integrity of the resulting protein. The experiments rely on the correct structure of the HSP27 ACD. In figure S1A, the P182L variant seems to be more aggregation prone, indicative of a potentially unstable, not correctly folded protein.*

This is an important point raised by the reviewer. To address this issue in the manuscript, we have compared circular dichroism (CD) spectra of wild-type HSP27 and the P182L variant after purification from inclusion bodies. Importantly, upon overexpression in *E. coli*, the wild-type version of HSP27 is present in the soluble fraction and, thus, its CD spectrum provides a reference for the natively folded form.

As shown in the figure below, the CD spectrum of the P182L variant, which was purified under denaturing conditions and refolded, matches the spectrum of WT HSP27. This indicates that the refolded form of P182L adopts the same secondary structure as WT HSP27. Notably, the β -strand-rich conformation is evident from profile of the CD spectra and the minimum at 218 nm. Both proteins were present at a concentration of 20 μ M in 20 mM sodium phosphate, 100 mM NaCl, 2 mM EDTA at pH 7.4. This figure is now included in a modified Appendix Figure S1, and provides good evidence that the mutant’s aberrant oligomerization is not some artefact due to improper folding.

Figure included for comment #1.3 from Reviewer 1: Circular dichroism (CD) spectra of WT HSP27 (red) and the P182L variant (blue). Both samples were prepared in 20 mM sodium phosphate, 100 mM

NaCl, 2 mM EDTA at pH 7.4. The total protein concentration was 20 μ M for both samples, and the spectra were collected at 20 °C. The P182L sample was purified from inclusion bodies under denaturing conditions and subsequently refolded. The WT sample was purified from the soluble fraction, and, therefore, represents a control for the correctly folded protein.

1.4 Please include the PDB code for the 3D structure in figure 1D.

We have now included the PDB code (4mjh) for the structure listed in Figure 1D.

1.5 The authors do not discuss the oxidation state of the full length proteins in the methods section. They could show a non-reducing gel for the (presumably reduced) protein they used for the full-length protein experiments. The C137S mutant - according to the materials and methods section - was used for gel filtration, but the protein used for NMR was oxidized, hence a different polypeptide.

We have now included details about the oxidation state of the protein in the Methods section. Full-length HSP27 and the P182L variant were purified from *E. coli* in the absence of reducing agents, and these proteins were therefore oxidized. There is only a single cysteine residue in the amino-acid sequence of HSP27, which is C137 that forms an inter-molecular disulfide bond. For NMR measurements, we used the wild-type form of cHSP27 with the same disulfide bond intact. The C137 disulfide prevents exchange between the monomer and dimer, which otherwise degrades the quality of the NMR spectra due to exchange on the millisecond-timescale, even when the monomer is only populated to ca 1% (*cf.* Alderson *et al.* (2019) *Nat. Commun.*, PMID: 30842409). Therefore, our *in vitro* experiments were performed on the oxidized form, except in Figure 2A, where we used the C137S variant of cHSP27 to demonstrate the considerable shift in mass when comparing cHSP27 to the full-length protein to the P182L variant of the full-length protein.

Of note, we and others have previously shown that reduction of the C137 disulfide bond does not significantly change the structure of the α -crystallin domain dimer (Rajagopal *et al.* (2015) *J. Biomol. NMR*, PMID: 26243512; Clouser *et al.* (2017) *Cell Stress Chaperones*, PMID: 28332148; Alderson *et al.* (2019) *Nat. Commun.*, PMID: 30842409). For example, the crystal structure of oxidized cHSP27 (unpublished) is essentially identical to the reduced form (PDB 4mjh), with a backbone RMSD of < 0.5 Å. Likewise, we previously demonstrated that the C137S mutation mimics the reduced form of the WT protein (Alderson *et al.* (2019) *Nat. Commun.*, PMID: 30842409).

1.6 Substitution of proline at position 182 with an isoleucine results in a dramatically reduced affinity of CTD with respect to ACD. How does this result explain the change of the oligomeric size of the protein?

We thank the reviewer for this insightful question. Indeed, we found that the Pro-to-Leucine mutation at Pro182 lowers the binding affinity of the C-terminal domain (CTD) for the α -crystallin domain (ACD). We have added a sentence in the Discussion on p. 22 to help clarify how we think this lowered affinity contributes to the increase in oligomer size. Namely, we think that two mechanisms could be responsible: (1) the lower affinity of the IxI/V motif in the P182L mutant may cause subunits to exchange more slowly (Baldwin AJ, *et al.* (2011) *J. Mol. Biol.*, PMID: 21839749), or (2) the P182L mutation could facilitate inter-oligomer interactions involving the NTD of one oligomer and the available IxI/V binding site of another. This is based on work from a recent paper from the Klevit group (Clouser AF, *et al.* (2019) *eLife*, PMID: 31573509), which

demonstrated that the ⁶VPF⁸ motif near the N-terminal region of the NTD can bind to the same site as the IxI/V motif, albeit with significantly lower affinity. In the case of the latter, this inter-oligomer interaction would be diminished in the WT form of the protein, since the WT IxI/V motif present within a given oligomer would more readily occupy the IxI/V binding site.

In the course of the revision, we also derived induced pluripotent stem cells (iPSCs) from a CMT patient harboring the heterozygous P182L mutation and differentiated these into spinal motor neurons. We also observed that immunostained HSP27 forms large, aggregate-like clusters in the iPSCs, which indicates that P182L becomes large within motor neurons. These results both support and extend our *in vitro* results and those obtained in HeLa cells.

From an updated version of Figure 3: (F) Immunostaining of HSP27 (white) in motor neurons differentiated from CMT patient-derived iPSCs that harbor the heterozygous P182L mutation reveals large, cytoplasmic aggregates (center and right panels). The control motor neurons that contain WT HSP27 do not show evidence of any aggregates. The nucleus is stained with Hoechst (blue). Scale bar, 10 μ m.

1.7 *In figure 2B there is a very limited amount of particles present in the negative stain micrograph. Please show a larger field/a more densely populated area.*

See also response to 3.2. We thank the reviewer for pointing this out, and on reflection have decided to remove the negative-stain EM images from the manuscript due to inevitable ambiguities from such analyses.

1.8 *Figure 2 C: The result shown in the "Expansion" panels is astonishing considering the "Confocal" image right above. At this low magnification (and resolution) both cell lines look quite similar. It would be helpful and interesting to include a normal confocal scan of the pre-expansion sample at a higher magnification. This would allow comparison of the expansion microscopy data (higher resolution of course) to the lower-resolution fluorescence distribution. The drastic difference observed after expansion should also be evident at normal confocal resolution.*

The improved resolution with expansion microscopy is indeed spectacular and allows us to observe the striking difference in cytoplasmic distribution between wild-type and P182L mutant HSP27 forms. This prominent change in fluorescence distribution is, however, not at all clear when using conventional diffraction-limited confocal microscopy, even at the highest magnification (63x 1.4 NA objective). In the Timmerman lab, we have been performing (conventional) microscopy on numerous P182L samples for many years, and although it was always evident that the P182L mutation leads to more cells with large cytoplasmic aggregates

(see also Supplementary Figure 2 of manuscript), we have never observed these alterations in the fluorescence distribution of cells without large aggregates. In fact, we were very surprised by seeing the drastic increase in resolution and the astonishing difference when imaging these cells with expansion microscopy.

Furthermore, on other sample types, we also observed great resolution increases when using expansion microscopy. Although the theoretical gain in resolution with this technique is 'only' 4.3 fold, an important factor is that this improvement not only occurs laterally (xy resolution, going from approximately 250 nm to 60 nm), but also increases in the axial dimension (z resolution) because the gel expands isotropically in all dimensions. In practice, this drastically increases the ability to resolve small structures such as the fluorescence spots of HSP27, by increasing lateral resolution and greatly improving contrast through 'thinner' optical sectioning. We have added an additional high resolution confocal (63x 1.4 NA objective) zoom to Figure 2, which will indeed be helpful to make the comparison and to show more explicitly that these differences are not clearly observable with diffraction-limited microscopy. Please find this addition below for your convenience:

1.9 *In figure 2D, the spot density is lower according to the data shown. However, the expression level of HSP27P182L seems also lower than that of HSP27 according to figure 5C. The authors should discuss this. Also, the fluorescence of each spot seems to roughly double on average (cf. Fig 2E). Combined, this could explain the reduced number of foci.*

For microscopy experiments, wild-type and mutant HSP27 were transiently transfected into HeLa cells, which implies that there will always be some unavoidable inter-cell variability in expression. However, on average, the expression levels are similar, as also shown in the western blot included in our response to comment 2.1 below. The range of inter-cell variability is relatively low under our transfection conditions and can be evaluated in Fig. 2C within the upper confocal panel or in Appendix Figure S2. In addition, cells with moderate expression levels are picked for imaging (e.g. in Supplementary Figure 2 where we avoided the five aggregate-containing P182L cells and the four WT cells with very high expression levels). As a result, wild-type and P182L HSP27 cells on average have the same level of expression. Extracting HSP27 expression directly from the measured spot intensities in expansion microscopy samples, as the referee suggests, also reflects a similar expression levels for HSP27 wild-type and P182L, as is shown in the three charts below. Multiplying the average spot intensity (which is approximately doubled in P182L) with the number of spots per area (which is approximately halved in P182L)

gives the same integrated spot intensity for both genotypes. We have now explicitly added this information in the Results section, and these graphs are included in a modified Appendix Figure S2.

To allow the referees to also visually evaluate the inter-cell variation in spot distribution and intensity we provide below a figure with extra examples of expansion microscopy images, similar to Figure 2C: P182L and wild-type HSP27 in the upper and lower row, respectively.

As an alternative estimate of HSP27 expression, we also measured the raw (mean) intensity on the complete images (using the identical sets of images that were analyzed for spot extraction and measurement). This type of measurement is, however, a bit complicated by the higher levels of diffuse background signal in wild-type HSP27, presumably originating from a higher number of out-of-focus assemblies or diffuse signals. Indeed, the graphs below show that raw image intensities are somewhat lower (t-test p-value < 0.01) for P182L compared to wild-type HSP27 when no background subtraction is applied, but have similar values for both genotypes (statistically non-significant, p-value 0.3) when measurements are performed after intensity background subtraction (subtraction of Gaussian Blur image with 500 pixel radius), as is considered best practice when performing intensity measurement. These graphs are included in a modified Appendix Figure S2.

In sum, all these results support the main conclusion that P182L shifts the assembly of HSP27 towards larger oligomers, which, at the same expression level, also results in fewer detectable assemblies.

1.10 *Figures 3/4: The studies on the affinities of the peptides are limited to the isolated ACD and NMR. If the concluded mechanism is right, the wildtype and mutant peptides should be able to influence the oligomer distribution of wt HSP27 (and vice versa influences should be observed for the P182L peptide). The respective titration experiments with full length Hsp27 are of high interest and needed to test the claimed mechanism.*

We thank the reviewer for this comment. The reviewer suggests experiments in which the peptides that mimic the C-terminal region are added to full-length HSP27 to see how they influence oligomerization. Intuitively, attractive though these experiments are, and we have attempted such experiments in the past, they have proven not to be feasible in practice. We note that other groups have attempted such experiments, and they were also unable to detect peptide binding to WT HSP27 (Freilich R, *et al.* (2018) *Nat. Commun.*, PMID: 30385828), even with the artificial peptide containing the F185H mutation that serves to enhance affinity.

In addition, neither full-length WT or P182L HSP27 are amenable to conventional solution-state NMR studies due to its large molecular mass that renders most NMR signals undetectable. For example, previous NMR studies were only able to observe signals from the final 20 residues in the disordered C-terminal region, E185-K205 (Carver JA, *et al.* (1995) *FEBS Lett.*, PMID: 7649277; Alderson TR *et al.* (2017) *Cell Stress Chaperones*, PMID: 28547731). The IxI/V motif was not detected in these NMR spectra. The Klevit group has recently succeeded in obtaining full-length HSP27 samples for NMR analyses (Clouser AF, *et al.* (2019) *eLife*, PMID: 31573509), but these constructs necessitated mutations in the N-terminal domain and the IxI/V motif, a strategy which cannot be implemented in our work given its focus on the IxI/V motif. Together, these issues make it challenging to study full-length HSP27 by NMR in the presence of peptides

Instead, given the above limitations, we prefer to focus on our new hetero-oligomerization and co-immunoprecipitation data involving native interactors. These data, included in modified forms of Figure 5 and Appendix Figure S3, show the P182L variant hetero-oligomerizes with WT HSP27, and that P182L binds with higher affinity to five different IxI/V containing proteins. For

BAG3, we also showed that the BAG3 IxI/V motif is responsible for this interaction. This indicates that the P182L variant occupies its own $\beta 4/\beta 8$ groove less frequently than the WT protein, thereby enabling binding to other IxI/V-containing proteins.

1.11 *Can the results obtained for the ACD experiments be directly be transferred to full-length protein? The NMR data were recorded using the oxidized ACD whereas the other experiments were performed with full-length (presumably) reduced protein.*

As the reviewer has pointed out, our NMR studies were performed on the oxidized form of the ACD whereas the experiments in cells used the reduced protein. However, the oxidation state of HSP27 does not affect binding of the IxI/V motif, as the disulfide bond is approximately 20 Å from the peptide binding site, and the disulfide does not affect the conformation of the IxI/V binding site in the $\beta 4/\beta 8$ groove. The solution structure of oxidized ACD is nearly identical to the crystal structure of the reduced ACD (Rajagopal *et al.* (2015) *J. Biomol. NMR*, PMID: 26243512), and there are no changes to the chemical shifts of the $\beta 4/\beta 8$ groove upon formation or reduction of the disulfide bond (Alderson TR, *et al.* (2019) *Nat. Commun.*, PMID: 30842409). In addition, a crystal structure of the similar protein α B-crystallin bound to a peptide containing its IxI/V motif (PDB: 4m5s) is identical to that of the E117C mutant, which contained a disulfide-linked dimer (PDB: 4m5t) (Hochberg GKA, *et al.* (2014) *Proc. Natl. Acad. Sci. U S A*, PMID: 24711386). Notably, E117 in α B-crystallin is in the same position as C137 in HSP27. Finally, unpublished results from our laboratory indicate that reduction of the disulfide bond in cHSP27 has no impact on the binding affinity of the IxI/V-containing peptide. We therefore conclude that the oxidation state of HSP27 does not affect IxI/V binding.

Regarding the direct transfer of our results from the ACD to the full-length protein, we direct the reviewer to our response to comment 1.10 above. In brief, the full-length protein is too large and polydisperse to be studied by solution-state NMR methods. Preliminary results from methyl-TROSY indicate that many resonances are absent from the spectrum, which suggests the oligomers may be too heterogeneous or undergoing widespread conformational exchange that precludes characterization.

1.12 *In the NMR experiments to determine the affinity of the C-term domain peptides to HSP27 the authors see differences in affinity of about one order of magnitude between the assays. It seems an orthogonal method is needed in this context.*

The weak binding affinity of the IxI/V motif for the ACD renders it challenging to detect by other biophysical methods. For example, in a recent paper from the Gestwicki lab (Freilich R, *et al.* (2018) *Nat. Commun.*, PMID: 30385828), they were unable to detect binding of the IxI/V motif-bearing peptide to cHSP27 by using isothermal titration calorimetry (ITC). Instead, the authors had to modify the sequence of the peptide, replacing a Phe with His, in order to elevate the binding affinity to detect a more robust ITC signal. This means that the attenuated binding affinity of the P182L peptide would surely be undetectable by ITC.

By contrast, NMR is an ideal method to measure weak binding affinities, as it can accurately detect interactions with affinities ranging in the μ M-to-M range. For instance, a 1.0 M binding affinity was recently reported in a solution-state NMR study from the Shimada group (Toyama Y, *et al.* (2018) *Proc. Natl. Acad. Sci. U S A*, PMID: 29581303).

1.13 The authors don't list the residues that were used to calculate the K_d using the CSP data. They however provide this information in the methods section for the CPMG data. The residues should also be given for CSP fitting.

We apologise for this omission, and we have now listed the residues used for fitting the CSPs in the Methods section on p. 36.

1.14 The authors did not assess the effects of the P182S mutant which also causes Charcot-Marie-Tooth disease. Ser seems to be the second most highly populated residue in disordered regions after Pro. It would be interesting to see how this mutation behaves in some of the assays.

The P182S variant of HSP27 was previously studied *in vitro* by the Gusev group (Chalova AS, *et al.* (2014) *Biochim. Biophys. Acta*, PMID: 25220807). This variant was found to form very large oligomers in solution by both size exclusion chromatography (SEC) and dynamic light scattering (DLS). This suggests that the P182S globally behaves similarly to P182L, and that the P182S IxI/V motif might also bind with lower affinity to the ACD. We have added a sentence in the Discussion on p. 23 to incorporate this into the manuscript.

1.15 The authors argue that for the wild type Hsp27 only resonances of the disordered CTR were observed. Other studies have shown, however, that it is also feasible to observe ordered regions of small heat shock proteins using MAS solid-state NMR (e.g. Jehle *et al.*, 2010). There, an ordered CTD was observed. It would be interesting to compare solid-state NMR spectra of wt and P182L to better understand the structure and dynamics of the CTD-bound state.

We thank the reviewer for the interesting comment, but we think that MAS ssNMR studies on HSP27 are beyond the scope of our current study. However, MAS ssNMR would be of interest for future investigations aimed at untangling the oligomeric mechanisms of HSP27.

We also note that, in the previous MAS solid-state NMR study that was referenced, the authors observed an ordered CTD for the protein α B-crystallin only at low temperatures (e.g. 0 °C). However, at room temperature and above, both solid-state and solution-state NMR spectra on methyl groups show that the CTD is flexible and predominantly unbound (e.g. Baldwin AJ *et al.* (2011) *J. Am. Chem. Soc.*, PMID: 21650202). This is further exemplified by solution-state ^1H - ^{15}N NMR spectra of full-length HSP27 and α B-crystallin, which reveal intense resonances for the disordered C-terminal region, indicating that these residues are not stably bound, but are free to tumble very rapidly in solution. This is evident in a recent publication (Mainz A *et al.* (2015) *Nat. Struct. Mol. Biol.*, PMID: 26458046), wherein the disordered C-terminal region of α B-crystallin was observed in solution at room temperature (Figure 2). Multiple sets of resonances were observed in the 2D ^1H - ^{15}N HSQC spectrum for certain residues in this region, which is consistent with slow cis-trans isomerization about the many Xxx-Pro bonds in the C-terminal region (*cf.* Alderson TR, *et al.* (2017) *Cell Stress Chaperones*, PMID: 28547731). If the CTR were to exist in a rigid, bound conformation, the resonances would not be detected in amide HSQC-based NMR experiments due to the very large size of the oligomers.

1.16 V111 is shown to be hydrogen-bonded to the CTR peptide in figure 1E. According to figure 3A, it is an intermediate exchange residue. However, despite its potential relevance for CTR binding, it is not shown or discussed in the CPMG experiments. Could the authors comment on this?

We thank the reviewer for this observation. Indeed, V111 forms a hydrogen bond to T184 in the bound peptide. However, this hydrogen bond involves the carbonyl group of V111 acting as the acceptor for the amide proton of T184 (V111^{CO}-T184^{HN}). Our CPMG relaxation dispersion experiments were performed on the ¹⁵N nucleus, which means they are sensitive to chemical shift fluctuations involving the amide nitrogen. In the CPMG relaxation dispersion experiment, we observe only a small ¹⁵N dispersion for V111 ($R_{ex} < 2 \text{ s}^{-1}$), indicative of a small change in ¹⁵N chemical shift (e.g. $< 0.5 \text{ ppm}$). For this reason, we did not include in the fitting, as is common practice for relaxation dispersion measurements.

Since the small dispersion for V111 signifies a small ¹⁵N chemical shift change, the intermediate exchange that is observed during the ¹H-¹⁵N HSQC peptide titration must be caused by a relatively large ¹H chemical shift change upon binding. This situation is outlined in a recent publication from the Christodoulou group (Waudby CA, *et al.* (2019) *Sci. Reports*; Waudby CA, *et al.* (2020) *J. Biomol. NMR*). Indeed, in our peptide titration data, we observe a new resonance that appears *ca.* 0.3 ppm downfield in ¹H from the original V111 resonance. This would correspond to a $\Delta\omega$ of *ca.* 1130 rad s^{-1} at 600 MHz ¹H Larmor frequency, which would place the V111 resonance in the intermediate exchange regime ($\Delta\omega \sim k_{ex}$) assuming a k_{ex} value of 775 s^{-1} in the presence of *ca.* 2% bound cHSP27. We note that k_{ex} depends on the amount of added peptide since $k_{ex} = k_{off} + k_{on}[\text{peptide}]_{free}$. While we do not have resonance assignments for the peptide-bound form of cHSP27, we can estimate that the new V111 resonance is moving downfield in the ¹H dimension based on exchange-induced shifts to the ground-state resonance of V111 at early stages in the titration. This is included in a figure below that has been added for the reviewer's convenience but is not included in the revised manuscript.

We have added text to the Results section on p. 15 of the revised manuscript to discuss V111.

Figure included for comment 1.16 from Reviewer #1: Intermediate exchange for the V111 resonance during the peptide titration. (Left) 2D ¹H-¹⁵N HSQC spectra of ¹⁵N-labeled cHSP27 in the absence (black) and presence of an increasing amount of C-terminal peptide (purple-blue-green-orange-red). The resonance from free V111 is boxed at 8.53/124.71 ppm. The spectra corresponding to peptide:cHSP27 ratios of 2:1, 5:1, and 10:1 are contoured at a lower level than 0:1, 0.25, 0.5:1, and 1:1. This is to indicate the appearance of a new resonance that may correspond to the bound form of V111,

which is boxed at 8.83/124.94 ppm. The corresponding chemical shift changes between free and putatively bound V111 are indicated above the arrow. In the lower-left corner, these values are converted into radians-per-second (rad s^{-1}) to enable comparison to k_{ex} . Note that the k_{ex} value measured for ca. 2% of peptide-bound cHSP27 is $774 \pm 44 \text{ s}^{-1}$. The large difference in ^1H chemical shift ($\Delta\omega_{\text{H}} = 1131 \text{ rad s}^{-1}$; $k_{\text{ex}} < \Delta\omega_{\text{H}}$) would place V111 in the slow-intermediate exchange regime relative to ^1H and in the fast exchange regime relative to ^{15}N ($\Delta\omega_{\text{N}} = 88 \text{ rad s}^{-1}$; $k_{\text{ex}} \gg \Delta\omega_{\text{N}}$). **(Right)** Zoomed-in region showing the small, downfield ^1H shift for V11 in the presence of a small amount of peptide. The 0.25:1 (purple) and 0.5:1 (blue) peptide:cHSP27 samples show a progressive downfield shift in the ^1H dimension. The ^1H frequency of each peak is indicated with a vertical line. This is consistent with the bound form of V111 resonating downfield in ^1H relative to the free state. Our ^{15}N CPMG relaxation dispersion measurements indicate that the ^{15}N chemical shift change between free and bound V111 is small ($< 0.5 \text{ ppm}$). Since the resonance for V111 broadens during the titration, the exchange broadening must arise from a large change in the ^1H resonance frequency. The new resonance boxed in the left panel is consistent with these changes.

1.17 *The authors claim that at least 22 proteins should be affected in their interaction with P182L. Besides mentioning previous studies on PCBP1 they only assess one of the claimed interaction partner, BAG3. Data on additional proteins seem required to support a general model.*

We thank the reviewer for this suggestion, and we agree that a broader study of the P182L partners would strengthen the manuscript. To this end, we reanalysed one of our interactomics datasets (first published in Haidar et al. (2019) *Autophagy*, PMID: 30669930) and studied the P182L interactors in more detail. We observed that 8 out of the 11 significantly enriched interactors for the P182L variant contained one or more I/V-x-I/V motifs in their sequence (see figure below). We performed new co-immunoprecipitation assays, and confirmed that these interactors bind more readily to the P182L protein. Note that we could not test all the 8 hits because there were no antibodies available for some (KIAA1217 and FAM96B) or their expression was very low (CHORDC1 and ARPC5).

Most interactions were only detectable with the P182L but not the WT protein, presumably caused by the transient nature of the chaperone-client complexes that prevents the interaction partner from being efficiently captured with the WT protein. Only two well-known stable interactors (HSPB5 and BAG3) were also detected with the WT pull-down. Interestingly, also these two proteins were bound more to the P182L mutant.

This increased binding of I/V-x-I/V containing proteins consolidates our model further since our NMR-data showed that due to the reduced binding of the P182L peptide, the $\beta 4/\beta 8$ groove is more accessible for other proteins. The strong enrichment of I/V-x-I/V containing interactors thus confirms that the P182L has its hydrophobic binding pocket more available for these proteins, hence their enrichment in the dataset.

Finally, we also made use of an established mutation, S155Q (e.g. Baughmann HER, *et al.* (2020) *Proc. Natl. Acad. Sci. U S A*), that disrupts IxI/V binding to the $\beta 4/\beta 8$ groove. The S155Q/P182L double mutant no longer pulls down UBXN1, FAM83D, or PCBP1, which indicates these interactions rely upon binding of an IxI/V motif to the $\beta 4/\beta 8$ groove of HSP27. The S155Q/P182L remains able to pull down BAG3 and HSPB5, albeit to a lesser extent than P182L, which could indicate additional binding sites or interaction mechanisms beyond the $\beta 4/\beta 8$ groove.

We think that this additional set of experiments, and the data they provided, have greatly benefitted the manuscript, and thank the reviewer for the suggestion. For more details on hetero-oligomerization, please see our reply to comment 1.21 below.

Updated Figure 5. The P182L variant binds with enhanced affinity to IxI/V-containing proteins. (A) Percentage of [V/I]X[V/I] tripeptides among (*top*) the total number of all tripeptides (% tripeptides) or (*middle*) the total number of [V/I]X[V/I] tripeptides in the structured (black) or disordered regions (red) ([V/I]X[V/I] %) shown as a function of the X residue type. (*Bottom*) The difference between [V/I]X[V/I] % for disordered and structured regions shows the enrichment (positive) or depletion (negative) of specific [V/I]X[V/I] motifs in disordered regions. The grey bars indicate the [V/I]L[V/I] and [V/I]P[V/I] motifs studied

in this work, *i.e.* ILV and IPV. **(B)** Volcano plot representing the interactors that are enriched for the P182L mutant versus GFP as a negative control, obtained with affinity-enrichment mass spectrometry. Proteins that were co-immunoprecipitated significantly more ($p < 0.05$) with the P182L variant are shown in the upper right quadrant. Interactors with one or more [I/V]-x-[I/V] motifs are displayed in red while other significantly enriched interactors are displayed in black. **(C)** Co-immunoprecipitation using anti-V5 beads from HeLa cells stably overexpressing V5-epitope-tagged HSP27 (WT or P182L mutant) and transiently transfected with BAG3-eGFP. Western blots for anti-GFP (BAG3), V5 (HSPB1), and β -actin. **(D)** The relative amount of immunoprecipitation in panel C was quantified and represented relative to the amount of protein pulled-down with the P182L. Three independent replicates were used for quantitation. Structural schematics of WT and P182L HSP27 are shown to the right with Pro182 and Leu182 indicated. P182L has a lower affinity for its IxI/V motif (Self IxI/V binding). **(E)** Co-immunoprecipitation using anti-GFP beads from HeLa cells stably overexpressing V5-epitope-tagged HSP27 (WT or P182L mutant) and transiently transfected with BAG3-eGFP (WT or IPV-mutant). Western blots for anti-GFP (BAG3), V5 (HSPB1), and α -tubulin. NS indicates a non-specific band. The locations of the IPV-to-GPG mutations in BAG3 are depicted below. **(F)** The same as panel C, except that V5 epitope-tagged P182L or S155Q/P182L HSP27 were immunoprecipitated. The mutation of S155 (red spheres) to Gln155 (purple spheres) disrupts binding of the IxI/V motif to the $\beta 4/\beta 8$ groove. The double mutant S155Q/P182L no longer pulls down UBXM1, FAM83D, or PCBP1. HSPB5 and BAG3 are likely pulled down due to hetero-oligomerization with endogenous HSP27 and HSPB5. Structural schematics of WT and S155Q HSP27 are shown to the right with Ser155 and Gln155 indicated. S155Q disrupts the IxI/V binding site (All IxI/V binding).

1.18 *Based on figure 5C, the authors claim stronger binding of BAG3 to HSP27-P182L in co-IP experiments. Based on ECL luminescence, more BAG3 seems to be present in the Co-IPs with HSP27-P182L. But it seems that more BAG3 than HSP27-P182L is present in the complex. How can this be explained with the proposed model assuming a single binding site for IXI on HSP27?*

From the images presented that were presented in Figure 5C (note that this figure has been updated in the revised manuscript, and it now corresponds to Figure 5E), it cannot be concluded whether there is more (or less) BAG3 per HSPB1 in the complex. Images for HSPB1 (anti-V5) and BAG3 (anti-GFP) are taken separately, which subjects each detection to a different exposure time and contrast setting. In addition, the affinity of anti-GFP and anti-V5 antibodies to their respective epitope is not the same, so signal detection of HSPB1 (V5) and BAG3 (GFP) occurs on a different scale. Western Blot intensities can only be compared using the same antibody on the same membrane.

We have clarified this point in the legend of Figure 5 on p. 20 to note that BAG3 and HSP27 are detected with two different antibodies.

1.19 *Figure 5D: Why is the control switched from actin to tubulin here?*

As a control for ourselves, we also probed the membrane of the Figure 5D for HSPB8 (as a positive control) (note this is now Figure 5E in the updated version of the manuscript). However, this antibody gives a few nonspecific bands, one of which is near the β -Actin band. To avoid confusion, we therefore decided to switch the loading control from β -Actin to α -Tubulin, which is located about 10 kDa higher on the membrane. This avoids any potential overlap with the nonspecific bands that are detected by the HSPB8 antibody.

1.20 Why does BAG3 not pull HSP27 in this experiment? Here and in figure 5C, quantification of the ratio of the bands seems required.

BAG3 does pull down some WT HSP27, but the reviewer is correct in stating that it is not visible in the figure presented in the manuscript. The settings used for the HSP27 detection were tailored to avoid overexposure of the input fractions of HSP27 (so the exposure time was kept to a minimum). However, for the reviewer's inspection, we have increased the contrast of the image used in Figure 5D (see below). This shows a weak band for the WT protein, which disappears when both IPV-motifs of BAG3 are mutated.

Figure as presented in manuscript

Same image plus higher contrast image

1.21 The authors postulate that HSP27 will interact with HSP27P182L in vivo sequestering 'functional' Hsp27 into 'dysfunctional' Hsp27P182L complexes, hence explaining the dominance of the mutant allele. The authors do have both proteins at hand. So hetero-oligomer-formation could be assessed.

We thank the reviewer for this helpful comment, and we have now tested if the WT and P182L forms of HSP27 assemble into hetero-oligomers. Indeed, using gel filtration chromatography, we see that the wild-type form assimilates into the larger P182L oligomers. This was observed

by a depletion in the WT elution peak after mixing equimolar ratios of WT and P182L HSP27, which were equilibrated at room temperature overnight before injection onto the column.

Figure included for comment 1.21 from Reviewer #1: Hetero-oligomerization of WT and P182L HSP27 oligomers. Equimolar ratios of WT and P182L HSP27 (250 μ g each) were mixed and equilibrated overnight at room temperature in 20 mM sodium phosphate, 100 mM NaCl, 2 mM EDTA at pH 7. The sample was then injected onto a Superose 6 10/300 column (black). The same amount of WT HSP27 (20 μ g) was injected onto the column (red). The depletion of the WT HSP27 elution peak at ca. 13.8 mL in the mixture indicates the formation of hetero-oligomers between P182L and WT HSP27.

Moreover, in our cell-based experiments, we see that the increased insolubilization propensity of P182L is accompanied by an increased insoluble fraction of WT, endogenous HSP27. When we pull down epitope-tagged P182L, we also see that WT, endogenous HSP27 is pelleted, which indicates the two forms are stably interacting, most likely in a hetero-oligomer based on the SEC-MALS data above.

Figure included for comment 1.21 from Reviewer #1: The P182L variant hetero-oligomerizes with WT HSP27 in HeLa cells. (A) *top*: Co-IP with anti-V5 beads used to pull down the overexpressed forms of HSP27. The membranes were then probed with an anti-HSP27 antibody. The P182L variant pulls down the endogenous WT. Additionally, in the input fraction it is evident that the insolubilization of P182L promotes insolubilization of the endogenous WT protein. *Bottom*: Soluble versus insoluble fractions of HeLa cell lysate. The insolubilization of the P182L variant is accompanied by an increased insolubilization of endogenous WT HSP27. Antibodies to both HSP27 (HSPB1) and the V5-epitope (HSPB1-V5) were used to ensure visualization of endogenous HSP27 as well as the epitope-tagged form.

Because WT/mutant hetero-oligomers were reported previously for the P182S variant by the Gusev group (Chalova AS, *et al.* (2014) *Biochim. Biophys. Acta*, PMID: 25220807), and many other disease-causing variants of HSP27 are known to hetero-oligomerize with the WT protein. Most notably, the hetero-oligomerization between P182L and WT HSP27 was specifically shown via co-IP in an earlier paper (Almeida-Souza L, *et al.* (2010) *J. Biol. Chem.*, PMID: 20178975). Additionally, evidence of hetero-oligomerization between HSP20 (i.e. HSPB6) and HSP27 has been reported (Weeks SD, *et al.* (2018) *Sci. Rep.*), as well as hetero-oligomers between HSP27 and other sHSPs (Mymrikov EV, *et al.* (2020) *J. Bio. Chem.*). It is therefore not surprising that WT and P182L HSP27 would readily hetero-oligomerize.

We have added a new Supporting Figure (Appendix Figure S3) that contains the results of our hetero-oligomerization experiments. In addition, we have added text to the Results (p. 21) and Discussion (p. 23) sections of the manuscript.

1.22 Concerning the bioinformatic analysis the authors need to clarify their view on the IXI binding site in the ACD-beta4-beta8 groove. Is this a "substrate" binding site? Or do the authors envision this site as a specific binding site for interaction partners regulating the oligomer distribution of HSP27? Overall, this genome-wide analysis is interesting. But according to figure S7 C,D, VDV and VPV look much more interesting than IPV (largest deviation. The authors do not discuss these. Adding some statistical analysis would enable the reader to judge the relevance of the highlighted [I/V]P[I/V] enrichment independently. It would also allow the assessment of other potential enrichments like [V/I]S[V/I] (cf. Figs 5A, S7E), VDV, VPV (cf. Fig S7D) or depletions as central Phe, Trp, Tyr (cf. Fig S7E) or VRV, IGV (cf. Fig S7D). The bioinformatics section needs to be streamlined and reworked.

We thank the reviewer for this helpful comment. We have now performed a statistical analysis on the bioinformatics data shown in Appendix Figure S7. Namely, we calculated χ^2 values to assess the statistical significance of the observed frequencies of [V/I]-x-[V/I] motifs relative to expected values. In this analysis, the null hypothesis is that the observed frequencies of [V/I]-x-[V/I] motifs are accurately described by the underlying amino acid frequencies of [V/I] and x, where $x = \{A, C, D, \dots, V, W, Y\}$. For each of the four motifs (IxI, IxV, VxI, and VxV), we find that the χ^2 values are >160 or >400 for disordered and structured regions, respectively. Given that there are $v = 19$ degrees of freedom for each motif, a χ^2 value would need to be lower than 30.144 to accept the null hypothesis, *i.e.* to obtain a p value > 0.05. By contrast, our χ^2 values are all significantly larger than 30.144, which means that the observed frequencies cannot be explained simply by amino acid frequencies – therefore, the null hypothesis is rejected for each motif ($p < 0.001$). When we sum up the total χ^2 values for each of the four motifs, we also obtain a p value that indicates a high level of significance ($p < 0.001$), indicating that amino acid frequencies alone do not describe the observed numbers of motifs.

These data have been included in Appendix Figure S8 and are shown below for convenience.

Updated panel from Appendix Figure S7: (D) The observed number of IxI, IxV, VxI, and VxV motifs in structured regions of the proteome compared against the expected number of such motifs calculated using the frequency of amino acids in structured regions. The same plot is included for disordered regions using the frequencies of amino acids in disordered regions. For the structured and disordered plots shown here, the R^2 values from a linear regression (not plotted) are respectively 0.89 and 0.86 with slopes of 0.83 and 1.12. Bottom panel: stacked bar graphs showing reduced χ^2 values for each residue type for each motif (IxI, IxV, VxI, VxV) for structured (left) or disordered (right) regions. The total χ^2 value, summed over each residue type ($j=1, 2, 3, \dots, 20$) and each motif ($i=1, 2, 3, 4$) is shown in the upper left region. The χ^2 values are highly significant ($p < 0.001$) for this system with $\nu = (20-1) \times 4 = 76$ total degrees of freedom, which indicates that the null hypothesis – that the frequency of [V/I]-x-[V/I] motifs is described solely by amino acid frequencies – is not correct. Rather, other factors, most likely some forms of evolutionary selection, have led to a non-random distribution of [V/I]-x-[V/I] motifs. For each residue type, statistical significance was inferred from the summed χ^2 value across the four motifs ($\nu = 3$). Residue types with an asterisk (*) indicate a p value < 0.05 , with most having p values < 0.001 . The “ns” symbol denotes not significant, *i.e.* p value > 0.05 .

Referee #2:

The goal of this work is to investigate the alterations in the structure-function relationship of the inherited mutation P182L in human heat shock protein Hsp27 by investigating the oligomeric size, distribution, and chaperone activity of the mutant protein. This is a mutation that is linked to Charcot-Marie-Tooth disease, a prevalent neuropathy. The mutation is located in a well-conserved IxI/V motif and it disrupts protein-protein interactions, leading to more interactions with other IxI/V motif proteins (Bag3) and increases sHsp oligomeric size. This study is of moderate-high significance as it suggests a general mechanism for how IxI/V motif proteins interact with one another, which is important for maintenance of many cellular protein-protein interaction networks.

This is a well-written, thorough study that demonstrates disruption of the IxI/V motif alters oligomerization of Hsp27 and increases interactions with other IxI/V motif-containing proteins. Interactions with Hsp27 ACD is decreased through a mechanism that likely involves an increase in conformational space sampling by the CTR.

Minor Concerns:

Results:

2.1 *Are there similar levels of protein expression of Hsp27 P182L and wildtype protein? Comparison of the oligomeric distributions is made in Figure 2, but are overall levels similar?*

We thank the reviewer for raising this concern, which is similar to comment 1.9 from Reviewer 1. Indeed, the expression levels of the WT and P182L protein are similar. We controlled for this in our experiments by quantifying the expression levels. Note, that the P182L is found partially in the insoluble fraction (in line with Ackerley *et al.* 2006, DOI: [10.1093/hmg/ddi452](https://doi.org/10.1093/hmg/ddi452)) and therefore its expression level may seem to be lower in experiments where only the soluble fraction is displayed, such as Figure 5E. However, when we quantified the sum of both the soluble and insoluble fractions in three independent experiments, the total amount of WT and P182L was shown to be equal with, if anything, a slight increase for P182L (see figure below).

We have added these data to Appendix Figure S2 and included them below for convenience. In addition, we updated text in the Results section on p. 9 to indicate the similar expression levels.

Updated panels from Appendix Figure S2: (C) Quantification of the total soluble and insoluble fractions of WT or P182L HSP27 in three independent experiments. The samples were obtained from HeLa cells transiently expressing V5 epitope-tagged HSP27 or the P182L variant. (D) A representative western blot underlying the data in panel C.

2.2 *Binding of the mutant peptide to the ACD is similar in location to the wildtype peptide. CD spectroscopy could confirm altered configuration of the mutant peptide.*

We thank the reviewer for this comment. To compare the conformations of the wild-type and P182L peptides, we recorded 1D ^1H NMR spectra of the two peptides in the same buffer used for NMR titration studies. The NMR spectra are indicative of disordered peptides and show that the P182L mutation does not cause the formation of secondary structure.

We have added the figure below to Appendix Figure S3 and updated the Results section on p. 13 to include new text.

Appendix Figure S3. Comparison of ^1H NMR spectra of the wild-type (red) and P182L (blue) peptides. One-dimensional ^1H NMR spectra were recorded on a Varian Inova spectrometer operating at a static magnetic field strength of 14.1 T with the temperature set to 25 °C. The samples were prepared by dissolving natural abundance peptides at a concentration of 2 mM in 30 mM sodium phosphate, 2 mM EDTA at pH 7 with 6% D_2O added to maintain lock. ^1H spectra were recorded with 64 scans, a maximum acquisition time (t_1) of 1 s, and an inter-scan delay of 1.5 s for a total duration of 2.7 minutes. Post-acquisition signal processing was performed with NMRpipe and was identical for both datasets. The spectral intensities are normalized to the most intense resonance line. For both peptides, the resonances cluster in the disordered region of the ^1H spectrum (8-8.5 ppm), indicative of the absence of appreciable secondary structure. Side-chain NH_2 resonances are observable at 7.4 ppm.

2.3 Does P183L bind to multiple other proteins in vivo aside from BAG3 that was expressed?

We thank the reviewer for this suggestion to broaden our study and evaluate other binding partners as well. Because Reviewer #1 also asked a similar question, we refer the reviewer to our response to comment 1.17 above for more detail.

Discussion:

2.4 Why would disrupting IxI/V motif (and self-association) lead to larger oligomers?

We thank the reviewer for this interesting comment. We have updated our Discussion section on p. 22 to incorporate some of the text that is included below.

First, we know that the IxI/V motif in mammalian sHSPs is not critical for self-association; ergo, one can assume that disrupting this motif or weakening the interaction will not prevent self-association. Indeed, this has been observed previously with mutants such as the IPV->GPG mutant in HSP27 (Freilich R *et al.* (2018) *Nat. Commun.*), which remains oligomeric. In fact, the GPG mutant appears significantly larger than the WT protein by negative-stain EM. Second, recent work from the Kleivit group has demonstrated that the NTD can bind to the IxI/V binding site ($\beta 4/\beta 8$ groove) in the ACD (Clouser A *et al.* (2019) *eLife*). They noted that the canonical $^{181}\text{IPV}^{183}$ binds to the $\beta 4/\beta 8$ groove with higher affinity than the $^6\text{VPF}^8$ motif that is present in the N-terminal region of the NTD. Therefore, in the wild-type sequence, the $^{181}\text{IPV}^{183}$ motif would 'out-compete' the $^6\text{VPF}^8$ motif for binding to the $\beta 4/\beta 8$ groove. However, in the case of our P182L variant, we know from our present manuscript that that the $^{181}\text{ILV}^{183}$ binds with

significantly weaker affinity to the $\beta 4/\beta 8$ groove. This could enable the $^6\text{VPF}^8$ motif to preferentially bind at this site, and if the $^6\text{VPF}^8$ motif was contributed from the NTD of a nearby oligomer, this could lead to severely dysregulated oligomerization.

Examining the role of the NTD remains outside the scope of our current manuscript that is focused on the P182L CMT-causing mutation; however, we hope to examine more details of the oligomerization mechanism(s) of HSP27 in a forthcoming manuscript. New and exciting results from the sHSP field point to a critical role of the NTD in oligomerization.

2.5 *Are the larger oligomers formed solely from Hsp27?*

Yes, our samples used for *in vitro* were purified to high levels of homogeneity and, therefore, are comprised of HSP27 alone. Our *in cellulo* studies cannot rule out that other components might be present in the large oligomers, but the qualitative agreement between our *in cellulo* and *in vitro* observations suggests that P182L forms larger oligomers than the WT sample. In our hetero-oligomerization experiments, for example, we find that WT, endogenous HSP27 can co-assemble with P182L. Therefore, the large P182L oligomers contain, at the very least, P182L and WT forms of HSP27.

We have added a sentence in the Discussion on p. 21 to incorporate this in the manuscript.

2.6 *Is the decrease in chaperone activity due to oligomerization disruption?*

This does not appear to be the case, as we tested two different concentrations of P182L (20 μM and 0.5 μM), and P182L was unable to prevent substrate aggregation in both cases. Because oligomerization is a concentration-dependent process, lowering the concentration of P182L monomers by a factor of 40 should impact the overall oligomeric state. The lack of activity at a concentration of 0.5 μM (where WT is highly active) suggests that the oligomerization state is not a reason for disrupted activity.

2.7 *Any benefit to enhancing IxI/V motif interactions?*

This is a very interesting comment, and we thank the reviewer for posing it. In this manuscript, we show that weakening the IxI/V-ACD interaction dysregulates oligomerization, leading to formation of very large oligomers. While we do not have any experimental data, at present, we speculate that strengthening the IxI/V-ACD interaction would dysregulate HSP27 oligomerization and subunit exchange kinetics. Given the universal nature of the IxI/V motif in sHSPs, we speculate that the affinity of the IxI/V motif for the ACD is tuned to optimally balance subunit exchange kinetics, oligomerization, and chaperone activity. Dysregulating the IxI/V binding affinity in any way will likely affect one or more of these aspects.

We have updated the Discussion on p. 23 in reference to this comment.

2.8 *Are the IxI/V motif interactions enhanced if more conformational space is sampled?*

Should the mutant $^{181}\text{ILV}^{183}$ motif sample more of conformational space in its unbound form, then the change in conformational entropy, ΔS_{conf} , upon binding to the structured ACD will be more negative than that of the WT $^{181}\text{IPV}^{183}$ motif. Assuming that the change in enthalpy, ΔH , upon binding is the same for the two IxI/V motifs discussed here, then the less favorable ΔS for

the P182L variant would result in a lower binding affinity. However, as the reviewer has pointed out, bimolecular interactions also depend on numerous other factors, including solvation, hydrogen bonding, and VDW interactions. Moreover, ΔS_{conf} is but one aspect of the entropy change, ΔS_{total} , which includes ΔS_{conf} as well as contributions from the solvent ($\Delta S_{\text{solvent}}$), rotational-translational aspects ($\Delta S_{\text{r-t}}$), and other sources.

We have updated the Results on p. 16 in reference to this comment.

Referee #3:

The manuscript by Alderson et. Al. deals with the HSP27 human chaperone. The authors address how a point mutation in the HSP27 C-terminal disordered region (P182L) i) increases oligomer formation and ii) reduces chaperone activity.

I have a number of remarks regarding the manuscript, these often refer to the interpretation of the data and should be discussed in detail. Currently, I have the feeling that the authors fail to clearly explain the "unexpected mechanistic explanation for the consequences of a mutation in the conserved IxI/V motif, which causes a severe form of CMT.", as they write in the discussion.

Major remarks:

3.1 *Does the P182L mutation stimulate aggregation? This appears to be the case, especially in figure 1G. Do the blue lines actually represent the aggregation of the substrate, or rather the aggregation of the substrate plus HSP27. In that case the more unstable P182L version of HSP27 obscures the data. Please comment and correct if needed, e.g. by using a significantly lower concentration of HSP27 compared to substrate.*

We thank the reviewer for this comment on the chaperone activity assays. Yes, the aggregation of substrate stimulates P182L aggregation. The P182L protein is stable on its own when incubated at 40 °C for 3 hours at a concentration of 20 µM (Appendix Figure S1), where only a very small increase in light scattering is observed. In Figure 1F within the main text, we used 0.5 µM P182L – which is a 40-fold lower concentration than Appendix Figure S1 – and the extent of self-aggregation of P182L would be negligible at this concentration over the 3-hour timescale. Therefore, P182L co-aggregates with substrate to yield the large increase in light scattering. We note that other sHSPs have been found to co-aggregate with substrates, including Hsp42 as identified by the Bukau group in a recent study (Ungelenk S, et al. (2016) *Nat. Commun.*).

We have updated our Results on p. 7 to note that P182L stimulates aggregation, similar to the reported results above.

3.2 *The negative stain EM images (Figure 2B) are unclear and can most likely not be used to draw conclusions regarding aggregation size. First: the shown views are very small and aggregates of various sizes can always be found on EM grids. A statistical analysis of a large number of particles is required to determine cluster sizes. Second, many proteins spontaneously aggregate on EM grids due to the harsh staining conditions. The particle size that is visible on the grids is thus not necessarily representative of particle size in solution. This can also be seen from a comparison of panels A and B in Figure 2. Panel A shows that the oligomer size of the P182L protein is a factor of ~50 larger than the WT protein. In the EM, the difference is much less.*

We note that the radius of hydration of the P182L, as determined by SEC-MALS, is only a factor of 4 larger than the WT sample (38 nm vs. 9 nm). This would imply that the diameter of the P182L particles detected by negative-stain EM should similarly be a factor of ca. 4 larger than the WT particles. Because volume scales with the cubed radius, the observation that the P182L particles are ca. 4-fold wider is consistent with a 50-fold increase in mass.

Based on this comment and that from Reviewer #1, we have decided to remove the negative-stain EM images from the manuscript because of their low quality.

3.3 *Figure 2C-E. This analysis can only hold in case the WT and the P182L proteins are expressed at the same levels. Please provide proof for that. The spot-density multiplied with the spot-intensity should e.g. be the same for both proteins. Page 8, Line 8: please reword drastic to two-fold.*

We thank the referee for pointing this out, and they are correct with this assumption. As referee #2 listed the same concern in comment 2.1, please allow us to refer to the above comment for a more detailed response. We have also changed the word “drastic” to “two-fold”.

We have added additional text in the results section stating that expression levels are similar for both wild-type and P182L HSP27. The new data can be found in an updated version of Appendix Figure S2.

3.4 *Figure 3B, C. Why are the residues in panel B not fitted in panel C? Please show the same residues in B and C and some of those also in A.*

We are grateful for the reviewer for noticing this. We have now added text to the legend of Figure 3 (p.14) to indicate that we only fit resonances to equation 1 that are in the fast exchange regime. Outside of this regime, equation 1 no longer applies.

This is because equation 1, which is used to fit CSPs to obtain a K_d , is only valid for resonances that are in the fast exchange regime ($k_{ex} > \Delta\omega$ in rad s^{-1}). The residues in Figure 3A were not used to fit CSPs to equation 1 because these residues are in intermediate exchange ($k_{ex} \sim \Delta\omega$), and therefore they become severely broadened during the titration. For example, residues V111, K114, and L157, which are shown in Figure 3A, have disappeared by the 2nd and 3rd titration points for the wild-type and P182L peptides, respectively. Their chemical shifts no longer depend on the K_d in a simple manner (i.e., equation 1 no longer applies).

3.5 *Page 10: "The P182L mutation thus lowers the binding affinity for the ACD by nearly one order of magnitude from ca. 125 μM to 1 mM". This sounds like a lot, but the difference is only slightly more than factor of 2 in DELTA-G. The error in the K_D for the P182L peptide appears very small as the endpoint of the titration is not defined. The authors should take uncertainties in the protein and peptide concentration into account when extracting the errors of the affinities.*

We appreciate the reviewer commenting on this. However, we note that the $\Delta\Delta G$ corresponds to ca. 5 kJ/mol, which is about the energy of a hydrogen bond. We have added text to the Results section on p. 12 to discuss this.

We note that the same batch of protein was used for both titration measurements, so any error in protein concentration is similarly propagated. The errors in peptide concentration were on the order of 1-2% based on the manufacturer's report on sample impurity, and therefore these are negligible in the error propagation.

3.6 *Figure 3E: the peptide is hardly visible.*

We thank the reviewer for noticing this. We have modified Figure 3E to include an arrow to indicate the position of the peptide.

3.7 *Fitted exchange rates and populations need to be provided for the CPMG experiments and should be discussed in detail also in the main text, now only the material and methods contain significant discussions (speculations).*

We thank the reviewer for this comment, and we now discuss the fitted parameters for the CPMG experiments in the Results text, which can be found on p. 14-15.

3.8 *Do the DELTAomega values that are fitted in the CPMG analysis agree with the delta omega values that are visible in the spectra or that are fitted in the kD titration experiments?*

We thank the reviewer for the insightful question. While we do not have resonance assignments for the peptide-bound form of the protein, the new resonances that appear in the spectrum of the peptide-bound form are consistent with our fitted ^{15}N $|\Delta\omega|$ values.

We note, however, that the cHSP27 sample with a 10-fold molar excess of peptide contains resonances that remain broadened at 25 °C, presumably due to intermediate exchange kinetics. Therefore, we collected HSQC spectra on this sample as a function of temperature to shift the exchange kinetics to faster timescales, in the hope that this would increase the spectral quality and reveal some of the bound-state resonances. Indeed, as can be seen in the Figure below, at higher temperatures we begin to see the bound state signals (boxed), which are otherwise undetectable at lower temperatures.

Figure included for comment 3.8 from Reviewer #3: NMR spectra of peptide-bound cHSP27 as a function of temperature. Shown here are 2D ^1H - ^{15}N HSQC spectra of ^{15}N -labeled cHSP27 in the presence of a 10-fold molar excess of WT peptide. Spectra were recorded at a static magnetic field strength of 14.1 T (^1H Larmor frequency of 14.1T) from 25 °C (blue) to 50 °C (red) in 5 °C increments. New peaks, which are boxed, appear at higher temperatures. These boxed peaks were otherwise exchange-broadened and undetectable or present at very low signal-to-noise ratios at lower

temperatures. The appearance of such peaks with increasing temperature suggests that the residual exchange in the peptide-bound form has moved from the intermediate to fast exchange regime.

We do not have resonance assignments for the peptide-bound form of cHSP27. So, we are limited to making informed guesses about the identity of the new bound-state resonances. We used the ^{15}N $|\Delta\omega|$ values obtained from CPMG relaxation dispersion and searched for new resonances in the vicinity of the ground state resonance \pm the ^{15}N $\Delta\omega$ value. In cases where the peptide titration revealed the direction of the ^{15}N chemical shift change (see the figure below, left), we assumed that the sign of the ^{15}N $|\Delta\omega|$ from CPMG would match the directly detected exchange-induced shifts. For example, with this procedure of informed guesses, we obtain good agreement between the fitted values of ^{15}N $|\Delta\omega|$ with those directly detected in the HSQC spectrum of the peptide-bound state (figure below, right). The RMSD between the measured values of ^{15}N $|\Delta\omega|$ and observed values of ^{15}N $|\text{CSP}|$ between the free and bound forms is 0.35 ppm.

Our interpretation here makes various assumptions, so we are only including this for the reviewer and not in the manuscript. We hope to obtain resonance assignments of the peptide-bound form to enable a future investigation aimed at elucidating the binding mechanism in more detail.

Figure #2 included for comment 3.8 from Reviewer #3: Exchange-induced shifts by peptide binding to cHSP27 and estimates of the agreement between ^{15}N $|\Delta\omega|$ values derived from CPMG RD and those in the HSQC spectra. (Left) 2D ^1H - ^{15}N HSQC spectra of ^{15}N -cHSP27 alone (purple) or in the presence of a sub-stoichiometric amount of WT peptide (cyan). The addition of more peptide causes these signals to disappear entirely, rendering it impossible to follow the signals during the titration. However, the small movement of resonances away from their original frequencies, indicated by the arrows, reveals the directions that the resonances are moving upon peptide binding. **(Right)** Since new resonances appear in the mostly-to-fully bound state (especially at high temperatures, see Figure above), we can therefore use the ^{15}N $|\Delta\omega|$ values obtained from CPMG RD in combination with these exchange-induced shifts to estimate the assignments of the bound-state resonances. The correlation of the measured ^{15}N $|\Delta\omega|$ values obtained from CPMG RD (y-axis) with those measured directly by comparing the HSQC spectra of free and peptide-bound cHSP27 yields a good agreement with an RMSD value of 0.35 ppm. The tentative assignments of the bound state resonances were obtained by using the exchange-induced shifts to identify the direction of the peak movement (relative to the free state). Then, the CPMG RD-derived ^{15}N $\Delta\omega$ values were used to search for a new, bound-state resonance that appeared with a ^{15}N frequency close to $\omega_{\text{free}} \pm \Delta\omega$, where ω_{free} refers to the ^{15}N frequency of the free state and the plus/minus symbol refers to the sign of the ^{15}N chemical shift change derived as above. In

cases where the bound-state resonance was observable only at higher temperatures, we used the observed temperature-dependence of the ^{15}N chemical shift to back-calculate its frequency at 25 °C.

3.9 Do the k_{on} and k_{off} rates agree with the rates that are reported by TITAN?

TITAN fits K_{d} values that agree with the CSP data for the WT and P182L peptide samples. Notably, the k_{off} for the P182L sample is in close agreement with the CPMG relaxation dispersion data (840 s^{-1} measured by CPMG vs. 774 s^{-1} fitted with TITAN). However, the k_{off} for the WT sample is roughly a factor of 4 faster as measured by CPMG compared to that fitted by TITAN. Since the CPMG experiment is an experimental probe of the off rate, we chose to interpret that value directly whereas the TITAN software fits lineshapes to the chemical shift titration data. The origin of the discrepancy could arise from the fact that the chemical shift titration data are more sensitive to an additional binding mode than the CPMG relaxation dispersion data.

We note that a previous NMR study on IxI/V binding to alphaB-crystallin found evidence for multiple conformations in the peptide-bound form (Delbecq S, *et al.* 2012 *EMBO J*, PMID: 23188086). While this is not immediately evident from our HSQC spectra of peptide-bound cHSP27, the fact that the CSP and CPMG RD data do not quantitatively agree points to a more complex underlying mechanism. We are intending to investigate the binding mechanism of the IxI/V motif in more detail in a follow-up study.

3.10 *Why are the k_{off} rates used together with the k_{D} to get the k_{on} rates. One could also have used the k_{on} rates with the k_{D} to get the k_{off} rates. If the authors had done that, I assume that they would observe a difference in the k_{off} rates and that they would conclude that the differences in k_{off} determine the differences in k_{D} . This would contradict the conclusion that "...NMR data point to attenuated affinity and association rate of the P182L IxI/V motif."*

We fixed the k_{off} value because our CPMG relaxation dispersion measurements indicated that the k_{off} rate was unchanged for the two peptides. We note that $k_{\text{ex}} = k_{\text{off}} + k_{\text{on}}[\text{L}]_{\text{free}}$, where $[\text{L}]_{\text{free}}$ refers to the concentration of the free ligand. Since k_{ex} is a fitting parameter and $[\text{L}]_{\text{free}}$ can be calculated from the difference between the known amount of added ligand and the fitted amount of bound ligand (via the fitted parameter p_{B}), we can solve explicitly for $k_{\text{off}} = k_{\text{ex}}(1-p_{\text{B}})$ and $k_{\text{on}} = (k_{\text{ex}}p_{\text{B}})/([\text{L}]_{\text{tot}} - [\text{L}]_{\text{bound}})$. The calculated values of k_{off} are nearly identical for the WT and P182L peptides at 759 and 840 s^{-1} , respectively. When we calculate the k_{on} values obtained from CPMG data for the WT and P182L peptides, we obtain a factor of 4 decrease in k_{on} for the P182L peptide: $3.0 \times 10^5 \text{ M}^{-1} \text{ s}^{-1}$ for WT compared to $8.3 \times 10^4 \text{ M}^{-1} \text{ s}^{-1}$ for P182L.

Since the difference in affinity, as measured by CPMG relaxation dispersion, originates from an attenuated k_{on} value, whereas k_{off} was unchanged, we chose to fix k_{off} and solve for k_{on} in the equation $K_{\text{d}} = k_{\text{off}}/k_{\text{on}}$. We have added text to the Methods section on p. 40 to address this important point.

3.11 *The MD simulations of the WT and P182L peptides shows that the removal of the proline increases flexibility. This finding is trivial and as expected. The authors then link the "preformed" conformation of the WT peptide with its increased affinity. It is clear that bi-molecular*

interactions are much more complex than that. Differences in solvation, flexibility of the peptide in the binding pocket (entropy), differences H-bonding propensity, differences in hydrophobicity and much more also determine the final affinity. The correlation that the authors find is most likely only one small part of a much more complex binding event. This should be clearly mentioned as the current presentation of the data is a huge over-simplification.

We thank the reviewer for this comment, and we are grateful that they raised this important point. Notably, we now mention that other factors beyond flexibility are important for molecular interactions, including hydrogen bond propensity, hydrophobicity, and solvation. We note, as well, that VDW interactions between the I and V in the IPV motif are also important, since the GPG mutant (discussed in comment #3.14 below) abolishes binding.

We have now updated our Results and Discussion sections on p. 16-17 and 21-22 to reflect the many other contributions to binding.

3.12 *The bioinformatic analysis is limited and very basic. The authors find that X=PRO is most likely in the [LV]-X-[LV] motif. This is hardly surprising, as PRO is a residue that interferes with secondary structure elements. Maybe the graph (Figure 5) should be scaled by the abundance of the individual aminoacids in disordered regions. In many regards, the lower panel of Figure 5 just shows the propensity that amino acids have to be in disordered regions (independent of the flanking [L/V] regions).*

We thank the reviewer for the helpful comment, and we address the concerns below. Our bioinformatics analysis is focused on the identity of the amino acid, X, within the motif [Ile/Val]-X-[Ile/Val]. While Pro is indeed enriched in disordered regions, this does not automatically indicate that it will also be enriched in [Ile/Val]-X-[Ile/Val] motifs, as both Ile and Val are depleted in disordered regions. Our original manuscript did not contain a statistical analysis on this topic, and we apologize for this omission. We have now updated the manuscript with statistical tests on the bioinformatics data. Namely, we performed a χ^2 test on the dataset to compare the observed vs. expected frequencies of [V/I]x[V/I] motifs. In brief, we found that amino acid frequencies alone cannot explain the observed numbers of [V/I]x[V/I] motifs. We direct the reviewer to our reply to comment #1.22 above for more details.

Were the reviewer's comment to be true – that the propensity of amino acids accounts for the observed differences in motifs – then the observed number of [Ile/Val]-X-[Ile/Val] motifs across the proteome would match expectations based on the frequencies of amino acids in the proteome (*cf.* equation 12, Methods). However, our χ^2 test indicates that Pro is statistically enriched in [V/I]-X-[V/I] motifs as compared to expectations from amino acid frequencies alone (Appendix Figure S8).

For example, the probability of finding a VPV motif within a disordered region, assuming that it simply scales by amino acid frequencies, is the frequency of Val squared times the frequency of Pro within disordered regions, multiplied by the total number of tripeptides. However, we observe ca. 600 VPV motifs in the disordered proteome, whereas amino acid frequencies predict only 400 such motifs. This is a 50% increase over the expected value that is based on amino acid frequencies present in the disordered proteome. This large discrepancy cannot be explained alone by amino acid frequencies, or the enrichment of Pro in disordered regions.

We have added text to the Results on p. 18-19 to address the reviewer's comment.

3.13 *The authors identified proteins that contain [L/V]-P-[L/V] motifs in a number of proteins. I am not sure if the GO analysis reveals any insights. The number of motifs in the most enriched groups and the group-size for the most enriched groups are very small (3 out of 21). The fold enrichment is thus likely highly uncertain. In addition, what do the authors conclude from the analysis. Does it make sense that many different processes are found? Is the chaperone known to be involved in the processes that are identified?*

We thank the reviewer for this comment. Because the GO Analysis was not sufficiently informative (or critical to the conclusions of the manuscript), we have decided to remove Appendix Table S5.

3.14 *Page 19: ". Indeed, we created a BAG3 mutant in which the IPV motifs were mutated to GPG". Based on the paper, I wonder if the authors also tested an ILV mutant for the BAG3 protein, as the mutant in the HSP29 protein is also a PtoL. This makes the experiments somewhat inconsistent. The GPG motif should have the same preformed conformations as that IPV motif, showing that the preformed conformation is not the only thing that counts (see above).*

The reviewer is correct in noting that the GPG motif should resemble the IPV motif with respect to the Pro182 ϕ/ψ dihedral angles, notwithstanding the increased flexibility imparted by Gly. However, removal of the hydrophobic residues adjacent to Pro182 in the mutation GPG abolishes binding of this motif (*cf.* Rauch JN, *et al.* (2017) *J. Mol. Biol.*), which indicates that other factors beyond flexibility are important for IxI/V binding, as established by the reviewer in the comment #3.11 above.

A Pro-to-Leu (IPV-to-ILV) mutant of BAG3 was tested for binding to HSP27 in a recent paper by the Kampinga group (Meister-Broekema M, *et al.* (2018) *Nat. Commun.*). It is important to note that BAG3 has two IPV motifs that bind to HSP27, leading to a BAG3:HSP27 stoichiometry of 2:1. The mutation P209L in BAG3 replaces the native IPV motif with an ILV motif, which is analogous to the P182L mutant of HSP27. Notably, the P209L form of BAG3 no longer binds with a 2:1 stoichiometry to HSP27; rather, the binding stoichiometry is reduced to 1:1, indicating the P209L mutant abolishes one of the binding interactions. These data can be found in Figure 2A of the above-mentioned paper.

We have updated our Results section on p. 22 to include mention of the importance of van der Waals contacts in the IxI/V binding mechanism, and how replacing the Ile and Val residues with Gly disrupts binding.

3.15 *The authors fail to address why the P182L mutant forms larger oligomers in solution. Based on the reduced affinity between the unfolded part of HSP29 and the folded core, I would expect smaller assemblies. In the discussion they mention a mechanism by which "concomitant binding of two IxI/V motifs facilitates subunit exchange via ejection of a monomeric subunit from the oligomer". I am afraid that I really don't understand this. What are the two IxI/V motifs binding to? In my understanding, they can bind to one surface patch, either inter- or intramolecular. The P182L mutation weakens both (as it is the same interaction) and the oligomers should be smaller. Please explain in detail.*

We have attempted to clarify this point in the Discussion on p. 22. A previous study (Baldwin *et al.* (2011) *J. Mol. Biol.*, PMID: 21839749) found a correlation between the rate of subunit exchange and the binding kinetics of the IxI/V motif. Given this framework, were the binding

kinetics of the IxI/V motif to be attenuated, this would result in slower subunit dissociation kinetics and lead to a net increase in oligomer size.

However, we now also examine a mechanism that is consistent with the above framework in which the IxI/V mutation may increase oligomer size. We refer Reviewer #3 to comments 1.6 and 2.4 above, in which we discuss in more detail the potential role of a second IxI/V-like motif that exists in the NTD and can bind to the β 4/ β 8 groove. We note that a recent cryo-EM study from the Buchner laboratory (Kaiser CJO, *et al.* (2019) *Nat. Struct. Mol. Biol.*) found that the similar sHSP, α A-crystallin, forms oligomers in which 70-80% of the subunits are bound to their own IxI/V motif (intramolecular). This suggests that intermolecular IxI/V binding is not necessarily a prerequisite for oligomerization.

3.16 *Page 20: "Therefore, attenuating the binding affinity of the IxI/V motif in the CTR would enable other sites in the NTD to bind more frequently and alter the oligomeric landscape of HSP27." Why would the oligomeric state be influenced when another part of the NTD interacts with the core domain? The intermolecular interaction remains present. Please explain.*

We thank the reviewer for this comment. For the P182L variant, we speculate that either the NTD or CTR from one oligomer is able to bind the available IxI/V binding site of another oligomer, thereby increasing the effective size. For the wild-type protein, however, the IxI/V binding site within a given oligomer is less available meaning that it is less likely to become occupied by another oligomer. Also, we note as above in comment 3.15, intermolecular CTR-ACD interactions are not necessary for oligomerization; rather, the CTR-ACD interaction within the WT protein might well be intramolecular. Lowering the affinity of the CTR-ACD interaction could enable an NTD-ACD interaction that is inter-molecular in nature.

We have now adapted the Discussion on p. 22 with this comment in mind. We hope that this expanded explanation, motivated by the reviewer's helpful comment, will aid in clarifying our thoughts.

3.17 *Finally, I fail to understand why the P182L mutation has reduced activity. The active surface patch of the chaperone is more accessible, and the activity should thus increase. Is the reduced activity solely due to the increase cluster size? Then the chaperone activity could be increased at a lower concentration, where oligomers are more likely to be smaller. Did the authors test this?*

We thank the reviewer for this comment on the reduced activity of P182L. We note that the 'active surface patch' of the chaperone does not necessarily correspond to the IxI/V binding site – previous publications (Mainz A, *et al.* (2015) *Nat. Struct. Mol. Biol.*; Freilich R, *et al.* (2018) *Nat. Commun.*; Baughman *et al.* (2020) *PNAS*) have shown that different regions of sHSPs contribute to chaperone activity for different substrates. In fact, in Baughman *et al.*, they suggest that binding of the substrate tau to the β 4/ β 8 groove does not lead to productive chaperoning. Instead, interactions with the N-terminal region are required to delay aggregation. So, the increased availability of the β 4/ β 8 groove may increase the interaction of substrates with the ACD, and thereby reduce the effective chaperone-capacity of HSP27 towards the client.

To control for possible differences in oligomer size, we added HSP27 at two different concentrations 20 μ M and 0.5 μ M, which were also chosen to be near equimolar with the two substrates. Under these conditions, the P182L variant was unable to prevent substrate

aggregation whereas wild-type HSP27 was highly effective at preventing aggregation. The two concentrations tested differ by a factor of 40, implying that the reduced activity is not due to oligomer size

We have updated our Discussion on p. 25 to speculate on possible mechanisms for the altered activity of the P182L variant.

Minor remarks:

3.18 References to the structure should be included in the figure legend 1D and E.

We have now included the relevant PDB ID (4mjh) in Figure 1.

Dear Dr. Alderson,

Thank you for submitting your revised manuscript to the EMBO Journal. Your study has now been seen by the original referees and their comments are provided below. The referees appreciate the added changes and are overall supportive of the study. However, as you can see below they have a few remaining points that needs to be addressed.

I would like to ask you to address the remaining points in a final revision. When you submit the revised manuscript please also address the following points:

- Please upload individual figure files.
- Callout to Fig 3E callout is missing.
- There are callouts to Supplementary Table 1 + 2, but I don't think they exist.
- Please add a ToC to the Appendix
- I see that the same control image is used in FigureS5A/B. Can you please confirm this is correct and if so mention this in the figure legend.
- I think it would be good to add the Python code to the manuscript.
- We require a Data Availability Section. As far as I can see no data deposition in external databases is needed for this paper. If I am correct please then state: This study includes no data deposited in external repositories
- We encourage the publication of source data, particularly for electrophoretic gels and blots, with the aim of making primary data more accessible and transparent to the reader. It would be great if you could provide me with a PDF file per figure that contains the original, uncropped and unprocessed scans of all or key gels used in the figure? The PDF files should be labeled with the appropriate figure/panel number, and should have molecular weight markers; further annotation could be useful but is not essential. The PDF files will be published online with the article as supplementary "Source Data" files.
- Our publisher has also done their pre-publication check on your manuscript. When you log into the manuscript submission system you will see the file "Data Edited Manuscript". Please take a look at the word file and the comments regarding the figure legends and respond to the issues. Please also use this version when you resubmit the revised version with the marked changes. Just makes it easier for me to see the changes.
- We include a synopsis of the paper (see <http://emboj.embopress.org/>). Please provide me with a general summary statement and 3-5 bullet points that capture the key findings of the paper.
- We also need a summary figure for the synopsis. The size should be 550 wide by 400 high (pixels). You can also use something from the figures if that is easier.

That should be it. If you have any further questions please just ask me

With best wishes

Karin

Karin Dumstrei, PhD
Senior Editor
The EMBO Journal

Further information is available in our Guide For Authors:

The revision must be submitted online within 90 days; please click on the link below to submit the revision online before 29th Jul 2020.

Link Not Available

Referee #1:

The authors' detailed responses and revisions are highly appreciated. The new data strengthen the study and clarify a number of issues. A few queries remain to be adequately addressed.

Specific comments

1.3 The P182L mutant protein was purified under denaturing conditions and refolded. The authors need to control for the structural integrity of the resulting protein. The experiments rely on the correct structure of the HSP27 ACD. In figure S1A, the P182L variant seems to be more aggregation prone, indicative of a potentially unstable, not correctly folded protein.

This is an important point raised by the reviewer. To address this issue in the manuscript, we have compared circular dichroism (CD) spectra of wild-type HSP27 and the P182L variant after purification from inclusion bodies. Importantly, upon overexpression in *E. coli*, the wild-type version of HSP27 is present in the soluble fraction and, thus, its CD spectrum provides a reference for the natively folded form.

Q: The CD spectra answer part of the question raised, but what about the stability of the mutant?

1.7 In figure 2B there is a very limited amount of particles present in the negative stain micrograph. Please show a larger field/a more densely populated area.

See also response to 3.2. We thank the reviewer for pointing this out, and on reflection have decided to remove the negative-stain EM images from the manuscript due to inevitable ambiguities from such analyses.

Q: I do not see which „inevitable ambiguities“ negative-stain EM should have. Given the surprisingly large size of the complex, it should be actually straightforward to visualize it. I only asked for a larger field view. EM images would be very interesting to get an (initial) idea on the organization and homogeneity of the mutant complex - compared to the wild type protein.

1.10 Figures 3/4: The studies on the affinities of the peptides are limited to the isolated ACD and NMR. If the concluded mechanism is right, the wildtype and mutant peptides should be able to influence the oligomer distribution of wt HSP27 (and vice versa influences should be observed for the P182L peptide). The respective titration experiments with full length Hsp27 are of high interest and needed to test the claimed mechanism.

We thank the reviewer for this comment. The reviewer suggests experiments in which the peptides that mimic the C-terminal region are added to full-length HSP27 to see how they influence oligomerization. Intuitively, attractive though these experiments are, and we have attempted such experiments in the past, they have proven not to be feasible in practice. We note that other groups have attempted such experiments, and they were also unable to detect peptide binding to WT HSP27 (Freilich R, et al. (2018) Nat. Commun., PMID: 30385828), even with the artificial peptide containing the F185H mutation that serves to enhance affinity.

Q: It is appreciated that the suggested experiments were performed. How is the mechanism suggested by the authors affected by this result? What aspects in addition to the binding of the IXI motif (which would be adequately represented by a peptide) need to be considered?

This should be discussed in the manuscript.

1.11 Can the results obtained for the ACD experiments be directly be transferred to full-length protein? The NMR data were recorded using the oxidized ACD whereas the other experiments were performed with full-length (presumably) reduced protein.

As the reviewer has pointed out, our NMR studies were performed on the oxidized form of the ACD whereas the experiments in cells used the reduced protein. However, the oxidation state of HSP27 does not affect binding of the IXI/V motif, as the disulfide bond is approximately 20 Å from the

peptide binding site, and the disulfide does not affect the conformation of the IxIV binding site in the β 4/ β 8 groove. The solution structure of oxidized ACD is nearly identical to the crystal structure of the reduced ACD (Rajagopal et al. (2015) J. Biomol. NMR, PMID: 26243512), and there are no changes to the chemical shifts of the β 4/ β 8 groove upon formation or reduction of the disulfide bond (Alderson TR, et al. (2019) Nat. Commun., PMID: 30842409). In addition, a crystal structure of the similar protein α B-crystallin bound to a peptide containing its IxIV motif (PDB: 4m5s) is identical to that of the E117C mutant, which contained a disulfidelinked dimer (PDB: 4m5t) (Hochberg GKA, et al. (2014) Proc. Natl. Acad. Sci. U S A, PMID: 24711386). Notably, E117 in α B-crystallin is in the same position as C137 in HSP27. Finally, unpublished results from our laboratory indicate that reduction of the disulfide bond in cHSP27 has no impact on the binding affinity of the IxIV-containing peptide. We therefore conclude that the oxidation state of HSP27 does not affect IxIV binding.

Regarding the direct transfer of our results from the ACD to the full-length protein, we direct the reviewer to our response to comment 1.10 above. In brief, the full-length protein is too large and polydisperse to be studied by solution-state NMR methods. Preliminary results from methylTROSY indicate that many resonances are absent from the spectrum, which suggests the oligomers may be too heterogeneous or undergoing widespread conformational exchange that precludes characterization.

Q: Indeed NMR experiments on the full-length protein seem out of reach for this study. However, what should be done is to explain in the discussion which experiments were done with truncated version and to whether these results can be transferred to the full-length proteins - or whether there are any caveats.

Referee #2:

The neuropathy-causing P182L mutation disregulates interactions of HSP27

The goal of this work is to investigate the alterations in the structure-function relationship of the inherited mutation P182L in human heat shock protein Hsp27 by investigating the oligomeric size, distribution, and chaperone activity of the mutant protein. This is a mutation that is linked to Charcot-Marie-Tooth disease, a prevalent neuropathy. The mutation is located in a well-conserved IxIV motif and it disrupts protein-protein interactions, leading to more interactions with other IxIV motif proteins (Bag3) and increases sHsp oligomeric size. This study suggests a general mechanism for how IxIV motif proteins interact with one another, which is important for maintenance of many cellular protein-protein interaction networks, primarily through NMR data.

This is a well-written manuscript that is of importance within the field of small heat shock proteins and molecular chaperones. The study contextualizes previous data and suggests mechanistic insights for this mutation in regard to the protein structure-function relationship.

The authors have sufficiently responded to concerns posed by the reviewers. Edits made in the text and in the updated figures more accurately describe potential mechanistic implications of the data.

Referee #3:

After re-reading the manuscript, I see significant improvements. Nevertheless, I have still major concerns regarding the on and off rate discussions.

I am still confused about figures 1F and G. The authors write "The absence of self-aggregation indicates that P182L stimulates substrate aggregation". Do the aggregates contain the substrate only or the substrate with P182L? Are these insoluble aggregates, or large soluble oligomers? Is the P182L form of the protein a "reverse" chaperone that stimulates aggregation of itself and the client proteins. And if this is the case, how would that influence the in vivo experiments that have been performed? Is the level of functional HSP27 not very lowered, because the chaperone, together with a client protein has aggregated into a dysfunctional state.

I don't get the consistency in the statement "the ensemble-averaged hydration radius of P182L at 38 nm is four-fold larger than the 9-nm oligomers formed by WT HSP27, consistent with a ca. 30-fold increase in mass." A four-fold increase in radius of a sphere gives a 64-fold increase in volume and thus a 64 fold increase in mass.

Regarding the fitting of the kD using the NMR data. The authors write in the rebuttal letter that the selected residues that are in fast exchange as only then equation 1 holds. That is true, but in that case, the authors should use equations that are valid for intermediate exchange to fit the other residues and not just remove those from the analysis.

Rebuttal letter (point 3.5): "The errors in peptide concentration were on the order of 1-2% based on the manufacturer's report on sample impurity, and therefore these are negligible in the error propagation." This clearly is an underestimation of the error in the concentration, the error must be much larger due to pipetting errors and so on. However, my question was related to the fact that the titration of the P182L peptide has not gone into saturation, which introduces very significant errors into the extraction of the kD value.

"($\Delta\omega_H = 0.3$ ppm or 1100 rad s⁻¹)... ($\Delta\omega_N = 0.2$ ppm or 90 rad s⁻¹)", that depends on the magnetic field strength of course.

Why don't the authors add figure 2 of the rebuttal letter into the supplement. If it should convince me, why should it not convince other readers?

I am still highly sceptic regarding the fact that the CPMG derived kD and the CSP derived kD are not in agreement. A factor of 4 is not very high, but definitely significant. The authors use the explanation that the CPMG experiments might pick up additional exchange processes that are not directly due to the binding-unbinding of the peptide. In case that is true, the on and off rates of that are determined using CPMGs are inaccurate and the dispersion cannot be used to determine the on and off rates. I am not saying that the data is wrong, but one cannot "cherry-pick" results that fit to a model and discard results that don't fit. What if the kD is actually not correct or much more inaccurate than stated (see above)?

I still think that the discussion that correlates peptide flexibility with the on-rate is still very hand-waving. What about differences in solvation of proline v/s leucine, what about differences in hydrogen bond potential. The authors pick the possibility of flexibility to back up their statement and simply wave other possibilities by stating "...among other potential factors...".

Response to reviewers

We thank the editor and the reviewers for their careful consideration of our revised manuscript, especially given the pressures on everyone's time at the moment. The reviewers were positive in their assessment of our research, stating that "*this is a well-written manuscript that is of importance within the field of small heat shock proteins and molecular chaperones*" and "*this study suggests a general mechanism for how IxI/V motif proteins interact with one another, which is important for maintenance of many cellular protein-protein interaction networks.*"

However, a few remaining concerns were raised, mainly with respect to the interpretation of the peptide binding kinetics. Below, we have responded to each concern and described the specific changes to the manuscript. The reviewers' original comments are shown in *italic* and our replies are coloured blue.

Referee #1:

The authors' detailed responses and revisions are highly appreciated. The new data strengthen the study and clarify a number of issues. A few queries remain to be adequately addressed.

Specific comments:

Q related to 1.3: *The CD spectra answer part of the question raised, but what about the stability of the mutant?*

We have added text to the Results on page 7 to mention that the thermal stability of the P182L variant is not significantly compromised, as shown by a prior study where the P182S variant melted at a comparable temperature to the WT protein (Chalova AS, *et al.* (2014) *BBA*, PMID: 25220807). Namely, the P182S variant melted at ~64 °C as compared to the WT protein at 70 °C. We were unable to repeat this exact experiment for the P182L mutant due to the (continued) closure of the lab for COVID-19. However, our Appendix Figure S1A shows that the P182L variant is stable at 40 °C at 20 μM, which is a 40-fold higher concentration than used in Figure 1F. This indicates that the P182L, like the other CMT-causing P182S variant, has a thermal stability that is similar to the WT protein.

Q related to 1.7: *I do not see which "inevitable ambiguities" negative-stain EM should have. Given the surprisingly large size of the complex, it should be actually straightforward to visualize it. I only asked for a larger field view. EM images would be very interesting to get an (initial) idea on the organization and homogeneity of the mutant complex - compared to the wild type protein.*

We agree with the reviewer that such EM data do not need to be ambiguous; we called them such given that in small fields of view there could be bias due to the restricted sampling. Were we able to access the labs we could in principle collect more data, but feel that this would not add to the manuscript in any substantive way, given the convincing in vitro SEC-MALS and in cell expansion microscopy data (which was bolstered significantly by the addition of the iPSC data in the revision).

Q related to 1.10: *It is appreciated that the suggested experiments were performed. How is the mechanism suggested by the authors affected by this result? What aspects in addition to the binding of the IXI motif (which would be adequately represented by a peptide) need to be considered?*

This should be discussed in the manuscript.

We thank the reviewer for this insightful comment. We have added text to the Results section on p. 17 and the Discussion section on page 23 to discuss the F185H mutation and its role in increasing binding affinity.

We note that F185 is poorly resolved in the cHSP27-peptide crystal structure (PDB: 4mjh), and its side chain is not modeled in the structure. This suggests that F185 is dynamic and not stably interacting with other atoms. To visualize the impact of the F185H mutation *in silico*, we replaced F185 with a His residue in PyMol. Our analysis below (Figure 1) shows that the H^{ε2} proton of His185 could potentially form a stabilizing hydrogen bond with the backbone carbonyl from Pro150, located in the loop between $\beta 6+7$ and $\beta 8$ strands. Therefore, in addition to the van der Waals contacts established between the IxI/V motif and the hydrophobic $\beta 4/\beta 8$ groove, polar interactions that are outside the immediate IxI/V motif also contribute to the binding mechanism of the peptide.

Figure 1. The F185H mutation could create a new hydrogen bond between the H185 side chain and the P150 carbonyl. The mutation was introduced into the crystal structure of cHSP27 bound to the IxI/V-bearing peptide (PDB: 4mjh) using the PyMol mutagenesis tool. The peptide is colored grey, the F185H mutation is colored green, and cHSP27 is colored black. The $\beta 4$ and $\beta 8$ strands are labeled. Residues that are shown in sticks are colored by element type (oxygen is red and nitrogen is blue). The 2.6-Å distance was calculated from the coordinates of the H185(N^{ε2}) and P150(CO) atoms, and it does not take into account the otherwise absent H^{ε2} proton nor the distance of the N^{ε2}-H^{ε2} bond. The H185 rotamer was selected manually and does not reflect an energy-minimized structure.

Q related to 1.11: *Indeed NMR experiments on the full-length protein seem out of reach for this study. However, what should be done is to explain in the discussion which experiments were done with truncated version and to whether these results can be transferred to the full-length proteins - or whether there are any caveats.*

We thank the reviewer for raising this concern. We note that a recent NMR study from the Klevit laboratory (Clouser AF, *et al.* (2019) *eLife*, PMID: 31573509) showed that the spectrum of the ACD overlays well with that of the full-length protein. This suggests that the structure of the ACD is similar in both constructs. Notably, the authors had to add multiple mutations (triple phosphomimic and ¹⁸¹IPV¹⁸³-to-¹⁸¹GPG¹⁸³) to make HSP27 amenable to solution-state NMR. These mutations prevent us from analyzing full-length P182L in a similar approach.

We have modified the text in the Discussion on page 22 to specify which experiments made use of which construct.

Referee #2:

The neuropathy-causing P182L mutation dysregulates interactions of HSP27

The goal of this work is to investigate the alterations in the structure-function relationship of the inherited mutation P182L in human heat shock protein Hsp27 by investigating the oligomeric size, distribution, and chaperone activity of the mutant protein. This is a mutation that is linked to Charcot-Marie-Tooth disease, a prevalent neuropathy. The mutation is located in a well-conserved IxI/V motif and it disrupts protein-protein interactions, leading to more interactions with other IxI/V motif proteins (Bag3) and increases sHsp oligomeric size. This study suggests a general mechanism for how IxI/V motif proteins interact with one another, which is important for maintenance of many cellular protein-protein interaction networks, primarily through NMR data.

This is a well-written manuscript that is of importance within the field of small heat shock proteins and molecular chaperones. The study contextualizes previous data and suggests mechanistic insights for this mutation in regard to the protein structure-function relationship.

The authors have sufficiently responded to concerns posed by the reviewers. Edits made in the text and in the updated figures more accurately describe potential mechanistic implications of the data.

We thank the reviewer for the positive feedback and compliments about our work. We are grateful for Reviewer #2's suggested changes that helped to strengthen the conclusions drawn from our study.

Referee #3:

After re-reading the manuscript, I see significant improvements. Nevertheless, I have still major concerns regarding the on and off rate discussions.

We are grateful for the reviewer's comments that helped to improve our manuscript.

*I am still confused about figures 1F and G. The authors write "The absence of self-aggregation indicates that P182L stimulates substrate aggregation". Do the aggregates contain the substrate only or the substrate with P182L? Are these insoluble aggregates, or large soluble oligomers? Is the P182L form of the protein a "reverse" chaperone that stimulates aggregation of itself and the client proteins. And if this is the case, how would that influence the *in vivo* experiments that have been performed? Is the level of functional HSP27 not very lowered, because the chaperone, together with a client protein has aggregated into a dysfunctional state.*

We thank the reviewer for asking for a deeper understanding of this phenomena, and we agree that it concerns a particularly interesting aspect. We have updated our manuscript on page 26 of the Discussion to expand on this important point.

The Mayer, Mogk, and Bukau groups have previously published on the aggregation-enhancing activity of the sHSP Hsp42 (Ungelenk S, *et al.* (2016) *Nat. Commun.*, PMID: 27901028), which catalyzes the aggregation of its substrate proteins *in vitro*. Such activity has been termed "aggregase" activity, and it was found that it provided cellular fitness to *S. cerevisiae* during heat stress. Moreover, it was noted that sHSPs often co-aggregate with their substrates *in vivo* (references 17-20 therein). Aggregase activity could reduce the burden on the protein quality control system during stress by storing misfolded proteins in aggregated states, thus minimizing the number of potentially toxic soluble oligomers and enabling eventual disaggregation by the disaggregase machinery.

We did not measure whether the soluble/insoluble aggregates in Figure 1 contained both P182L and substrate or only the substrate (and we apologize that we are not able to repeat this experiment due to closure of the lab for COVID-19). Thus, we are hesitant to infer too much from these light scattering experiments, which only partially replicate the scenario found *in vivo*. From these experiments, we can only conclude that, under these conditions, the P182L variant does not exhibit the same chaperone activity as the WT protein.

The altered chaperone activity of P182L is not caused by misfolding or lack of stability, as evidenced by Appendix Figure 1. Rather, the altered chaperone activity is caused by the P182L mutation. In addition, we showed that a significant fraction of P182L remains soluble *in vivo* (Appendix Figure S2C, 2D), ruling out that aggregation of the mutant is the single mechanism driving the cell into a dysfunctional state. Moreover, altered protein-protein interactions (Figure 5) were detected by immunoprecipitating P182L from the soluble fraction, indicating that at least part of the cellular dysregulation stems from the non-aggregated form of the P182L mutant.

So, we agree with the reviewer that this aspect of self-aggregation versus co-aggregation merits further investigation (and note that it is an area of active research for other sHSPs and chaperones). However, we would propose to perform this in a more relevant cellular context by purifying P182L aggregates, such as those found in the stem-cell derived neurons, and use mass spectrometry based proteomics to identify other co-aggregating factors which might be more relevant to disease than the artificial substrate used in Figure 1. We speculate that competition

for binding of P182L to substrates, IxI/V-containing proteins, and other P182L molecules regulates its overall status in the cell but a more rigorous investigation should be performed in the future to address this question in greater detail.

I don't get the consistency in the statement "the ensemble-averaged hydration radius of P182L at 38 nm is four-fold larger than the 9-nm oligomers formed by WT HSP27, consistent with a ca. 30-fold increase in mass." A four-fold increase in radius of a sphere gives a 64-fold increase in volume and thus a 64 fold increase in mass.

The reviewer's calculation is correct under the assumption that ensemble-averaged HSP27 oligomers form a sphere of uniform density. However, the oligomers comprised of HSP27 almost certainly are not of this structure (Haley DA, *et al.* (2000) *J. Mol. Biol.*, PMID: 10764595), and we therefore, expect there to be deviations from spherical calculations.

To obtain additional insight, we can use empirical equations that relate the radius of hydration (R_h) to the number of residues (n) of a protein as outlined in Wilkins DK, *et al.* (1999) *Biochemistry*, PMID: 10600103. For folded proteins, $R_h(n) = 4.75n^{0.29}$, and inserting values of n for WT and P182L to account for their masses, we obtain an expected R_h of 6 nm for WT and 16 nm for P182L, corresponding to a 2.7-fold increase for P182L. However, both values are smaller than what we have measured experimentally. For fully denatured proteins, $R_h(n) = 2.21n^{0.57}$, and the R_h ratio between WT and P182L increases to ca. 7-fold, while the R_h values themselves become far too large (30 nm and 207 nm).

We know that HSP27 is neither fully structured nor fully disordered, as the protein contains a structured ACD that is flanked by disordered N- and C-terminal domains. Therefore, the dependence of HSP27's R_h on molecular mass will most likely reflect a combination of the empirical equations shown above for structured and disordered proteins. Indeed, the measured 4.2-fold difference in R_h falls between the expected differences of 2.7 (structured) and 7 (disordered), falling closer to the expectations for structured proteins.

We have updated the text on p. 8 to include the relation between hydrodynamic radius and molecular mass.

Regarding the fitting of the kD using the NMR data. The authors write in the rebuttal letter that the selected residues that are in fast exchange as only then equation 1 holds. That is true, but in that case, the authors should use equations that are valid for intermediate exchange to fit the other residues and not just remove those from the analysis.

To include other exchange regimes in the determination of the K_d , we used the software TITAN (Waudby CA, *et al.* (2016) *Sci. Rep.*, PMID: 27109776), which performs density operator calculations in Liouville space and simulates 2D spectra, accounting for chemical exchange via the Bloch-McConnell equations. Thus, all exchange regimes are considered. Notably, TITAN reports the same K_d as that obtained by fitting the fast-exchange CSPs in Figure 3C.

We have updated the text on p. 13 to include mention of this.

Rebuttal letter (point 3.5): "The errors in peptide concentration were on the order of 1-2% based on the manufacturer's report on sample impurity, and therefore these are negligible in the error propagation." This clearly is an underestimation of the error in the concentration, the error must

be much larger due to pipetting errors and so on. However, my question was related to the fact that the titration of the P182L peptide has not gone into saturation, which introduces very significant errors into the extraction of the k_D value.

As we are limited by the solubility of the P182L peptide (ca. 5 mM) and the low affinity of this peptide, we were not able to saturate the cHSP27-P182L interaction. However, we note that the 2D lineshape fitting implemented in TITAN has been shown to obtain accurate K_d values even when the binding reaction has not been fully saturated. For example, see Figure S1 in the original TITAN paper (Waudby CA, *et al.* (2016) *Sci. Rep.*, PMID: 27109776). Furthermore, we obtain the same K_d from fitting CSPs to the analytical equation and from simulating 2D spectra in the presence of chemical exchange with TITAN, which suggests that there are enough data points over a sufficient range, such that fitting errors are not significant.

"($\Delta\omega_H = 0.3$ ppm or 1100 rad s⁻¹).... ($\Delta\omega_N = 0.2$ ppm or 90 rad s⁻¹)", that depends on the magnetic field strength of course.

We thank the reviewer for noticing this omission. We have modified the text on p. 15 as below: " $(\Delta\omega_H = 0.3$ ppm or 1100 rad s⁻¹ at 14.1 T) ... ($\Delta\omega_N = 0.2$ ppm or 90 rad s⁻¹ at 14.1 T)"

Why don't the authors add figure 2 of the rebuttal letter into the supplement. If it should convince me, why should it not convince other readers?

We appreciate the comment of the reviewer and agree that the reader might benefit from this figure. We have added a new Appendix Figure S8, which compares the CPMG RD chemical shift changes with those measured directly from HSQC spectra of the free and peptide-bound protein.

We note that we do not have resonance assignments for the peptide-bound form of cHSP27, and our analysis was therefore not quantitative. For the peaks analyzed in Appendix Figure S8, we categorized them based on our degree of confidence. Two resonances can be followed during the titration (T110 and S154) – even though they are broadened – and we are therefore confident about their assignments. For the remaining signals, we are hesitant to over-analyze the data without assignments.

I am still highly sceptic regarding the fact that the CPMG derived k_D and the CSP derived k_D are not in agreement. A factor of 4 is not very high, but definitely significant. The authors use the explanation that the CPMG experiments might pick up additional exchange processes that are not directly due to the binding-unbinding of the peptide. In case that is true, the on and off rates of that are determined using CPMGs are inaccurate and the dispersion cannot be used to determine the on and off rates. I am not saying that the data is wrong, but one cannot "cherry-pick" results that fit to a model and discard results that don't fit. What if the k_D is actually not correct or much more inaccurate than stated (see above)?

We thank the reviewer for this comment, and we agree with the statement. To avoid any bias, we have replaced the bar graph in Figure 4C with one showing the rates obtained directly from the CPMG analysis. This removes the circular reasoning as presented originally when using CSP and CPMG data. Interpreting the CPMG data directly is clearer and shows that the association rate for the P182L peptide is attenuated.

Regarding the discrepancy between CPMG and CSP K_d values, at present we do not have a validated explanation, but we speculate that additional microsecond dynamics in the bound state cause line-broadening. This discrepancy has also been observed for peptide binding to α B-crystallin (*c.f.* Olga Tkachenko's DPhil Thesis, University of Oxford) and warrants further investigation.

I still think that the discussion that correlates peptide flexibility with the on-rate is still very hand-waving. What about differences in solvation of proline v/s leucine, what about differences in hydrogen bond potential. The authors pick the possibility of flexibility to back up their statement and simply wave other possibilities by stating "...among other potential factors....".

We refer the Reviewer to Reviewer #1's comment above about the F185H mutation.

We have added text on p. 17 in the Results section to discuss the role of solvation and hydrogen bond potential in binding of the peptide bearing the IxI/V motif. We have also discussed the role of polar interactions on p. 23.

Dear Reid, Justin, Andrew and Vincent

I have provided below the link for you to upload the revised manuscript.

Let me know if we need to discuss anything further

best Karin

Karin Dumstrei, PhD
Senior Editor
The EMBO Journal

Further information is available in our Guide For Authors:

The revision must be submitted online within 90 days; please click on the link below to submit the revision online before 25th Oct 2020.

Referee #1:

The authors have addressed my queries in a satisfactory manner.
The only point I still don't understand is why there is (obviously) only one EM picture available. It seems highly unusual to take just one micrograph of a sample once it is in the electron microscope. If there are more, they should be added in the supplement.

Referee #3:

The authors now performed a line-shape analysis using the program TITAN to extract the kD and the off-rate that are associated with the interaction of ACD with WT and P128L peptides. They write that the kD they obtain is highly similar to the kD that is obtained from the simple analysis of CSPs. The power of TITAN (line-shape analysis) lies in the fact that this approach provides direct insights in the kinetics of the binding process. I am sure that the authors noticed that the difference in the kD between the WT and P128L peptide (in the TITAN analysis) can be explained to a large degree by a difference in the off-rate. This contradicts the conclusion that the authors draw based on the CPMG analysis, where the difference in kD is solely due to the difference in the on-rate. There are thus still larger discrepancies between the interpretation of the authors and the data.

I suggest that an additional NMR expert reviewer has a look at this in detail.

Page 13, Line 5: "1100 rad⁻¹" should read "1131 rad s⁻¹" (both the number and the unit are wrong).

Images in the rebuttal letter should be in the supplement, as not only the reviewers, but also the readers should have access to this information.

Response to reviewers

We are grateful to the editor and the reviewers for their consideration of our revised manuscript and their careful comments. Below, we have responded to the comments and described the specific changes to the manuscript. The reviewers' original comments are shown in *italic* and our replies are coloured blue.

Reviewer #1:

The authors have addressed my queries in a satisfactory manner. The only point I still don't understand is why there is (obviously) only one EM picture available. It seems highly unusual to take just one micrograph of a sample once it is in the electron microscope. If there are more, they should be added in the supplement.

The original image was representative of other images in the P182L negative-stain EM dataset. Based on Reviewer #1's earlier comments, we decided to remove the EM image from the manuscript due to their poor quality. However, this was not because we didn't have more available – we do. We have included additional EM images below for Reviewer #1's examination, and we have also added them to the manuscript as an additional supplementary figure as requested (Appendix Fig. S2).

Appendix Figure S2. Negative-stain electron microscopy of HSP27. Negative-stain electron microscopy (EM) images for WT (A-D) and P182L (E-H). Scale bars are depicted in the lower left corner of each image. The dashed boxes in panels C and G correspond to the regions that were subsequently imaged at higher magnification in panels D and H, respectively.

Reviewer #3

The authors now performed a line-shape analysis using the program TITAN to extract the K_d and the off-rate that are associated with the interaction of ACD with WT and P128L peptides. They write that the K_d they obtain is highly similar to the K_d that is obtained from the simple analysis of CSPs. The power of TITAN (line-shape analysis) lies in the fact that this approach provides direct insights in the kinetics of the binding process. I am sure that the authors noticed that the difference in the K_d between the WT and P128L peptide (in the TITAN analysis) can be explained to a large degree by a difference in the off-rate. This contradicts the conclusion that the authors draw based on the CPMG analysis, where the difference in K_d is solely due to the difference in the on-rate. There are thus still larger discrepancies between the interpretation of the authors and the data.

The reviewer continues to make an excellent point about the apparent discrepancies in the NMR analyses, and we appreciate their efforts in ensuring that the presentation of this aspect of this manuscript is exhaustive. Previously, we concentrated on two key observations: that the apparent binding of mutant is in terms of K_d roughly one order of magnitude weaker than the WT, and that the difference is principally due to variation in the on rate. We agree that, at face value, there is an apparent discrepancy between the values obtained from the CPMG and 'line-shape' analysis from the software TITAN. We have re-evaluated our presentation to provide a more detailed description of the data and analysis in the manuscript, which we have summarised below. In essence, the discrepancy comes down to different assumptions made in the various analyses. While a more detailed experimental characterisation of the various states is outside the scope of this work, we have performed a considerable amount of additional analysis to demonstrate and quantify the sources of apparent conflict. Overall, the quantitative picture we obtain – that the mutant peptide binds more weakly than the WT, and the difference is due to a change in the on rate – is extremely clear, and the various measurements we present are self-consistent.

We agree with the sentiment expressed by the reviewer previously: bi-molecular interactions can actually be complex binding events, and a typical two-state "on/off" model is often a simplification (Sugase K, Dyson HJ, Wright PE (2007) *Nature* 447: 1021-25; Schneider R, *et al.* (2015) *J. Am. Chem. Soc.* 137: 1220-29; Delaforge E, *et al.* (2018) *J. Am. Chem. Soc.* 140: 1148-58). In our NMR titration data, we see evidence for three conformational states, corresponding to one in which the peptide is unbound and two where it is bound. The evidence for there being two distinct bound states is two-fold. Firstly, the fitted R_2 values from TITAN for residues in the vicinity of the binding pocket under conditions of excess peptide are extremely high, which manifests itself as a substantial reduction in signal intensity of the observed 'bound' state (Appendix Fig. S6A). Secondly, when we elevate the temperature from 25 °C to 50 °C, we are able to essentially completely recover the signal (Appendix Fig. S6B, Appendix Fig. S7A). This tells us that, at 25°C and at high peptide concentrations, our observed 'bound' state is actually an equilibrium between at least two conformational states, and hence we need to consider at least three states in total to explain the data.

Importantly for this article, focused on the biological relevance of the interactions between wild type and mutant peptides for HSP27, when we perform a naïve 'two-state' analysis of both the CPMG and titration data using TITAN we obtain 'apparent' K_d values (true within the constraints of the two-state model) from each experiment. From these, we can determine the difference in binding affinity between wild type and mutant peptides ($\Delta\Delta G$ values, Appendix Table S2,

Appendix Table S3). We quantitatively see the same result in both the titration and CPMG analyses: the K_d for the mutant peptide is roughly one order of magnitude lower than that WT.

However, the point is fundamentally interesting, and so we have now performed a more detailed analysis of the data to account for the three states. In essence, under these conditions when performing a two-state analysis of the CPMG data, it is not prudent to make the typical assumption that the R_2 values of the two states are the same (Palmer AG III, *et al.* (2001) *Methods Enzymol.* 339:204-238); we know from the titration data that the R_2 of the bound states is exceedingly large because of the interplay between (at least) two conformational states. We nevertheless obtain a consistent K_d value when we group the 'fast' and 'slow' exchanging residues in TITAN, and can reasonably infer that, despite the simplification, a two-state fit of the titration data using TITAN has well characterised the overall equilibrium between free and bound conformational states. This conclusion is similar to that published previously for the related protein α B-crystallin, where NMR titration data (Delbecq SP, *et al.* (2012) *EMBO J* 31: 4587-94) agrees well with that from native mass spectrometry (Hilton GR, *et al.* (2013) *Philos. Trans. R. Soc. Lond. B* 368: 20110405).

We can accommodate these two observations by performing an adjusted analysis of the CPMG data in which we fix the population of the two states to be consistent with the K_d values measured from the titration data using TITAN (4.9% and 7.2% bound for the WT and mutant, respectively), and allow the relaxation rates of the 'bound' state to have a substantial contribution from chemical exchange ($\Delta R_2 \neq 0$). The resulting fits are excellent (Appendix Fig. S11A, $\chi^2_{\text{red}} = 1.39$ and 1.86 for WT and P182L, respectively). This reveals how the TITAN fits and CPMG analysis are quantitatively consistent with each other. Most notably, the chemical shift differences obtained from CPMG with the assumption $\Delta R_2 \neq 0$ are essentially unchanged from those that we obtain from the naïve two-state fits with $\Delta R_2 = 0$ (Appendix Fig. S11A, RMSD = 0.28 ppm and 0.44 ppm for WT and P182L, respectively), which is because these values come primarily from the differences in the CPMG curves with magnetic field strength, whereas kinetic factors like on- and off-rates are independent of field.

Taken together, this analysis shows us that the TITAN data and the CPMG results become consistent only when we interpret the equilibrium in terms of a 'three-state' model, involving a free, and two bound states, where the two bound conformational states are distinct from each other and are in intermediate exchange at 25 °C.

From this point of consolidation, we then sought to ascertain reliable estimates for the rate constants for with peptide binding and unbinding from our various measurements and analyses. When we performed a detailed error analysis of the uncertainties on the on- and off-rates from TITAN, we obtain very large uncertainties. This is largely owing to correlation between various parameters and the fact that we have to use a two-state model to fit three-state data. Notably, the error distributions for these parameters are not centrally distributed, and so we need to quote vastly different values for our '+' and '-' error bars (Appendix Table S2). By contrast, our measured values from CPMG experiments are significantly more precise. While the absolute value measured for the on and off rates does vary depending on how we perform the CPMG analysis (either $\Delta R_2 = 0$, or $\Delta R_2 \neq 0$), the overall kinetic $\Delta\Delta G$ values (Appendix Table S3) for the on and off rates remain consistent. This confirms that (as we reported in previous versions of this manuscript), the most significant difference in the rates between WT and mutant is indeed the on rate.

Overall, a more detailed characterisation of the three states would be possible using a range of sophisticated NMR experiments, but this is far beyond the scope of this article. For here,

complexity aside, it is evident that no matter which NMR experiment we use, the K_d of the mutant is an order of magnitude weaker than the WT, and the majority of this change we can rationalise in terms of a decrease in the on rate. We thank the reviewer for their detailed inspection of these values.

To describe this new analysis in the revised manuscript, and how it reconciles the line-shape analysis of the titration data and the CPMG data we have made the following modifications:

- 1) We make it clear from the outset that the data reveals at least three states (p. 13-14).
- 2) We have justified the use of a two-state model for extracting the K_d from the titration data (p. 16-17)
- 3) We have included the following supplementary figures (Appendix Fig. S6, Appendix Fig. S7, Appendix Fig. S11) that show:
 - a. The “missing” NMR signal for the bound state is fully recovered at high temperature (Appendix Fig. S6A, S6B, Appendix Fig. S7A, S7B, S7C)
 - b. The chemical shifts in the CPMG relaxation dispersion analysis are unchanged between the two-state fit and the global fit of the data (Appendix Fig. S11A)
 - c. The ΔR_2 values fitted by TITAN in the titration analysis correspond to the same residues that have missing intensity in peptide-bound cHSP27 (Appendix Fig. S7)
- 4) We have tabulated the salient thermodynamic and kinetic parameters for each of the WT and mutant, and differences (or fold-change) between them (Appendix Table S2).
- 5) We have added text to the discussion that remarks on multiple states involved here in the context of similar observations on different systems in the literature (p. 25).

Appendix Tables S2 and S3, and a new Appendix Fig. S11 are included below:

Appendix Table S2. Kinetic parameters for WT and P182L peptide binding to cHSP27. Values of k_{on} ($M^{-1} s^{-1}$), k_{off} (s^{-1}), and K_d (μM) as derived from the titration and CPMG RD analyses on peptide-bound cHSP27. The K_d refers to the dissociation constant of peptide binding to cHSP27, with k_{off} and k_{on} the dissociation and association rates, respectively. The superscripts ^{CPMG}, ^{CPMG, ΔR_2} , and ^{TITAN} respectively refer to values determined using CPMG RD with a two-state fit, CPMG RD using a fixed p_B at the value expected from the K_d and $\Delta R_2 \neq 0$, or lineshape analysis with the software TITAN. The values listed within square brackets correspond to a parameter with a non-Gaussian χ^2 distribution, and the values are the minimum and maximum values that have values of $\exp(-(\chi^2 - \chi^2_{min})/2) \geq 0.6$. The k_{ex} value for TITAN was calculated based on the k_{off} and k_{on} rates and the calculated p_B (see Methods) using the equation $k_{ex} = k_{off} + k_{on}[L]_{free}$.

Peptide	Dataset	K_d (μM)	k_{off} (s^{-1})	k_{on} ($M^{-1} s^{-1}$)	p_B (%)	k_{ex} (s^{-1})
WT	CSP	124 \pm 13	----	----	4.9 \pm 0.5	----
	TITAN	132 \pm 58	207 \pm 85	1.6 \pm 0.7 $\times 10^6$	4.9 \pm 2.2	217 \pm 130
	CPMG	1,762 \pm 120	515 \pm 33	2.9 \pm 0.1 $\times 10^5$	2.4 \pm 0.1	528 \pm 30
	CPMG ΔR_2	124 (fixed)	346 \pm 25	2.8 \pm 0.2 $\times 10^6$	4.9 (fixed)	364 \pm 21
P182L	CSP	1030 \pm 118	----	----	7.2 \pm 0.7	----
	TITAN	1285 [700, 2000]	771 [350, 1750]	6.0 [3.8, 9.1] $\times 10^5$	7.0 [5.5, 8.8]	828 [386, 1836]
	CPMG	8,373 \pm 534	610 \pm 39	7.3 \pm 0.2 $\times 10^4$	2.0 \pm 0.1	622 \pm 35
	CPMG ΔR_2	1030 (fixed)	547 \pm 41	5.0 \pm 0.2 $\times 10^5$	7.2 (fixed)	590 \pm 33

Appendix Table S3. Energetic parameters for WT and P182L peptide binding to cHSP27. The differences in free energy ($\Delta\Delta G = \Delta G_{WT} - \Delta G_{P182L}$) are shown for the fitted values of K_d , k_{off} , and k_{on} . All units are in kcal mol^{-1} with the calculations performed using a temperature of 298.15 K. The datasets CSP, TITAN, CPMG, and CPMG ΔR_2 respectively refer to values determined from fitting fast-exchange resonances, fast- and slow-exchange resonances via TITAN, CPMG RD with a two-state fit, and CPMG RD using a fixed p_B at the value expected from the K_d and $\Delta R_2 \neq 0$. The values listed within square brackets correspond to a parameter with a non-Gaussian χ^2 distribution and are the minimum and maximum values that have values of $\exp(-(\chi^2 - \chi^2_{min})/2) \geq 0.6$. The k_{ex} value for TITAN was calculated based on the k_{off} and k_{on} rates and the calculated p_B (see Methods) using the equation $k_{ex} = k_{off} + k_{on}[L]_{free}$.

Dataset	$\Delta\Delta G K_d$	$\Delta\Delta G k_{off}$	$\Delta\Delta G k_{on}$
CSP	-1.25 ± 0.19	----	----
TITAN	$-1.34 [0.49, -2.29]$	$-0.78 [-0.24, -1.84]$	$0.58 [0.37, 0.88]$
CPMG	-0.92 ± 0.09	-0.10 ± 0.01	0.82 ± 0.06
CPMG ΔR_2	-1.25	-0.27 ± 0.03	1.02 ± 0.08

Appendix Figure S11. Global analysis of the dispersion and titration data. (A) Correlation plot showing the fitted $^{15}\text{N} |\Delta\omega|$ values in CPMG RD experiments on $^2\text{H}/^{15}\text{N}$ -cHSP27 in the presence of WT peptide. The x axis shows the $^{15}\text{N} |\Delta\omega|$ values obtained from a standard fit to a two-state model with ΔR_2 ($\Delta R_2 = R_{2,A} - R_{2,B}$) set to zero, and the y axis shows the fit performed with p_B fixed at the value expected based on a K_d of 125 μM (4.9%) and ΔR_2 ($\Delta R_2 = R_{2,A} - R_{2,B}$) allowed to be nonzero. The RMSD is shown in the upper-left corner. The reduced χ^2 values for the two fitting approaches are shown in the lower-right corner with χ^2_{red} corresponding to the naïve two-state model and $\chi^2_{\text{red}, \Delta R_2}$ corresponding to the fit with p_B fixed at 4.9% and $\Delta R_2 \neq 0$. (B) The fitted k_{ex} and p_B values from panel A for the two fitting approaches. Note that p_B was fixed at 4.9% in the $\Delta R_2 \neq 0$ fitting approach. (C) The fitted values of ΔR_2 shown alongside the fitted $^{15}\text{N} |\Delta\omega|$ values. (D-F) the same as panels A-C except for the CPMG RD experiments on $^2\text{H}/^{15}\text{N}$ -cHSP27 in the presence of the P182L peptide with p_B fixed at 7.2%. (G-L) *Top*: ^{15}N dispersions from $^2\text{H}/^{15}\text{N}$ -cHSP27 in the presence of the WT or P182L peptide for residues T113 (G, J), T151 (H, K), and S156 (I, L). Data at 600 and 500 MHz are respectively shown with filled and empty circles. The best-fit values for the standard, 2-state model are shown in dashed lines, whereas the global model with fixed p_B and a non-zero ΔR_2 value are shown in solid lines. *Bottom*: the difference between the best-fit values at 600 (filled) and 500 MHz (empty circles), with Δ referring to the values from the global model minus the values from the standard 2-state model.

Page 13, Line 5: "1100 rad-1" should read "1131 rad s-1" (both the number and the unit are wrong).

We thank the reviewer for highlighting this mistake and have updated the text accordingly. We moved this sentence to the Methods on p. 44.

Images in the rebuttal letter should be in the supplement, as not only the reviewers, but also the readers should have access to this information.

We have included Figure 1 from the previous Response to Reviewers as a new Appendix Figure S14 (referenced on p. 25 of the main text).

Dear Justin, Reid, Andrew and Vincent,

Thank you for submitting your revised manuscript to The EMBO Journal. Your study has now been re-reviewed by referee # 3 and the comments are provided below. As you can see the referee appreciates the introduced changes and support publication here. There are just two minor points that have to be resolved. When you submit the revised version will you also take care of the following points:

- Thanks for providing source data. For source data for main figures please upload one source data file per figure. For source data for Appendix figures please compile them as one file per figure and zip then together as one file.
- Some references have more than 10 authors listed. Please cut authors after 10 authors followed by et al.

That should be all!

With best wishes

Karin

Karin Dumstrei, PhD
Senior Editor
The EMBO Journal

When assembling figures, please refer to our figure preparation guideline in order to ensure proper formatting and readability in print as well as on screen:
<https://bit.ly/EMBOPressFigurePreparationGuideline>

Further information is available in our Guide For Authors:

The revision must be submitted online within 90 days; please click on the link below to submit the revision online before 7th Apr 2021.

Referee #3:

The authors have now very well addressed my concerns. I suggest publication when the following two points have been addressed:

1. For the P182L protein there are two fractions with different molecular weight in the sucrose gradient (Fig 2B). Please comment on that. The SEC-MALS data show a single population.
2. Figure 2E: The labels on the y-axis are spaced non-linearly. Please correct this, or in case this is intentional, comment on this.

Response to reviewers

We are grateful to the editor and the reviewer for their comments on our revised manuscript. Below, we have responded to the comments and described the specific changes to the manuscript. The reviewers' original comments are shown in *italic* and our replies are coloured blue.

Reviewer #3:

Referee #3:

The authors have now very well addressed my concerns. I suggest publication when the following two points have been addressed:

1. For the P182L protein there are two fractions with different molecular weight in the sucrose gradient (Fig 2B). Please comment on that. The SEC-MALS data show a single population.

We thank the reviewer for this interesting comment. It is well known that HSP27 oligomers can dissociate into smaller species at lower protein concentrations or upon phosphorylation or phosphomimicry of S15, S78, and S82 (e.g., Figure 1 in Jovcevski B, *et al.* (2015) *Chem. Biol.* 22: 186-195). In cases where significant populations of large and small oligomers are present, multiple elution peaks can be observed via size exclusion chromatography (SEC). However, at higher protein concentrations, HSP27 typically elutes as a single peak reflective of the larger oligomers. Observation of a single elution peak by SEC, however, does not mean that only a single population of oligomers is present, because HSP27 and other sHSPs form a polydisperse ensemble of oligomers (e.g., Hochberg GKA, *et al.* (2014) *Prog. Biophys. Mol. Biol.* 115: 11-20), and SEC elution profiles reflect the ensemble-averaged nature of the sample.

Our SEC-MALS data in Figure 2 were collected at an HSP27 monomer concentration of 40 μ M, which promotes formation of larger oligomers and, hence, explains why a single peak is observed. By contrast, the sucrose gradient data were collected at a lower total protein concentration, therefore leading to an increased population of smaller oligomers. In addition, the sucrose gradient data were collected from mammalian cells that may phosphorylate HSP27 and cause elevated dissociation of oligomers. Finally, there is also more than one population of WT HSP27 oligomers detected by the sucrose gradient assay, whereas there is only a single elution peak in the SEC-MALS data. These orthogonal assays highlight the complex nature of HSP27's oligomeric assembly, and no one technique can capture all aspects of this dynamic process on its own.

We have added a sentence to p. 29-30 of the manuscript to discuss the oligomeric populations detected by the SEC-MALS and sucrose gradient data:

*We note that, for both WT and P182L HSP27, two populations of oligomers are observed in the sucrose gradient assay, whereas only a single elution peak is obtained in the SEC-MALS experiments. Observation of a single elution peak by SEC-MALS, however, does not mean that only a single population of oligomers is present: HSP27 and other sHSPs form a polydisperse ensemble of oligomers (Hochberg & Benesch, 2014), and gel filtration elution profiles reflect the ensemble-averaged nature of the sample. In addition, our observation of two populations of oligomers by the sucrose gradient assay is consistent with analytical SEC performed as a function of HSP27 concentration (Jovcevski *et al.*, 2015). At high protein concentrations, only a single elution peak was observed, corresponding to large oligomers; decreasing the total*

protein concentration increased the population of smaller oligomers, and a second elution peak was observed (Jovcevski et al, 2015). Our SEC-MALS experiments were performed at an HSP27 concentration of 40 μ M (monomer), which promotes formation of large oligomers and a single elution peak via gel filtration. By contrast, our sucrose gradient assays contained a significantly lower HSP27 concentration, thereby promoting the formation of small and large oligomers. Finally, the sucrose gradient assay detected HSP27 from mammalian cells that may phosphorylate HSP27 and cause elevated dissociation of oligomers. These orthogonal assays highlight the complex nature of HSP27's oligomeric assembly, and no one technique can capture all aspects of this dynamic process on its own.

We think that this addresses any confusion that the reader might have when assessing these data.

2. Figure 2E: The labels on the y-axis are spaced non-linearly. Please correct this, or in case this is intentional, comment on this.

We thank the reviewer for this comment. Figure 2E displays values of spot intensity on a square root scale. We have added the following text to the Figure 2 legend: “(E) ... *Data were plotted with a square-root transformation of the y-axis to improve the visualization of the complete distribution of intensities.*”

Dear Reid,

Thank you for submitting your revised manuscript. I have now had a chance to take a careful look at the introduced changes and all looks good!

I am therefore very pleased to accept the manuscript for publication here.

with best wishes

Karin

Karin Dumstrei, PhD
Senior Editor
The EMBO Journal

Please note that it is EMBO Journal policy for the transcript of the editorial process (containing referee reports and your response letter) to be published as an online supplement to each paper. If you do NOT want this, you will need to inform the Editorial Office via email immediately. More information is available here: https://emboj.embopress.org/about#Transparent_Process

Your manuscript will be processed for publication in the journal by EMBO Press. Manuscripts in the PDF and electronic editions of The EMBO Journal will be copy edited, and you will be provided with page proofs prior to publication. Please note that supplementary information is not included in the proofs.

Should you be planning a Press Release on your article, please get in contact with embojournal@wiley.com as early as possible, in order to coordinate publication and release dates.

If you have any questions, please do not hesitate to call or email the Editorial Office. Thank you for your contribution to The EMBO Journal.

Corresponding Author Name: Reid Alderson, Justin L. P. Benesch, Andrew J. Baldwin, Vincent Timmerman
Journal Submitted to: EMBO Journal
Manuscript Number: EMBOJ-2019-103811